# Uncertainty Estimation by Flexible Evidential Deep Learning

**Taeseong Yoon    Heeyoung Kim**
Department of Industrial and Systems Engineering, KAIST
{bigstar0423, heeyoungkim}@kaist.ac.kr

## Abstract

Uncertainty quantification (UQ) is crucial for deploying machine learning models in high-stakes applications, where overconfident predictions can lead to serious consequences. An effective UQ method must balance computational efficiency with the ability to generalize across diverse scenarios. Evidential deep learning (EDL) achieves efficiency by modeling uncertainty through the prediction of a Dirichlet distribution over class probabilities. However, the restrictive assumption of Dirichlet-distributed class probabilities limits EDL's robustness, particularly in complex or unforeseen situations. To address this, we propose *flexible evidential deep learning* ($\mathcal{F}$-EDL), which extends EDL by predicting a flexible Dirichlet distribution—a generalization of the Dirichlet distribution—over class probabilities. This approach provides a more expressive and adaptive representation of uncertainty, significantly enhancing UQ generalization and reliability under challenging scenarios. We theoretically establish several advantages of $\mathcal{F}$-EDL and empirically demonstrate its state-of-the-art UQ performance across diverse evaluation settings, including classical, long-tailed, and noisy in-distribution scenarios.

## 1  Introduction

Machine learning models have achieved remarkable predictive performance in diverse fields, including computer vision and natural language processing. However, their deployment in high-stakes applications—autonomous driving, medical diagnosis, and manufacturing—remains limited due to concerns about reliability and overconfident predictions [1, 2]. Robust uncertainty quantification (UQ) is essential for ensuring safer, more trustworthy decision-making in such contexts.

Effective UQ methods must meet two fundamental requirements: (i) computational efficiency, enabling integration into real-time systems, and (ii) generalizability to diverse and unforeseen scenarios. Classical UQ methods—Bayesian neural networks [3], Monte Carlo dropout [4], and deep ensembles [5]—are well established but computationally intensive, as they require multiple forward passes. In response, single forward pass UQ methods [6, 7, 8] have emerged as efficient alternatives, among which evidential deep learning (EDL) [9, 10, 11, 12] stands out. EDL predicts a Dirichlet distribution over class probabilities to quantify uncertainty, leveraging the conjugate prior structure of the Dirichlet distribution and its closed-form uncertainty measures. This approach enables computationally efficient UQ and demonstrates strong performance in downstream tasks, such as out-of-distribution (OOD) detection [12].

Despite these advantages, EDL may struggle to provide robust uncertainty estimates in complex or unforeseen scenarios, leading to suboptimal UQ performance. This is shown in Figure 1(a), which illustrates the epistemic uncertainty distributions for EDL trained on the Dirty-MNIST (DMNIST) dataset [8]. DMNIST combines clean MNIST (blue), representing clean in-distribution (ID) samples, with Ambiguous-MNIST (AMNIST, green), a noisy ID dataset containing ambiguous samples. Additionally, Fashion-MNIST (FMNIST, red) serves as an OOD dataset. To ensure a fair comparison, we

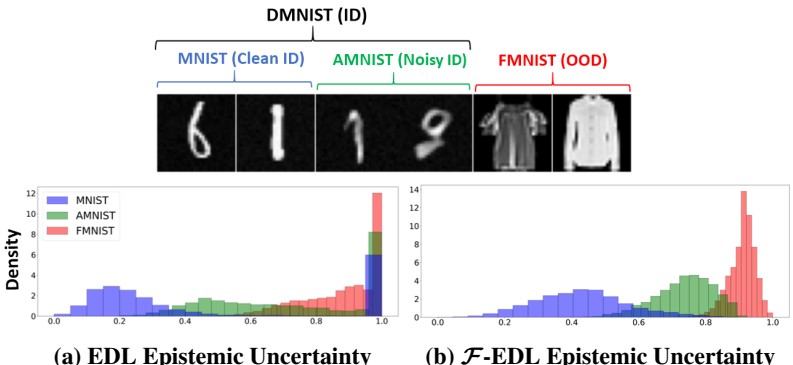

Figure 1: Epistemic uncertainty distributions with DMNIST as the ID dataset. The top row presents sample images from three datasets: MNIST, AMNIST, and FMNIST. Panels (a) and (b) display histograms depicting the epistemic uncertainty distributions obtained by the EDL and $\mathcal{F}$-EDL (proposed) models, respectively, across these datasets.

adopt a standard uncertainty metric used in EDL [9]. An ideal UQ model should assign progressively higher epistemic uncertainty from clean ID to noisy ID and OOD samples, as intended in the DMNIST benchmark [8]. However, EDL exhibits substantial overlap between noisy ID and OOD distributions, and even assigns high uncertainty to some clean ID samples, indicating limited generalization in noisy ID scenarios. In contrast, our proposed model, described below, achieves the desired separation among these datasets, as shown in Figure 1(b), demonstrating robust UQ and improved generalization under noisy conditions. We hypothesize that EDL's limitation stems from its core assumption that the class probabilities for each data point follow a Dirichlet distribution. While this assumption facilitates computational efficiency, it restricts the model's ability to express complex or ambiguous forms of uncertainty. This motivates the need for more flexible yet efficient UQ methods.

Building upon this insight, we propose *flexible evidential deep learning* ($\mathcal{F}$-EDL), a novel UQ framework that extends EDL by predicting a flexible Dirichlet (FD) distribution [13] over class probabilities. The FD distribution generalizes the Dirichlet, enabling a more expressive representation of uncertainty while retaining the computational efficiency of EDL through its conjugate prior—allowing a single forward pass UQ suitable for real-time applications. This expressiveness improves the model's generalizability, enabling robust UQ across diverse and challenging scenarios such as noisy ID data, tail classes, and distribution shifts. Beyond the FD-based structure, $\mathcal{F}$-EDL introduces a tailored objective function for uncertainty-aware classification and label-wise variance-based uncertainty measures for robust, generalizable UQ.

$\mathcal{F}$-EDL introduces several theoretical advancements that strengthen its UQ capabilities. First, we prove that the FD distribution serves as a conjugate prior to the categorical likelihood. Building on this, we prove that $\mathcal{F}$-EDL predicts the posterior FD distribution for each input by employing an improper prior with parameters learned from the data, addressing the limitations of fixed priors in traditional EDL. Second, we prove that $\mathcal{F}$-EDL converges to standard EDL under specific parameter settings, establishing it as a generalization of EDL. Third, we prove that $\mathcal{F}$-EDL can generate multimodal class probability distributions on the simplex, allowing it to capture complex and nuanced uncertainty patterns. Fourth, we prove that $\mathcal{F}$-EDL functions as a mixture model that adaptively combines EDL and softmax predictions using input-dependent mixture weights, demonstrating how it leverages the strengths of both approaches to achieve optimal predictions and enhanced UQ. Finally, we prove that $\mathcal{F}$-EDL admits a *generalized subjective logic* interpretation, modeling uncertainty as a structured mixture of class-specific opinions that capture the model's hesitation among plausible hypotheses.

We empirically evaluate $\mathcal{F}$-EDL across a wide range of UQ-related downstream tasks, including classification, misclassification detection, OOD detection, and distribution shift detection. Notably, $\mathcal{F}$-EDL consistently achieves state-of-the-art performance across diverse settings, including classical, long-tailed, and noisy ID scenarios, highlighting its robustness and generalizability. In addition, qualitative analyses show that $\mathcal{F}$-EDL captures interpretable multimodal uncertainty reflecting ambiguity across plausible classes, while demonstrating faithful epistemic behavior that decreases with more data.

## 2 Preliminaries

### 2.1 Evidential Deep Learning

Evidential deep learning (EDL) quantifies uncertainty in classification tasks by predicting a Dirichlet distribution over class probabilities. Introduced by Sensoy et al. [9], EDL is grounded in Dempster-Shafer Theory of Evidence (DST) [14] and subjective logic (SL) [15]. DST represents subjective beliefs by assigning belief masses to classes, capturing both class likelihoods and uncertainty. SL extends DST by formalizing these subjective beliefs using a Dirichlet distribution—enabling uncertainty-aware classification within a principled framework.

The core innovation of Sensoy et al. [9] is using a neural network to predict Dirichlet parameters. Let $D$ denote the input dimensionality and $K$ denote the number of classes. For a given input $\mathbf{x} \in \mathbb{R}^D$, the class probability distribution is represented as $\boldsymbol{\pi} \sim \mathrm{Dir}(\boldsymbol{\alpha})$, where the concentration parameters $\boldsymbol{\alpha} \in \mathbb{R}^K$ are defined as $\boldsymbol{\alpha} = \mathbf{1} + \mathrm{ReLU}(f_{\boldsymbol{\theta}}(\mathbf{x}))$. Here, $\mathbf{1} = (1, \ldots, 1) \in \mathbb{R}^K$ is a vector of ones, and $f_{\boldsymbol{\theta}} : \mathbb{R}^D \to \mathbb{R}^K$ represents the neural network parameterized by $\boldsymbol{\theta}$. The objective function of EDL for a given data point $(\mathbf{x}, \mathbf{y})$, where $\mathbf{y}$ is the one-hot encoded label, consists of two components: the expected mean squared error (MSE) and a Kullback-Leibler (KL) divergence penalty. It is formulated as follows:

$$\mathcal{L} = \underbrace{\mathbb{E}_{\boldsymbol{\pi} \sim \mathrm{Dir}(\boldsymbol{\alpha})}[\|\mathbf{y} - \boldsymbol{\pi}\|_2^2]}_{\text{Expected MSE over Dirichlet}} + \underbrace{\lambda \mathrm{D_{KL}}[\mathrm{Dir}(\tilde{\boldsymbol{\alpha}}) \,\|\, \mathrm{Dir}(\mathbf{1})]}_{\text{KL divergence penalty}},$$

where $\tilde{\boldsymbol{\alpha}} = \boldsymbol{\alpha} \odot (\mathbf{1} - \mathbf{y}) + \mathbf{y}$, and $\lambda$ is a regularization parameter.

### 2.2 Flexible Dirichlet Distribution

The flexible Dirichlet (FD) distribution [13] is a generalization of the Dirichlet distribution. The FD distribution is derived by normalizing a flexible Gamma (FG) basis. Specifically, the FG basis, denoted as $\mathbf{Y} = (Y_1, \ldots, Y_K)$, is constructed as follows:

$$Y_k = W_k + Z_k\, U, \ \forall k \in [K],$$

where $W_k \sim \mathrm{Gamma}(\alpha_k), \forall k \in [K]$, are independent Gamma random variables, and $U \sim \mathrm{Gamma}(\tau)$ is another independent Gamma random variable sharing the same scale parameter as each $W_k$. Here, the parameters satisfy $\alpha_k > 0, \forall k \in [K]$ and $\tau > 0$. The vector $\mathbf{Z} = (Z_1, \ldots, Z_K)$ follows a multinomial distribution, $\mathbf{Z} \sim \mathrm{Mu}(1, \mathbf{p})$, where $\mathbf{p} = (p_1, \ldots, p_K), 0 \le p_k \le 1, \forall k \in [K]$, and $\sum_{k=1}^{K} p_k = 1$. This FG basis is denoted as $\mathbf{Y} \sim \mathrm{FG}^K(\boldsymbol{\alpha}, \mathbf{p}, \tau)$, where $\boldsymbol{\alpha} = (\alpha_1, \ldots, \alpha_K)$. By combining independent Gamma random variables with a shared random component, the FG basis introduces flexibility, effectively modeling dependencies among components.

The FD distribution, denoted as $\mathbf{X} \sim \mathrm{FD}^K(\boldsymbol{\alpha}, \mathbf{p}, \tau)$ [1] , is defined as $\mathbf{X} = (X_1, \ldots, X_K)$ with $X_k = Y_k / \sum_{k=1}^{K} Y_k, \forall k \in [K]$. Due to its ability to capture complex dependencies and multimodal behavior, the FD distribution is well-suited for compositional data analysis [16, 13]. While previously unexplored for UQ, its flexibility makes it a compelling candidate for modeling uncertainty.

## 3 Flexible Evidential Deep Learning

The $\mathcal{F}$-EDL framework introduces an efficient and generalizable UQ methodology by leveraging the FD distribution to model a distribution over class probabilities. The adoption of the FD distribution in $\mathcal{F}$-EDL represents a principled generalization of EDL, rather than an ad-hoc extension of model flexibility. By introducing additional parameters $(\mathbf{p}, \tau)$, the model governs how evidence is allocated across classes (via $\mathbf{p}$) and concentrated in magnitude (via $\tau$), thereby enabling structured and interpretable uncertainty modeling—particularly for ambiguous inputs where multiple class hypotheses may be simultaneously plausible. Furthermore, the FD-based formulation offers an input-adaptive mechanism for evidence extraction, capturing how uncertainty evolves dynamically in response to the characteristics of each input. A detailed theoretical discussion is provided in Appendix D.

To operationalize this formulation within a neural framework, $\mathcal{F}$-EDL consists of three key components: (i) a distinctive model structure (Section 3.1), (ii) a tailored objective function (Section 3.2), and (iii) label-wise variance-based uncertainty measures (Section 3.3).

---

[1]We refer to this distribution as $\mathrm{FD}$, omitting the superscript when the dimensionality $K$ is clear from the context.

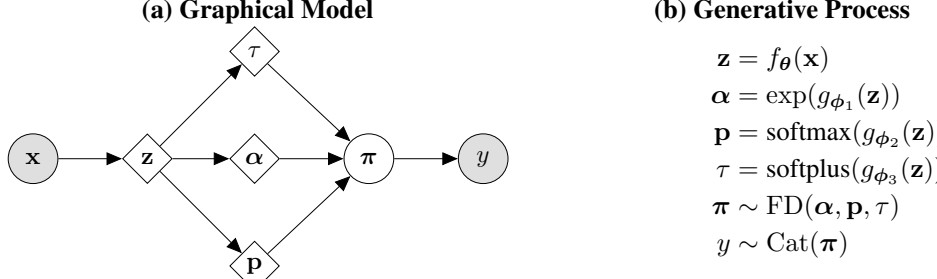

**(a) Graphical Model**

**(b) Generative Process**

$$\mathbf{z} = f_{\boldsymbol{\theta}}(\mathbf{x})$$
$$\boldsymbol{\alpha} = \exp(g_{\phi_1}(\mathbf{z}))$$
$$\mathbf{p} = \mathrm{softmax}(g_{\phi_2}(\mathbf{z}))$$
$$\tau = \mathrm{softplus}(g_{\phi_3}(\mathbf{z}))$$
$$\boldsymbol{\pi} \sim \mathrm{FD}(\boldsymbol{\alpha}, \mathbf{p}, \tau)$$
$$y \sim \mathrm{Cat}(\boldsymbol{\pi})$$

Figure 2: (a) Graphical model and (b) generative process of the proposed $\mathcal{F}$-EDL framework.

## 3.1 Model Structure

Let $\mathbf{x} \in \mathbb{R}^D$ denote an input with dimension $D$, and let $\mathbf{z} = f_{\boldsymbol{\theta}}(\mathbf{x}) \in \mathbb{R}^H$ denote its feature representation, where $f_{\boldsymbol{\theta}} : \mathbb{R}^D \to \mathbb{R}^H$ is a feature extractor parameterized by $\boldsymbol{\theta}$ and $H$ is the feature dimension. From $\mathbf{z}$, the model predicts the parameters of the FD distribution: concentration parameters $\boldsymbol{\alpha} \in \mathbb{R}_+^K$, allocation probabilities $\mathbf{p} \in \Delta^{K-1}$, and dispersion $\tau \in \mathbb{R}_+$, which together define a distribution over class probabilities $\boldsymbol{\pi} \in \Delta^{K-1}$, from which the label $y$ is sampled. These parameters are computed by three neural heads: $g_{\phi_1} : \mathbb{R}^H \to \mathbb{R}^K$, $g_{\phi_2} : \mathbb{R}^H \to \mathbb{R}^K$, and $g_{\phi_3} : \mathbb{R}^H \to \mathbb{R}$, parameterized by $\phi_1$, $\phi_2$, and $\phi_3$, respectively.

To ensure the non-negativity and stability of $\boldsymbol{\alpha}$, we use $\exp$ activation, following prior work [17, 12]. To map $\mathbf{p}$ onto the probability simplex, we use the $\mathrm{softmax}$ function. For $\tau$, we use the $\mathrm{softplus}$ activation for smoother gradients and improved empirical performance. We also apply spectral normalization [18] to $f_{\boldsymbol{\theta}}$ and $g_{\phi_1}$ to improve the robustness of $\boldsymbol{\alpha}$ and enhance the quality of UQ [6, 12]. This normalization further enforces Lipschitz continuity and constrains the magnitudes of network outputs, thereby implicitly regularizing $\boldsymbol{\alpha}$ without requiring explicit penalty terms.

The overall structure is illustrated in Figure 2, with the graphical model in panel (a) and the generative process in panel (b).

## 3.2 Objective Function

The $\mathcal{F}$-EDL framework is optimized using an objective function consisting of two components: (i) the expected MSE with respect to the FD distribution, following standard EDL practice for uncertainty-aware classification, and (ii) a Brier-score-based regularization term for $\mathbf{p}$, to promote input-dependent calibration and prevent degenerate solutions.

For a given data point $(\mathbf{x}, \mathbf{y})$, where $\mathbf{y}$ is the one-hot encoded representation of the label $y$, the objective function is defined as follows:

$$\mathcal{L} = \underbrace{\mathbb{E}_{\boldsymbol{\pi} \sim \mathrm{FD}(\boldsymbol{\alpha}, \mathbf{p}, \tau)}[\|\mathbf{y} - \boldsymbol{\pi}\|_2^2]}_{\text{Expected MSE over FD}} + \underbrace{\|\mathbf{y} - \mathbf{p}\|_2^2}_{\text{Regularization term}} .$$

$\mathcal{F}$-EDL enables analytic training due to the closed-form moments of the FD distribution, eliminating the need for sampling. It also simplifies optimization by reducing sensitivity to hyperparameters, unlike recent EDL methods that require careful tuning [10, 11].

## 3.3 Label-Wise Variance-Based Uncertainty Measures

After training, the predicted label for a testing example $\mathbf{x}^\star$ is determined by assigning the class with the highest expected class probability under the FD distribution: $\hat{y}(\mathbf{x}^\star) = \arg\max_{k \in [K]} \mathbb{E}_{\boldsymbol{\pi} \sim \mathrm{FD}(\boldsymbol{\alpha}, \mathbf{p}, \tau)}[\pi_k]$, where $\boldsymbol{\alpha}$, $\mathbf{p}$, and $\tau$ are predicted parameters for $\mathbf{x}^\star$.

To quantify uncertainty, we adopt a label-wise variance-based approach [19], replacing the Dirichlet with the FD distribution, enabling robust and interpretable UQ consistent with established axioms [20]. For each class $k \in [K]$, the predictive uncertainty for $\mathbf{x}^\star$, denoted by $\mathrm{TU}_k(\mathbf{x}^\star)$, is defined as the variance of the class label: $\mathrm{TU}_k(\mathbf{x}^\star) = \mathrm{Var}(y_k | \mathbf{x}^\star)$. This quantity is decomposed via the law of total

variance into aleatoric and epistemic components, denoted by $\mathrm{AU}_k(\mathbf{x}^\star)$ and $\mathrm{EU}_k(\mathbf{x}^\star)$, respectively:

$$\underbrace{\mathrm{Var}(y_k|\mathbf{x}^\star)}_{\mathrm{TU}_k(\mathbf{x}^\star)} = \underbrace{\mathbb{E}_{\boldsymbol{\pi}}[\mathrm{Var}(y_k|\boldsymbol{\pi},\mathbf{x}^\star)]}_{\mathrm{AU}_k(\mathbf{x}^\star)} + \underbrace{\mathrm{Var}_{\boldsymbol{\pi}}(\mathbb{E}[y_k|\boldsymbol{\pi},\mathbf{x}^\star])}_{\mathrm{EU}_k(\mathbf{x}^\star)}.$$

The total, aleatoric, and epistemic uncertainties for $\mathbf{x}^\star$ are computed by summing the respective class-wise uncertainties and substituting the moments of the FD distribution.

$$\mathrm{EU}(\mathbf{x}^\star) = \sum_{k=1}^{K} \frac{\mathbb{E}_{\boldsymbol{\pi}}[\pi_k](1 - \mathbb{E}_{\boldsymbol{\pi}}[\pi_k])}{(\alpha_0 + \tau + 1)} + \frac{\tau^2 p_k(1 - p_k)}{(\alpha_0 + \tau)(\alpha_0 + \tau + 1)},$$

$$\mathrm{TU}(\mathbf{x}^\star) = 1 - \sum_{k=1}^{K}(\mathbb{E}_{\boldsymbol{\pi}}[\pi_k])^2, \quad \mathrm{AU}(\mathbf{x}^\star) = \mathrm{TU}(\mathbf{x}^\star) - \mathrm{EU}(\mathbf{x}^\star),$$

where $\alpha_0 = \sum_{k=1}^{K}\alpha_k$ and $\mathbb{E}_{\boldsymbol{\pi}}[\pi_k] = \frac{\alpha_k + \tau p_k}{\alpha_0 + \tau}, \forall k \in [K]$.

Detailed derivations of the closed-form expressions for the objective function and uncertainty measures are presented in Appendix A, while the algorithms for $\mathcal{F}$-EDL are provided in Appendix B.

## 4 Theoretical Analysis

$\mathcal{F}$-EDL offers several theoretical advancements that enhance its UQ generalizability. First, we prove that the FD distribution is a conjugate prior to the categorical likelihood (Lemma 4.1). Building on this, we prove that the class probability distribution for a given input $\mathbf{x}$, derived from $\mathcal{F}$-EDL, corresponds to the posterior FD distribution for $\boldsymbol{\pi}$, obtained with an improper prior $\boldsymbol{\pi}(\mathbf{x}) \sim \mathrm{FD}(\mathbf{0}, \mathbf{p}_{\mathrm{prior}}(\mathbf{x}), \tau_{\mathrm{prior}}(\mathbf{x}))$, where the prior parameters $\mathbf{p}_{\mathrm{prior}}(\mathbf{x})$ and $\tau_{\mathrm{prior}}(\mathbf{x})$ are learned from the data (Theorem 4.2).[2] By introducing improper priors with input-dependent parameters, $\mathcal{F}$-EDL eliminates the need for manually specified priors required in traditional EDL.

Second, we prove that the class probability distribution for a given input derived from $\mathcal{F}$-EDL is equivalent to that of standard EDL under specific parameter settings: $\tau = 1$ and $p_k = \alpha_k / \sum_{k=1}^{K}\alpha_k$ for all $k \in [K]$ (Theorem 4.3). This establishes EDL as a special case of $\mathcal{F}$-EDL, demonstrating that our framework retains EDL's strengths while enabling greater expressiveness when needed.

Third, we prove that the class probability distribution for a testing example derived from $\mathcal{F}$-EDL forms a multimodal distribution over the simplex, represented as a mixture of Dirichlet distributions, with the number of distinct modes determined by the cardinality of the allocation probability vector $\mathbf{p}$, denoted as $\|\mathbf{p}\|_0$ (Theorem 4.4). This allows $\mathcal{F}$-EDL to capture complex, multimodal uncertainty—particularly useful for handling ambiguous inputs.

Fourth, we prove that the predictive distribution for a testing example derived from $\mathcal{F}$-EDL can be decomposed into contributions from EDL and softmax, with input-dependent mixture weights (Theorem 4.5). This demonstrates that $\mathcal{F}$-EDL functions as a mixture model of EDL and softmax, adaptively integrating their complementary strengths to achieve optimal performance in both prediction and UQ. Specifically, the EDL component tends to dominate for clean ID data, whereas for ambiguous or OOD inputs, the model interpolates between the two, maintaining robust performance across varying uncertainty levels.

Finally, we prove that $\mathcal{F}$-EDL admits a *generalized subjective logic* interpretation, modeling a mixture of $K$ class-specific subjective opinions (i.e., hypotheses), each defined by a shared base evidence $\boldsymbol{\alpha}$ and dispersion parameter $\tau$, and weighted by the hypothesis selection probabilities $\mathbf{p}$. [3] This perspective reveals that $\mathcal{F}$-EDL performs UQ via principled evidence aggregation grounded in subjective logic [15], rather than simply adding flexibility.

**Lemma 4.1.** *(Conjugacy of FD Prior) Let $y_i \sim \mathrm{Cat}(\boldsymbol{\pi}), \forall i \in [N]$, where $\boldsymbol{\pi} \sim \mathrm{FD}(\boldsymbol{\alpha}, \mathbf{p}, \tau)$. The posterior distribution of $\boldsymbol{\pi}$ given $\{y_i\}_{i=1}^{N}$ is also an FD distribution, $\boldsymbol{\pi}|\{y_i\}_{i=1}^{N} \sim \mathrm{FD}(\boldsymbol{\alpha}', \mathbf{p}, \tau)$, where $\boldsymbol{\alpha}' = (\alpha_1', \ldots, \alpha_K')$ and $\alpha_k' = \alpha_k + \sum_{i=1}^{N} \mathbf{1}_{\{y_i = k\}}, \forall k \in [K]$.*

---

[2]See Appendix C.2 for a more detailed explanation of the Bayesian interpretation of $\mathcal{F}$-EDL.

[3]See Appendix C.3 for a more detailed explanation of the subjective logic interpretation of $\mathcal{F}$-EDL.

**Theorem 4.2.** *(Bayesian Interpretation of $\mathcal{F}$-EDL) The class probability distribution for a given input $\mathbf{x}$, derived from $\mathcal{F}$-EDL, is equivalent to the posterior FD distribution of $\boldsymbol{\pi}$, with likelihood $\sum_{i=1}^{N} \mathbf{1}_{\{y_i=k\}} = (\exp(g_{\phi_1}(f_{\boldsymbol{\theta}}(\mathbf{x}))))_k, \forall k \in [K]$, and an improper prior $\boldsymbol{\pi}(\mathbf{x}) \sim \mathrm{FD}(\mathbf{0}, \mathbf{p}_{\mathrm{prior}}(\mathbf{x}), \tau_{\mathrm{prior}}(\mathbf{x}))$, where $\mathbf{p}_{\mathrm{prior}}(\mathbf{x})$ and $\tau_{\mathrm{prior}}(\mathbf{x})$ are input-dependent parameters learned from the data.*

**Theorem 4.3.** *(Generalization of EDL) The class probability distribution for a testing example $\mathbf{x}^\star$, derived from $\mathcal{F}$-EDL, is equivalent to the class probability distribution of EDL when $\tau = 1$ and $p_k = \alpha_k / \sum_{k=1}^{K} \alpha_k, \forall k \in [K]$.*

**Theorem 4.4.** *(Multimodality of $\mathcal{F}$-EDL)* [4] *The class probability distribution for a testing example $\mathbf{x}^\star$, derived from $\mathcal{F}$-EDL, is expressed as a mixture of Dirichlet distributions:*

$$p_{\mathcal{F}\text{-EDL}}(\boldsymbol{\pi}|\mathbf{x}^\star) = \sum_{k=1}^{K} p_k \, \mathrm{Dir}(\boldsymbol{\pi}|\boldsymbol{\alpha} + \tau \mathbf{e}_k),$$

*where $\mathbf{e}_k$ represents the $k$-th standard basis vector in $\mathbb{R}^K$. The number of distinct modes is determined by the cardinality of $\mathbf{p}$, i.e., $\|\mathbf{p}\|_0$.*

**Theorem 4.5.** *(Predictive Distribution Decomposition) Let $p_{\mathrm{EDL}}(y|\mathbf{x}^\star)$ and $p_{\mathrm{SM}}(y|\mathbf{x}^\star)$ denote the predictive distributions for a testing example $\mathbf{x}^\star$, derived from the EDL and softmax models, respectively. The predictive distribution for $\mathbf{x}^\star$, obtained by the $\mathcal{F}$-EDL framework, is decomposed into the contributions of the EDL and softmax models as follows:*

$$p_{\mathcal{F}\text{-EDL}}(y|\mathbf{x}^\star) = w_{\mathrm{EDL}}(\mathbf{x}^\star) \times p_{\mathrm{EDL}}(y|\mathbf{x}^\star) + w_{\mathrm{SM}}(\mathbf{x}^\star) \times p_{\mathrm{SM}}(y|\mathbf{x}^\star),$$

*where $w_{\mathrm{EDL}}(\mathbf{x}^\star) = \alpha_0/(\alpha_0 + \tau)$ and $w_{\mathrm{SM}}(\mathbf{x}^\star) = \tau/(\alpha_0 + \tau)$ are input-dependent mixture weights.*

**Proposition 4.6.** *(Subjective Logic Interpretation of $\mathcal{F}$-EDL) $\mathcal{F}$-EDL admits a generalized subjective logic interpretation, representing $K$ class-specific subjective opinions (i.e., hypotheses) defined by shared base evidence $\boldsymbol{\alpha}$, dispersion parameter $\tau$, and hypothesis selection probabilities $\mathbf{p}$.*

For the proofs and additional insights about the theorems, refer to Appendix C.

## 5   Related Works

**EDL methods.** EDL [21] refers to a family of UQ methods that predict a Dirichlet distribution over class probabilities to estimate uncertainty in classification. KL-PN [22] encourages concentrated Dirichlets for ID data and uniform Dirichlets for OOD data via KL divergence, while RKL-PN [23] uses reverse KL to mitigate issues with unintended multimodal distributions. However, both rely on OOD data, which is often unavailable in practice. To mitigate the reliance on OOD data, PostNet [24] and NatPN [25] leverage feature space density estimated using Normalizing Flow [26] to predict the posterior Dirichlet distribution. However, their performance heavily depends on accurate density estimation, which remains challenging in high-dimensional settings.

Building on EDL [9], several methods have attempted to address its limitations. $\mathcal{I}$-EDL [10] incorporates Fisher information to account for varying levels of data uncertainty. R-EDL [11] treats a prior parameter as a tunable hyperparameter and simplifies the loss by removing the variance term, thereby relaxing EDL's restrictive assumption—a motivation shared with our approach. DAEDL [12] uses feature space density during prediction with an alternative parameterization. However, all these methods remain constrained by the Dirichlet assumption, limiting their generalization to complex scenarios. Separately, distillation-based approaches [27, 28, 29] improve epistemic uncertainty estimation but typically require multiple forward passes, reducing their practicality. Beyond standard classification, EDL methods have also been extended to diverse tasks, including regression [30], domain adaptation [31], semantic segmentation [32, 33], calibration of large language models [34], and multi-view learning [35, 36, 37, 38]

On the other hand, recent studies have questioned the ability of EDL to represent epistemic uncertainty. Bengs et al. [39, 40] showed that its second-order loss is not a strictly proper scoring rule, leading to non-vanishing uncertainty. Jürgens et al. [41] found that regularized EDL enforces a fixed uncertainty

---

[4]Adapted from Proposition 4.1 of [13].

Table 1: UQ-related downstream task results are reported using the CIFAR-10 and CIFAR-100 datasets as the ID dataset. "Test.Acc." denotes the test accuracy, "Conf." denotes the AUPR scores for misclassification detection based on aleatoric uncertainty, and "SVHN / C-100" and "SVHN / TIN" indicate AUPR scores for OOD detection based on epistemic uncertainty, where the former corresponds to using CIFAR-10 as the ID dataset with SVHN and CIFAR-100 as the OOD datasets, and the latter corresponds to using CIFAR-100 as the ID dataset with SVHN and TinyImageNet as the OOD datasets. Baseline results for CIFAR-10 are sourced from existing literature [10, 11, 12]. Bold numbers indicate the best performance for each metric.

| | ID: CIFAR-10 | | | ID: CIFAR-100 | | |
|---|---|---|---|---|---|---|
| Method | Test.Acc. | Conf. | SVHN / C-100 | Test.Acc. | Conf. | SVHN / TIN |
| Dropout | $82.84_{\pm 0.1}$ | $97.15_{\pm 0.0}$ | $51.39_{\pm 0.1}$ / $45.57_{\pm 1.0}$ | $65.94_{\pm 0.6}$ | $92.00_{\pm 0.3}$ | $71.83_{\pm 2.0}$ / $74.93_{\pm 0.6}$ |
| EDL | $83.55_{\pm 0.6}$ | $97.86_{\pm 0.2}$ | $79.12_{\pm 3.7}$ / $84.18_{\pm 0.7}$ | $45.91_{\pm 5.6}$ | $91.28_{\pm 0.8}$ | $56.21_{\pm 3.1}$ / $70.13_{\pm 2.0}$ |
| $\mathcal{I}$-EDL | $89.20_{\pm 0.3}$ | $98.72_{\pm 0.1}$ | $82.96_{\pm 2.2}$ / $84.84_{\pm 0.6}$ | $66.38_{\pm 0.5}$ | $92.84_{\pm 0.1}$ | $67.51_{\pm 2.9}$ / $75.86_{\pm 0.3}$ |
| R-EDL | $90.09_{\pm 0.3}$ | $98.98_{\pm 0.1}$ | $85.00_{\pm 1.2}$ / $87.73_{\pm 0.3}$ | $63.53_{\pm 0.5}$ | $92.69_{\pm 0.2}$ | $61.80_{\pm 3.4}$ / $69.78_{\pm 1.3}$ |
| DAEDL | $91.11_{\pm 0.2}$ | $99.08_{\pm 0.0}$ | $85.54_{\pm 1.4}$ / $88.19_{\pm 0.1}$ | $66.01_{\pm 2.6}$ | $86.00_{\pm 0.3}$ | $72.07_{\pm 4.1}$ / $77.40_{\pm 1.6}$ |
| $\mathcal{F}$-EDL | $\mathbf{91.19}_{\pm 0.2}$ | $\mathbf{99.10}_{\pm 0.0}$ | $\mathbf{91.20}_{\pm 1.3}$ / $\mathbf{88.37}_{\pm 0.3}$ | $\mathbf{69.40}_{\pm 0.2}$ | $\mathbf{94.01}_{\pm 0.1}$ | $\mathbf{75.35}_{\pm 2.3}$ / $\mathbf{80.58}_{\pm 0.2}$ |

Table 2: AUPR scores for distribution shift detection from CIFAR-10 to CIFAR-10-C using aleatoric uncertainty estimates. $\mathcal{C} \in \{1, 2, 3, 4, 5\}$ denotes severity levels of corruption in CIFAR-10-C, averaged across 19 corruption types. Baseline results are sourced from [12].

| Method | $\mathcal{C} = 1$ | $\mathcal{C} = 2$ | $\mathcal{C} = 3$ | $\mathcal{C} = 4$ | $\mathcal{C} = 5$ |
|---|---|---|---|---|---|
| MSP | $56.39_{\pm 0.7}$ | $61.88_{\pm 1.1}$ | $65.86_{\pm 1.3}$ | $69.91_{\pm 1.5}$ | $75.01_{\pm 1.8}$ |
| EDL | $54.76_{\pm 0.3}$ | $59.01_{\pm 0.4}$ | $62.46_{\pm 0.5}$ | $65.87_{\pm 0.6}$ | $70.21_{\pm 0.8}$ |
| $\mathcal{I}$-EDL | $56.33_{\pm 0.2}$ | $61.52_{\pm 0.5}$ | $65.44_{\pm 0.5}$ | $69.45_{\pm 0.5}$ | $74.56_{\pm 0.5}$ |
| R-EDL | $57.37_{\pm 0.5}$ | $62.20_{\pm 1.0}$ | $65.74_{\pm 1.4}$ | $69.33_{\pm 1.9}$ | $73.58_{\pm 2.6}$ |
| DAEDL | $57.89_{\pm 0.3}$ | $63.23_{\pm 0.4}$ | $67.53_{\pm 0.4}$ | $72.21_{\pm 0.4}$ | $77.74_{\pm 0.4}$ |
| $\mathcal{F}$-EDL | $\mathbf{59.01}_{\pm 0.8}$ | $\mathbf{65.11}_{\pm 0.7}$ | $\mathbf{69.48}_{\pm 0.5}$ | $\mathbf{73.88}_{\pm 0.3}$ | $\mathbf{78.72}_{\pm 0.4}$ |

budget, yielding relative rather than absolute uncertainty measures. Shen et al. [29] further unified these critiques, revealing that the EDL objective collapses to a sample-size-independent Dirichlet target, behaving more like energy-based OOD detectors than true uncertainty estimators. However, $\mathcal{F}$-EDL empirically mitigates these issues through a more expressive FD-based formulation and distinct loss design, yielding better-calibrated epistemic uncertainty in practice.

**Deterministic uncertainty methods.** Beyond EDL, deterministic uncertainty methods (DUMs) [32] offer computationally efficient UQ in a single forward pass. DUMs are designed with two primary objectives: (i) applying regularization techniques to learn feature representations that distinguish between ID and OOD data, and (ii) quantifying uncertainty based on these regularized representations. Regularization methods include spectral normalization [6, 42, 43, 8, 44] and gradient penalties [7]. For uncertainty estimation, DUMs employ diverse strategies, including Gaussian processes [6, 42], radial basis function networks [7], Gaussian discriminant analysis [8], and kernel density estimation [43]. However, these methods often require significant modifications to the neural network architecture, hindering their practical applicability.

# 6 Experiments

We conducted comprehensive experiments to evaluate the generalizability and effectiveness of $\mathcal{F}$-EDL in UQ. Our evaluation consists of two parts. First, we performed a quantitative study (Section 6.2) across diverse ID scenarios—including classical, long-tailed, and noisy settings—focusing on UQ-related downstream tasks. We further conducted an ablation study to assess the contribution of each FD parameter. Second, we performed a qualitative analysis (Section 6.3) examining whether $\mathcal{F}$-EDL produces semantically meaningful uncertainty representations for ambiguous samples and whether its epistemic uncertainty decreases with increasing training data, as theoretically expected. The code for our model is publicly available at https://github.com/TaeseongYoon/F-EDL.

## 6.1 Overview of Experiments

**Tasks.** For each scenario, we conducted three primary UQ-related downstream tasks: (i) classification, (ii) misclassification detection, and (iii) OOD detection. In the classical setting, we additionally

Table 3: UQ-related downstream task results are reported using the CIFAR-10-LT dataset as the ID dataset under both heavily ($\rho = 0.01$) and mildly ($\rho = 0.1$) imbalanced settings.

| Method | Heavy Imbalance ($\rho = 0.01$) | | | Mild Imbalance ($\rho = 0.1$) | | |
|---|---|---|---|---|---|---|
| | Test.Acc. | Conf. | SVHN / C-100 | Test.Acc. | Conf. | SVHN / C-100 |
| Dropout | 39.22 $_{\pm3.1}$ | 63.62 $_{\pm2.7}$ | 33.33 $_{\pm1.7}$ / 54.17 $_{\pm1.1}$ | 70.87 $_{\pm3.0}$ | 89.82 $_{\pm2.3}$ | 37.37 $_{\pm1.4}$ / 61.18 $_{\pm1.3}$ |
| EDL | 42.62 $_{\pm2.7}$ | 82.63 $_{\pm1.7}$ | 51.99 $_{\pm3.8}$ / 66.86 $_{\pm0.9}$ | 79.09 $_{\pm0.4}$ | 95.36 $_{\pm0.1}$ | 72.18 $_{\pm2.1}$ / 80.09 $_{\pm0.7}$ |
| $\mathcal{I}$-EDL | 57.88 $_{\pm1.3}$ | 84.10 $_{\pm1.3}$ | 52.85 $_{\pm6.8}$ / 69.19 $_{\pm1.3}$ | 84.86 $_{\pm0.1}$ | 97.31 $_{\pm0.2}$ | 79.83 $_{\pm3.9}$ / 83.50 $_{\pm0.4}$ |
| R-EDL | 63.36 $_{\pm1.0}$ | 78.34 $_{\pm1.0}$ | 48.71 $_{\pm7.1}$ / 64.20 $_{\pm1.4}$ | 85.35 $_{\pm0.2}$ | 94.35 $_{\pm0.2}$ | 60.58 $_{\pm5.0}$ / 69.53 $_{\pm1.6}$ |
| DAEDL | 63.36 $_{\pm1.4}$ | 82.15 $_{\pm1.0}$ | 51.03 $_{\pm5.6}$ / 65.31 $_{\pm1.2}$ | 84.95 $_{\pm0.4}$ | 95.22 $_{\pm0.4}$ | 69.40 $_{\pm4.5}$ / 74.56 $_{\pm1.7}$ |
| $\mathcal{F}$-EDL | **63.73** $_{\pm1.4}$ | **85.99** $_{\pm1.7}$ | **62.56** $_{\pm2.8}$ / **70.18** $_{\pm2.0}$ | **85.46** $_{\pm0.2}$ | **97.60** $_{\pm0.1}$ | **85.36** $_{\pm1.5}$ / **83.64** $_{\pm0.7}$ |

Table 4: UQ-related downstream task results are reported using the DMNIST dataset as the ID dataset. "FMNIST" denotes AUPR scores for OOD detection based on epistemic uncertainty, with FMNIST as the OOD dataset.

| Method | Test.Acc. | Conf. | FMNIST |
|---|---|---|---|
| MSP | 83.90 $_{\pm0.1}$ | 96.01 $_{\pm0.0}$ | 98.10 $_{\pm0.3}$ |
| Dropout | 84.13 $_{\pm0.1}$ | 96.09 $_{\pm0.0}$ | 94.68 $_{\pm0.6}$ |
| DDU | 84.05 $_{\pm0.1}$ | 82.73 $_{\pm0.1}$ | 98.49 $_{\pm0.4}$ |
| EDL | 77.37 $_{\pm5.8}$ | 95.19 $_{\pm0.3}$ | 92.23 $_{\pm1.7}$ |
| $\mathcal{I}$-EDL | 83.46 $_{\pm0.2}$ | 95.58 $_{\pm0.0}$ | 94.11 $_{\pm0.9}$ |
| R-EDL | 83.41 $_{\pm0.1}$ | 95.58 $_{\pm0.1}$ | 90.91 $_{\pm5.6}$ |
| DAEDL | 84.12 $_{\pm0.1}$ | 95.93 $_{\pm0.0}$ | 99.44 $_{\pm0.2}$ |
| $\mathcal{F}$-EDL | **84.28** $_{\pm0.1}$ | **96.17** $_{\pm0.1}$ | **99.76** $_{\pm0.1}$ |

Table 5: Ablation study on $\mathcal{F}$-EDL using the DMNIST dataset as the ID dataset. "Fix-$\mathbf{p}$ (U), $\tau$" and "Fix-$\mathbf{p}$ (N), $\tau$" fix $\mathbf{p}$ to a uniform vector or normalized concentration parameters, respectively, and fix $\tau = 1$. "Fix-$\mathbf{p}$ (U)" and "Fix-$\mathbf{p}$ (N)" fix only $\mathbf{p}$; "Fix-$\tau$" fixes only $\tau$. The full model learns both.

| Variant | Test.Acc. | Conf. | FMNIST |
|---|---|---|---|
| Fix-$\mathbf{p}$ (U), $\tau$ | 83.34 $_{\pm0.2}$ | 95.62 $_{\pm0.1}$ | 97.22 $_{\pm1.0}$ |
| Fix-$\mathbf{p}$ (N), $\tau$ | 83.27 $_{\pm0.1}$ | 95.59 $_{\pm0.4}$ | 97.91 $_{\pm1.3}$ |
| Fix-$\mathbf{p}$ (U) | 83.39 $_{\pm0.1}$ | 95.60 $_{\pm0.1}$ | 96.94 $_{\pm1.1}$ |
| Fix-$\mathbf{p}$ (N) | 83.32 $_{\pm0.1}$ | 95.61 $_{\pm0.3}$ | 97.48 $_{\pm1.8}$ |
| Fix-$\tau$ | 83.39 $_{\pm0.1}$ | 95.60 $_{\pm0.1}$ | 98.46 $_{\pm0.1}$ |
| $\mathcal{F}$-EDL | **84.28** $_{\pm0.1}$ | **96.17** $_{\pm0.1}$ | **99.76** $_{\pm0.1}$ |

included distribution shift detection. Classification performance was assessed using test accuracy. Misclassification detection was evaluated using the area under the precision-recall curve (AUPR) score, with confidence defined as negative aleatoric uncertainty for ID samples (label: correct = 1, incorrect = 0). OOD detection was also evaluated using AUPR, based on negative epistemic uncertainty (labels: ID = 1, OOD = 0), assessing whether the model assigns higher uncertainty to OOD examples. Distribution shift detection was formulated as OOD detection by applying varying levels of corruption to a base dataset to simulate shifts. All AUPR scores were normalized to a 0-100 scale.

**Datasets.** In the classical setting, CIFAR-10 and CIFAR-100 [45] were used as the primary ID datasets. For the long-tailed setting, we used CIFAR-10-LT [46], an artificially imbalanced version of CIFAR-10, as the ID dataset. For the noisy setting, DMNIST, a variant of MNIST containing ambiguous data points, was used as the ID dataset. For OOD detection, SVHN [47] and CIFAR-100 served as OOD datasets when CIFAR-10 or CIFAR-10-LT was used as ID, while SVHN and TinyImageNet (TIN) were used as the OOD datasets for CIFAR-100. FMNIST was used as the OOD dataset for DMNIST. For distribution shift detection, we utilized CIFAR-10-C [48], a dataset created by applying continuous distribution shifts to CIFAR-10, as the OOD dataset.

**Baselines.** We compared $\mathcal{F}$-EDL against classical and state-of-the-art EDL methods, including EDL [9], $\mathcal{I}$-EDL [10], R-EDL [11], and DAEDL [12]. To highlight task difficulty, we also included Dropout [4] as a traditional UQ baseline. For distribution shift detection, we compared with MSP [48], a widely used benchmark. For the noisy setting, we evaluated both MSP and DDU [8], with DDU specifically introduced for DMNIST. All baselines were evaluated following the uncertainty estimation protocols described in their original papers.

**Implementation.** We adopted VGG-16 [49] for CIFAR-10 and CIFAR-10-LT, and ResNet-18 [50] for CIFAR-100. For DMNIST, we used a lightweight CNN with three convolutional and three dense layers. To predict the FD parameters $\mathbf{p}$ and $\tau$, we added two shallow MLPs with 1-2 layers depending on the dataset, introducing minimal overhead (e.g., 1.8% for VGG-16). At inference time, $\mathcal{F}$-EDL remains highly efficient without any post-hoc processing, running only 1.3% slower than EDL yet over 50% faster than DAEDL on CIFAR-10.

For clarity and brevity, we defer additional experimental explanations (Appendix E), extended results (Appendix F), and supplementary qualitative analyses (Appendix G) to the appendix.

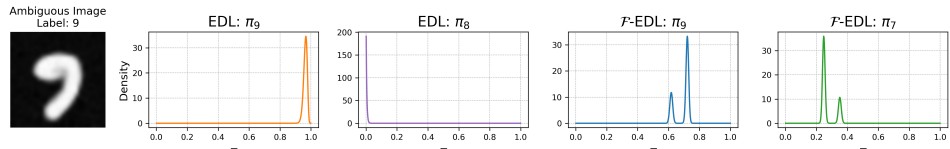

Figure 3: Posterior class probability distributions for a DMNIST input (true label: 9), which is visually ambiguous and resembles class 7. The leftmost panel shows the input image. The second and third panels display the marginal distributions of the two most probable classes predicted by EDL (denoted as $\pi_k$ for class $k$), while the fourth and fifth panels show those predicted by $\mathcal{F}$-EDL.

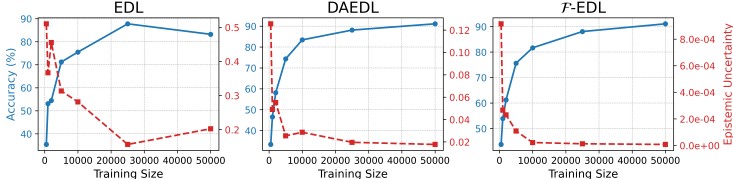

Figure 4: Test accuracy (blue, solid line) and epistemic uncertainty (red, dashed line) of EDL (left), DAEDL (middle), and $\mathcal{F}$-EDL (right) across increasing training set sizes: {500, 1000, 2000, 5000, 10000, 25000, 50000}. Accuracy is plotted on the left y-axis, and uncertainty is plotted on the right y-axis in each subplot.

## 6.2 Quantitative Study: UQ-Related Downstream Tasks across Diverse Settings

**UQ in Classical Setting.** We first evaluate $\mathcal{F}$-EDL in the classical setting, a standard benchmark for UQ models, using CIFAR-10 and CIFAR-100 across four UQ-related downstream tasks described in Section 6.1. As shown in Table 1 and Table 2, $\mathcal{F}$-EDL consistently outperforms competitive baselines across all tasks and datasets. These improvements in UQ performance are broadly attributed to two key innovations: (i) enhanced expressiveness enabled by generalizing beyond the Dirichlet family (Theorem 4.3), and (ii) the use of input-dependent improper priors that address the prior specification issue (Theorem 4.2). In addition, its strong performance in distribution shift detection reflects the model's ability to effectively combine the complementary strengths of softmax-based (MSP) and evidential (EDL) approaches to achieve optimal UQ (Theorem 4.5).

**UQ in Long-Tailed Setting.** To evaluate robustness under class imbalance, we further test $\mathcal{F}$-EDL on CIFAR-10-LT with two levels of imbalance: (i) mild ($\rho = 0.1$) and (ii) heavy ($\rho = 0.01$), across the three tasks described in Section 6.1. Effective UQ in long-tailed settings requires producing calibrated uncertainty estimates under class imbalance to detect misclassified examples and to distinguish rare ID inputs from OOD data. These capabilities were assessed through misclassification detection and OOD detection tasks, respectively. As shown in Table 3, $\mathcal{F}$-EDL demonstrates strong and consistent performance across all tasks and imbalance levels. This performance highlights $\mathcal{F}$-EDL's ability to provide reliable uncertainty estimates under complex distributional structures—a benefit attributed to its flexibility beyond the Dirichlet assumption (Theorem 4.3).

**UQ in Noisy Setting.** To evaluate robustness in noisy ID scenarios, we apply $\mathcal{F}$-EDL to the DMNIST dataset across the three tasks described in Section 6.1. Effective UQ in such settings requires leveraging aleatoric uncertainty to detect ambiguous inputs that are prone to misclassification and epistemic uncertainty to distinguish ID from OOD data. These capabilities were assessed via misclassification detection and OOD detection, respectively. As shown in Table 4, $\mathcal{F}$-EDL consistently achieves strong performance across all tasks. This performance highlights $\mathcal{F}$-EDL's improved ability to handle ambiguous inputs more effectively by representing multimodal opinions over plausible classes in a structured manner (Theorem 4.4 and Proposition 4.6).

**Ablation Study: Contributions of p and $\tau$.** To assess the contribution of additional parameters introduced in the FD distribution, we conduct an ablation study in the noisy ID setting using the DMNIST dataset. Specifically, we compare the full $\mathcal{F}$-EDL model to several simplified variants: two configurations that fix **p**—one to a uniform distribution ($\mathbf{p} = \mathbf{1}/K$) and another to the normalized concentration parameters ($\mathbf{p} = \boldsymbol{\alpha}/\|\boldsymbol{\alpha}\|_1$)—and another configuration that fixes $\tau = 1$. Combinations where both **p** and $\tau$ are fixed are also evaluated. As shown in Table 5, all fixed variants degrade in UQ performance, with "Fix-$\tau$" showing favorable trade-offs, but no single variant consistently outperforms the others. The full model, which learns both **p** and $\tau$, achieves the best results,

suggesting that the performance gains arise from the synergistic flexibility of the FD distribution, rather than merely increased capacity.

### 6.3 Qualitative Study: Multimodality and Faithful Epistemic Behavior

**Multimodal Uncertainty Representations for Ambiguous Inputs.** To assess whether the flexibility of $\mathcal{F}$-EDL leads to semantically meaningful uncertainty representations, we visualize class probability distributions for ambiguous inputs from the DMNIST dataset, on which the model was trained. For each test input, we extract the marginal distributions of the two most probable classes under both EDL and $\mathcal{F}$-EDL. As shown in Figure 3, EDL collapses the prediction into a single, overconfident mode, skewed toward one class, failing to capture the underlying uncertainty, with the second most probable class often misaligned with any visually plausible alternative. In contrast, $\mathcal{F}$-EDL produces multimodal distributions with distinct peaks near plausible classes—such as "7" and "9"—reflecting the model's hesitation in the presence of ambiguity. These qualitative improvements highlight $\mathcal{F}$-EDL's ability to express structured, multimodal representations, in line with its theoretical grounding (Theorem 4.4, Proposition 4.6).

**Faithfulness of Epistemic Uncertainty Estimates.** To evaluate whether $\mathcal{F}$-EDL provides faithful estimates of epistemic uncertainty, we examine how uncertainty evolves as the training dataset grows. A key characteristic of epistemic uncertainty is that it should decrease with more data, reflecting increased model confidence as knowledge improves [29]. Following the evaluation protocol proposed in [29], we train models on progressively larger subsets of CIFAR-10 and compute the average epistemic uncertainty over a fixed test set. As shown in Figure 4, $\mathcal{F}$-EDL demonstrates a clear, monotonic decrease in uncertainty with more data, unlike EDL and DAEDL, which show inconsistent trends. This suggests that $\mathcal{F}$-EDL successfully captures the theoretically desirable behavior wherein epistemic uncertainty consistently decreases as more data become available—empirically addressing a key limitation of standard EDL, whose epistemic uncertainty often fails to exhibit a reliable monotonic decline [29].

## 7 Conclusion

**Summary.** We propose $\mathcal{F}$-EDL, a novel UQ framework that maintains the efficiency of EDL while enhancing its generalizability by predicting a more expressive FD distribution over class probabilities. Theoretically, $\mathcal{F}$-EDL possesses several desirable properties that support reliable and robust UQ under complex conditions. Empirically, it achieves state-of-the-art performance across a variety of UQ-related downstream tasks in diverse and challenging ID scenarios, supported by qualitative evidence of interpretable multimodal predictions and faithful epistemic uncertainty.

**Limitations & Future Directions.** Despite its improved flexibility, $\mathcal{F}$-EDL faces several open challenges. First, it is currently limited to classification; extending it to regression, for instance, by building on evidential regression models [30], is a natural next step. Second, although $\mathcal{F}$-EDL provides a variance-based decomposition of uncertainty, it does not fully disentangle aleatoric and epistemic components—highlighting the need for further work on structured disentanglement, a longstanding challenge in UQ [51, 52]. Third, while $\mathcal{F}$-EDL empirically alleviates several theoretical limitations of EDL, it still relies on external regularization to control epistemic uncertainty [39], suggesting the need for an intrinsically stable training objective.

Beyond these aspects, $\mathcal{F}$-EDL opens multiple promising research avenues. First, it can serve as a foundation for developing UQ methods tailored to specific ID scenarios—such as long-tailed classification [53, 54, 55, 56], where combining $\mathcal{F}$-EDL with logit adjustment [57, 58, 59] could improve calibration under imbalance. Second, its modular architecture supports plug-and-play integration into existing EDL-based frameworks, allowing the Dirichlet components to be replaced with the more expressive FD formulation in downstream applications—such as trusted multi-view learning [35]—through the design of compatible evidence-fusion mechanisms.

## Acknowledgments and Disclosure of Funding

This work was supported by the National Research Foundation of Korea (NRF) grant funded by the Korea government (MSIT) (2023R1A2C2005453, RS-2023-00218913).

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

# Appendix for the Paper
## "Uncertainty Estimation by Flexible Evidential Deep Learning"

# A Additional Explanations about Objective Function and Uncertainty Measures

In this section, we present the closed-form representations for the formulas discussed in the main text. First, we derive the closed-form representation of the objective function for $\mathcal{F}$-EDL, as provided in Section 3.2. Next, we introduce the formulas for total, aleatoric, and epistemic uncertainty using variance-based uncertainty measures and derive the corresponding closed-form representations for $\mathcal{F}$-EDL, as outlined in Section 3.3.

## A.1 Objective Function

Let $\mathbf{y} = (y_1, \ldots, y_K)$ denote the one-hot encoded labels, and let $\boldsymbol{\pi} = (\pi_1, \ldots, \pi_K)$ represent the class probabilities. Furthermore, let $\boldsymbol{\alpha} = (\alpha_1, \ldots, \alpha_K)$, $\mathbf{p} = (p_1, \ldots, p_K)$, and $\tau$ denote the parameters of the FD distribution. The objective function is defined as:

$$\mathcal{L} = \underbrace{\mathbb{E}_{\boldsymbol{\pi} \sim \text{FD}(\boldsymbol{\alpha}, \mathbf{p}, \tau)}[\|\mathbf{y} - \boldsymbol{\pi}\|_2^2]}_{\mathcal{L}^{\text{MSE}}} + \underbrace{\|\mathbf{y} - \mathbf{p}\|_2^2}_{\mathcal{L}^{\text{reg}}},$$

where the first term, $\mathcal{L}^{\text{MSE}}$, represents the expected MSE over the FD distribution, and the second term, $\mathcal{L}^{\text{reg}}$, is a Brier-score-based regularization term for $\mathbf{p}$.

**Expected MSE over FD.** In the first term, the expected MSE can be expressed as:

$$\mathcal{L}^{\text{MSE}} = \mathbb{E}_{\boldsymbol{\pi}}[\|\mathbf{y} - \boldsymbol{\pi}\|_2^2].$$

Expanding the squared difference gives:

$$\mathcal{L}^{\text{MSE}} = \|\mathbf{y}\|_2^2 - 2(\mathbb{E}_{\boldsymbol{\pi}}[\boldsymbol{\pi}])^T \mathbf{y} + \mathbb{E}_{\boldsymbol{\pi}}[\|\boldsymbol{\pi}\|_2^2].$$

Rewriting this as a summation over $k$:

$$\mathcal{L}^{\text{MSE}} = \sum_{k=1}^{K} \left( y_k^2 - 2\mathbb{E}_{\boldsymbol{\pi}}[\pi_k] y_k + \mathbb{E}_{\boldsymbol{\pi}}[\pi_k^2] \right).$$

Using the relationship $\mathbb{E}_{\boldsymbol{\pi}}[\pi_k^2] = (\mathbb{E}_{\boldsymbol{\pi}}[\pi_k])^2 + \text{Var}_{\boldsymbol{\pi}}(\pi_k)$, this simplifies to:

$$\mathcal{L}^{\text{MSE}} = \sum_{k=1}^{K} (y_k - \mathbb{E}_{\boldsymbol{\pi}}[\pi_k])^2 + \sum_{k=1}^{K} \text{Var}_{\boldsymbol{\pi}}(\pi_k).$$

The mean of the FD distribution, $\mathbb{E}_{\boldsymbol{\pi}}[\pi_k]$, is given by:

$$\mathbb{E}_{\boldsymbol{\pi}}[\pi_k] = \frac{\alpha_k + \tau p_k}{\alpha_0 + \tau}, \forall k \in [K],$$

where $\alpha_0 = \sum_{k=1}^{K} \alpha_k$.

The variance of the FD distribution, $\text{Var}_{\boldsymbol{\pi}}[\pi_k]$, is given by:

$$\text{Var}_{\boldsymbol{\pi}}(\pi_k) = \frac{(\alpha_k + \tau p_k)(\alpha_0 - \alpha_k + \tau(1 - p_k))}{(\alpha_0 + \tau)^2 (\alpha_0 + \tau + 1)} + \frac{\tau^2 p_k (1 - p_k)}{(\alpha_0 + \tau)(\alpha_0 + \tau + 1)}, \forall k \in [K].$$

By substituting the mean and variance of the FD distribution, $\mathcal{L}^{\text{MSE}}$ becomes:

$$\mathcal{L}^{\text{MSE}} = \sum_{k=1}^{K} \left( y_k - \frac{\alpha_k + p_k \tau}{\alpha_0 + \tau} \right)^2 + \sum_{k=1}^{K} \frac{(\alpha_k + \tau p_k)(\alpha_0 - \alpha_k + \tau(1 - p_k))}{(\alpha_0 + \tau)^2 (\alpha_0 + \tau + 1)} + \frac{\tau^2 p_k (1 - p_k)}{(\alpha_0 + \tau)(\alpha_0 + \tau + 1)}.$$

**Regularization Term.** The second term, representing the regularization term for $\mathbf{p}$, is given by:

$$\mathcal{L}^{\text{reg}} = \|\mathbf{y} - \mathbf{p}\|_2^2 = \sum_{k=1}^{K} (y_k - p_k)^2.$$

**Combined Objective Function.** Combining the two terms, the complete objective function becomes:

$$\mathcal{L} = \mathcal{L}^{\text{MSE}} + \mathcal{L}^{\text{reg}}$$

$$= \sum_{k=1}^{K} \left( y_k - \frac{\alpha_k + p_k \tau}{\alpha_0 + \tau} \right)^2 + \sum_{k=1}^{K} \frac{(\alpha_k + \tau p_k)(\alpha_0 - \alpha_k + \tau(1 - p_k))}{(\alpha_0 + \tau)^2 (\alpha_0 + \tau + 1)} + \frac{\tau^2 p_k (1 - p_k)}{(\alpha_0 + \tau)(\alpha_0 + \tau + 1)}$$

$$+ \sum_{k=1}^{K} (y_k - p_k)^2.$$

## A.2 Uncertainty Measures

The closed-form expressions for the total, aleatoric, and epistemic uncertainties for a testing example $\mathbf{x}^\star$ can be derived using the variance-based uncertainty measures and the moments of the FD distribution. Below, we present the derivations for these uncertainty measures.

**Total Uncertainty.** The predictive uncertainty for each class $k \in [K]$, for a testing example $\mathbf{x}^\star$, denoted as $\text{TU}_k(\mathbf{x}^\star)$, is defined as the variance of the label for that class:

$$\text{TU}_k(\mathbf{x}^\star) = \text{Var}(y_k | \mathbf{x}^\star).$$

The total uncertainty for $\mathbf{x}^\star$, denoted as $\text{TU}(\mathbf{x}^\star)$, is obtained by summing the class-wise uncertainties:

$$\text{TU}(\mathbf{x}^\star) = \sum_{k=1}^{K} \text{Var}(y_k | \mathbf{x}^\star).$$

Using the law of total variance, this can be decomposed as:

$$\text{TU}(\mathbf{x}^\star) = \sum_{k=1}^{K} \mathbb{E}_{\boldsymbol{\pi}}[\text{Var}(y_k | \boldsymbol{\pi}, \mathbf{x}^\star)] + \sum_{k=1}^{K} \text{Var}_{\boldsymbol{\pi}}(\mathbb{E}[y_k | \boldsymbol{\pi}, \mathbf{x}^\star]).$$

Since $y_k \sim \text{Ber}(\pi_k), \forall k \in [K]$, we have:

$$\text{TU}(\mathbf{x}^\star) = \sum_{k=1}^{K} \mathbb{E}_{\boldsymbol{\pi}}[\pi_k(1 - \pi_k)] + \sum_{k=1}^{K} \text{Var}_{\boldsymbol{\pi}}(\pi_k).$$

Expanding the terms:

$$\text{TU}(\mathbf{x}^\star) = \sum_{k=1}^{K} (\mathbb{E}_{\boldsymbol{\pi}}[\pi_k] - \mathbb{E}_{\boldsymbol{\pi}}[\pi_k^2]) + \sum_{k=1}^{K} (\mathbb{E}_{\boldsymbol{\pi}}[\pi_k^2] - (\mathbb{E}_{\boldsymbol{\pi}}[\pi_k])^2).$$

Simplifying:

$$\text{TU}(\mathbf{x}^\star) = \sum_{k=1}^{K} \mathbb{E}_{\boldsymbol{\pi}}[\pi_k] - (\mathbb{E}_{\boldsymbol{\pi}}[\pi_k])^2.$$

Since $\sum_{k=1}^{K} \mathbb{E}_{\boldsymbol{\pi}}[\pi_k] = 1$, the total uncertainty simplifies to:

$$\text{TU}(\mathbf{x}^\star) = 1 - \sum_{k=1}^{K} (\mathbb{E}_{\boldsymbol{\pi}}[\pi_k])^2,$$

where $\mathbb{E}_{\boldsymbol{\pi}}(\pi_k) = \frac{\alpha_k + \tau p_k}{\alpha_0 + \tau}, \forall k \in [K]$.

**Epistemic Uncertainty.** The epistemic uncertainty for a testing example $\mathbf{x}^\star$, denoted as $\text{EU}(\mathbf{x}^\star)$, can be expressed as:

$$\text{EU}(\mathbf{x}^\star) = \sum_{k=1}^{K} \text{Var}_{\boldsymbol{\pi}}(\pi_k).$$

Substituting the expression for the variance of the FD distribution, we have:

$$\text{EU}(\mathbf{x}^\star) = \sum_{k=1}^{K} \frac{\mathbb{E}_{\boldsymbol{\pi}}[\pi_k](1 - \mathbb{E}_{\boldsymbol{\pi}}[\pi_k])}{\alpha_0 + \tau + 1} + \frac{\tau^2 p_k (1 - p_k)}{(\alpha_0 + \tau)(\alpha_0 + \tau + 1)},$$

where $\alpha_0 = \sum_{k=1}^{K} \alpha_k$ and $\mathbb{E}_{\boldsymbol{\pi}}[\pi_k] = \frac{\alpha_k + \tau p_k}{\alpha_0 + \tau}, \forall k \in [K]$.

**Aleatoric Uncertainty.** The aleatoric uncertainty for a testing example $\mathbf{x}^\star$, denoted as $\mathrm{AU}(\mathbf{x}^\star)$, is obtained by subtracting the epistemic uncertainty from the total uncertainty:

$$\mathrm{AU}(\mathbf{x}^\star) = \mathrm{TU}(\mathbf{x}^\star) - \mathrm{EU}(\mathbf{x}^\star).$$

# B Algorithm

We present the training algorithm and the prediction and UQ procedure for $\mathcal{F}$-EDL in Algorithm 1 and Algorithm 2, respectively. These algorithms are straightforward to implement, requiring no sensitive hyperparameter tuning or reliance on additional techniques.

---

**Algorithm 1** $\mathcal{F}$-EDL Training

---

**Input:** Training data $\mathcal{D}_{\mathrm{tr}} = \{(\mathbf{x}_i, y_i)\}_{i=1}^N$, initial model parameters $\{\boldsymbol{\theta}, \boldsymbol{\phi}_1, \boldsymbol{\phi}_2, \boldsymbol{\phi}_3\}$, maximum epoch $T_{\max}$, learning rate $\eta$
**for** $i = 1$ **to** $T_{\max}$ **do**
 Update the parameters using the Adam optimizer and $\mathcal{F}$-EDL objective function:

$$\left\{\boldsymbol{\theta} = \{W_{\boldsymbol{\theta}}^{(l)}, b_{\boldsymbol{\theta}}^{(l)}\}_{l=1}^{L_1}, \ \boldsymbol{\phi}_1 = \{W_{\boldsymbol{\phi}_1}^{(l)}, b_{\boldsymbol{\phi}_1}^{(l)}\}_{l=1}^{L_2}, \boldsymbol{\phi}_2, \boldsymbol{\phi}_3\right\} \leftarrow \mathrm{Adam}(\nabla\mathcal{L}^{\mathcal{F}\text{-EDL}}, \eta).$$

 Apply spectral normalization to the parameters $\boldsymbol{\theta}$ and $\boldsymbol{\phi}_1$:

$$\{W_{\boldsymbol{\theta}}^{(l)}\}_{l=1}^{L_1}, \{W_{\boldsymbol{\phi}_1}^{(l)}\}_{l=1}^{L_2} \leftarrow \mathrm{SpectNorm}(\{W_{\boldsymbol{\theta}}^{(l)}\}_{l=1}^{L_1}, \{W_{\boldsymbol{\phi}_1}^{(l)}\}_{l=1}^{L_2}).$$

**end for**
**Output:** Trained model parameters $\{\hat{\boldsymbol{\theta}}, \hat{\boldsymbol{\phi}}_1, \hat{\boldsymbol{\phi}}_2, \hat{\boldsymbol{\phi}}_3\}$

---

---

**Algorithm 2** $\mathcal{F}$-EDL Prediction and Uncertainty Quantification

---

**Input:** Testing example $\mathbf{x}^\star$ and trained model parameters $\{\hat{\boldsymbol{\theta}}, \hat{\boldsymbol{\phi}}_1, \hat{\boldsymbol{\phi}}_2, \hat{\boldsymbol{\phi}}_3\}$
**Step 1: Compute FD Parameters:**

$$\boldsymbol{\alpha}(\mathbf{x}^\star) = \exp\left(g_{\hat{\boldsymbol{\phi}}_1}(f_{\hat{\boldsymbol{\theta}}}(\mathbf{x}^\star))\right),$$
$$\mathbf{p}(\mathbf{x}^\star) = \mathrm{softmax}\left(g_{\hat{\boldsymbol{\phi}}_2}(f_{\hat{\boldsymbol{\theta}}}(\mathbf{x}^\star))\right), \quad \tau(\mathbf{x}^\star) = \mathrm{softplus}\left(g_{\hat{\boldsymbol{\phi}}_3}(f_{\hat{\boldsymbol{\theta}}}(\mathbf{x}^\star))\right).$$

**Step 2: Prediction:**

$$\hat{y}(\mathbf{x}^\star) = \underset{k \in [K]}{\mathrm{argmax}} \ \mathbb{E}_{\boldsymbol{\pi}}[\pi_k], \text{ where } \mathbb{E}_{\boldsymbol{\pi}}[\pi_k] = \frac{\alpha_k(\mathbf{x}^\star) + \tau(\mathbf{x}^\star)p_k(\mathbf{x}^\star)}{\alpha_0(\mathbf{x}^\star) + \tau(\mathbf{x}^\star)}, \forall k \in [K].$$

**Step 3: Uncertainty Quantification:**

$$\text{Total Uncertainty:} \quad \mathrm{TU}(\mathbf{x}^\star) = 1 - \sum_{k=1}^K (\mathbb{E}_{\boldsymbol{\pi}}[\pi_k])^2,$$

$$\text{Epistemic Uncertainty:} \quad \mathrm{EU}(\mathbf{x}^\star) = \sum_{k=1}^K \left\{ \frac{\mathbb{E}_{\boldsymbol{\pi}}[\pi_k]\big(1 - \mathbb{E}_{\boldsymbol{\pi}}[\pi_k]\big)}{\alpha_0(\mathbf{x}^\star) + \tau(\mathbf{x}^\star) + 1} \right.$$
$$\left. + \frac{\tau^2(\mathbf{x}^\star)p_k(\mathbf{x}^\star)\big(1 - p_k(\mathbf{x}^\star)\big)}{\big(\alpha_0(\mathbf{x}^\star) + \tau(\mathbf{x}^\star)\big)\big(\alpha_0(\mathbf{x}^\star) + \tau(\mathbf{x}^\star) + 1\big)} \right\},$$

$$\text{Aleatoric Uncertainty:} \quad \mathrm{AU}(\mathbf{x}^\star) = \mathrm{TU}(\mathbf{x}^\star) - \mathrm{EU}(\mathbf{x}^\star).$$

**Output:** Predicted label $\hat{y}(\mathbf{x}^\star)$ and uncertainty estimates $\{\mathrm{TU}(\mathbf{x}^\star), \mathrm{EU}(\mathbf{x}^\star), \mathrm{AU}(\mathbf{x}^\star)\}$

---

# C  Additional Explanations about Theorems

In this section, we expand on the theoretical foundations of $\mathcal{F}$-EDL (Section 4). First, we present detailed proofs for Lemma 4.1 through Theorem 4.5.[5] Second, we provide further insights into the Bayesian perspective underlying $\mathcal{F}$-EDL and elaborate on Theorem 4.2. Finally, we offer an extended explanation of the subjective logic (SL) interpretation associated with Proposition 4.6.

## C.1  Proofs for Theorems

We prove Lemma 4.1, Theorem 4.2, Theorem 4.3, Theorem 4.4, and Theorem 4.5 sequentially.

**Proof of Lemma 4.1**  The posterior distribution of $\boldsymbol{\pi}$ given $\{y_i\}_{i=1}^N$ can be expressed as:

$$p(\boldsymbol{\pi}|\{y_i\}_{i=1}^N) \propto p_{\mathrm{FD}}(\boldsymbol{\pi}; \boldsymbol{\alpha}, \mathbf{p}, \tau) \times p(\{y_i\}_{i=1}^N|\boldsymbol{\pi}).$$

Expanding each term, we have:

$$p(\boldsymbol{\pi}|\{y_i\}_{i=1}^N) \propto \prod_{k=1}^K \pi_k^{\alpha_k - 1} \sum_{k=1}^K p_k \frac{\Gamma(\alpha_k)}{\Gamma(\alpha_k + \tau)} \pi_k^\tau \times \prod_{i=1}^N \prod_{k=1}^K \pi_k^{\mathbf{1}_{\{y_i=k\}}}.$$

Simplifying the terms yields:

$$p(\boldsymbol{\pi}|\{y_i\}_{i=1}^N) \propto \prod_{k=1}^K \pi_k^{\alpha_k + \sum_{i=1}^N \mathbf{1}_{\{y_i=k\}} - 1} \sum_{k=1}^K p_k \frac{\Gamma(\alpha_k)}{\Gamma(\alpha_k + \tau)} \pi_k^\tau.$$

Recognizing the resulting form, we see that the posterior distribution of $\boldsymbol{\pi}$ retains the form of an FD distribution, i.e.,

$$p(\boldsymbol{\pi}|\{y_i\}_{i=1}^N) = p_{\mathrm{FD}}(\boldsymbol{\pi}|\boldsymbol{\alpha}', \mathbf{p}, \tau),$$

where $\boldsymbol{\alpha}' = (\alpha_1', \dots, \alpha_K')$ and each $\alpha_k'$ is given by:

$$\alpha_k' = \alpha_k + \sum_{i=1}^N \mathbf{1}_{\{y_i=k\}}, \forall k \in [K].$$

Thus, the FD distribution is conjugate to the categorical likelihood.

**Proof of Theorem 4.2.**  Assume that the prior distribution for a given input $\mathbf{x}$ is specified as:

$$\boldsymbol{\pi}(\mathbf{x}) \sim \mathrm{FD}\left(\boldsymbol{\alpha}_{\mathrm{prior}}(\mathbf{x}), \mathbf{p}_{\mathrm{prior}}(\mathbf{x}), \tau_{\mathrm{prior}}(\mathbf{x})\right),$$

where the parameters are defined as follows:

$$\boldsymbol{\alpha}_{\mathrm{prior}}(\mathbf{x}) = \mathbf{0}, \quad \mathbf{p}_{\mathrm{prior}}(\mathbf{x}) = \mathrm{softmax}(g_{\boldsymbol{\phi}_2}(f_{\boldsymbol{\theta}}(\mathbf{x}))), \quad \tau_{\mathrm{prior}}(\mathbf{x}) = \mathrm{softplus}(g_{\boldsymbol{\phi}_3}(f_{\boldsymbol{\theta}}(\mathbf{x}))).$$

Here, $\boldsymbol{\alpha}_{\mathrm{prior}}(\mathbf{x}) = \mathbf{0}$ represents an improper prior as the prior parameter $\boldsymbol{\alpha}$ is zero. The prior parameters for $\mathbf{p}$ and $\tau$ are learned for each input $\mathbf{x}$ using a neural network.

The likelihood is estimated using a neural network as:

$$\sum_{i=1}^N \mathbf{1}_{\{y_i=k\}} = \left(\exp(g_{\boldsymbol{\phi}_1}(f_{\boldsymbol{\theta}}(\mathbf{x})))\right)_k, \forall k \in [K].$$

Using Lemma 4.1, the posterior distribution of $\boldsymbol{\pi}$ for a given input $\mathbf{x}$ is obtained as:

$$\boldsymbol{\pi}(\mathbf{x})|\mathcal{D} \sim \mathrm{FD}\left(\exp(g_{\boldsymbol{\phi}_1}(f_{\boldsymbol{\theta}}(\mathbf{x}))), \mathrm{softmax}(g_{\boldsymbol{\phi}_2}(f_{\boldsymbol{\theta}}(\mathbf{x}))), \mathrm{softplus}(g_{\boldsymbol{\phi}_3}(f_{\boldsymbol{\theta}}(\mathbf{x})))\right).$$

This posterior distribution corresponds to the predicted FD distribution of $\boldsymbol{\pi}$ for a given input $\mathbf{x}$, as described in Section 3.1. Consequently, we conclude that the class probability distribution for a given input $\mathbf{x}$, derived from $\mathcal{F}$-EDL, is equivalent to the posterior FD distribution of $\boldsymbol{\pi}$, obtained with an improper prior with input-dependent parameters learned from the data.

---

[5]We omit the proof for Proposition 4.6, as it provides a conceptual interpretation rather than a formal mathematical statement. A detailed discussion is included in Appendix C.3.

**Proof of Theorem 4.3** The class probability distribution for a testing example $\mathbf{x}^\star$, derived from $\mathcal{F}$-EDL, is expressed as:

$$p_{\mathrm{FD}}(\boldsymbol{\pi}; \boldsymbol{\alpha}, \mathbf{p}, \tau) = \frac{\Gamma(\alpha_0 + \tau)}{\prod_{k=1}^{K} \Gamma(\alpha_k)} \prod_{k=1}^{K} \pi_k^{\alpha_k - 1} \sum_{k=1}^{K} p_k \frac{\Gamma(\alpha_k)}{\Gamma(\alpha_k + \tau)} \pi_k^\tau,$$

where $\alpha_0 = \sum_{k=1}^{K} \alpha_k$. By substituting $\tau = 1$ and $p_k = \alpha_k/\alpha_0, \forall k \in [K]$, the expression simplifies as follows:

$$p_{\mathrm{FD}}(\boldsymbol{\pi}; \boldsymbol{\alpha}) = \frac{\Gamma(\alpha_0 + 1)}{\prod_{k=1}^{K} \Gamma(\alpha_k)} \prod_{k=1}^{K} \pi_k^{\alpha_k - 1} \sum_{k=1}^{K} (\frac{\alpha_k}{\alpha_0}) \frac{\Gamma(\alpha_k)}{\Gamma(\alpha_k + 1)} \pi_k.$$

Since $\Gamma(\alpha_k + 1) = \alpha_k \Gamma(\alpha_k)$, this reduces to:

$$p_{\mathrm{FD}}(\boldsymbol{\pi}; \boldsymbol{\alpha}) = \frac{\Gamma(\alpha_0 + 1)}{\prod_{k=1}^{K} \Gamma(\alpha_k)} \prod_{k=1}^{K} \pi_k^{\alpha_k - 1} \sum_{k=1}^{K} (\frac{\pi_k}{\alpha_0}).$$

Because $\sum_{k=1}^{K} \pi_k = 1$, the expression becomes:

$$p_{\mathrm{FD}}(\boldsymbol{\pi}; \boldsymbol{\alpha}) = \frac{\Gamma(\alpha_0 + 1)}{\prod_{k=1}^{K} \Gamma(\alpha_k)} \prod_{k=1}^{K} \pi_k^{\alpha_k - 1} \times \frac{1}{\alpha_0}.$$

Rewriting using $\Gamma(\alpha_0 + 1) = \alpha_0 \Gamma(\alpha_0)$:

$$p_{\mathrm{FD}}(\boldsymbol{\pi}; \boldsymbol{\alpha}) = \frac{\Gamma(\alpha_0)}{\prod_{k=1}^{K} \Gamma(\alpha_k)} \prod_{k=1}^{K} \pi_k^{\alpha_k - 1}.$$

This is precisely the probability density function of the Dirichlet distribution, i.e.,

$$p_{\mathrm{FD}}(\boldsymbol{\pi}; \boldsymbol{\alpha}) = p_{\mathrm{Dir}}(\boldsymbol{\pi}; \boldsymbol{\alpha}).$$

Therefore, we conclude that the class probability distribution for $\mathbf{x}^\star$, derived from $\mathcal{F}$-EDL, is equivalent to the class probability distribution of EDL when $\tau = 1$ and $p_k = \alpha_k/\alpha_0, \forall k \in [K]$.

**Proof of Theorem 4.4** For this proof, we rely on the definition and notation of the FG basis and FD distribution, as detailed in Section 2.2.

To establish that the class probability distribution for a testing example $\mathbf{x}^\star$, derived from $\mathcal{F}$-EDL, is represented as a mixture of Dirichlet distributions and exhibits multimodality, we derive the probability density function of the FD distribution, starting from its FG basis representation.

Since $\mathbf{Z} \sim \mathrm{Mu}(1, \mathbf{p})$, the random variable $Z_k$ follows the distribution:

$$Z_k = \begin{cases} 1 & \text{with probability } p_k, \\ 0 & \text{with probability } 1 - p_k. \end{cases}$$

The conditional distribution of $Y_k$ given $Z_k$ is:

$$Y_k | Z_k = \begin{cases} W_k + U & \text{if } Z_k = 1, \\ W_k & \text{if } Z_k = 0, \end{cases}$$

where $W_k \sim \mathrm{Gamma}(\alpha_k), \forall k \in [K]$, and $U \sim \mathrm{Gamma}(\tau)$ are independent Gamma random variables sharing the same scale parameter. Consequently, $W_k + U$ is also Gamma-distributed: $W_k + U \sim \mathrm{Gamma}(\alpha_k + \tau)$.

Let $\boldsymbol{\pi} = (\pi_1, \dots, \pi_K)$ denote the class probability vector, where

$$\pi_k = \frac{Y_k}{\sum_{k=1}^{K} Y_k}, \forall k \in [K].$$

By the definition of the FD distribution, it follows that:

$$\boldsymbol{\pi} \sim \mathrm{FD}(\boldsymbol{\alpha}, \mathbf{p}, \tau),$$

where $\boldsymbol{\alpha} = (\alpha_1, \ldots, \alpha_K)$, $\mathbf{p} = (p_1, \ldots, p_K)$.

Using the conditional distribution of $Y_k$, the relationship between $\boldsymbol{\pi}$ and $Y_k, \forall k \in [K]$, and the property that normalizing a set of independent Gamma random variables yields a Dirichlet distribution, the conditional distribution of $\boldsymbol{\pi}|\mathbf{Z}$ is given by:

$$\boldsymbol{\pi}|\mathbf{Z} = \begin{cases} \mathrm{Dir}(\alpha_1, \ldots, \alpha_k + \tau, \ldots, \alpha_K) & \text{if } Z_k = 1 \text{ for some } k, \\ \mathrm{Dir}(\alpha_1, \ldots, \alpha_K) & \text{if } Z_k = 0 \text{ for all } k. \end{cases}$$

Since $\mathbf{Z} \sim \mathrm{Mu}(1, \mathbf{p})$, the case $Z_k = 0$ for all $k$ is impossible. Consequently, the distribution of $\boldsymbol{\pi}$ is derived as:

$$p(\boldsymbol{\pi}) = \sum_{k=1}^{K} P(Z_k = 1 \text{ for } k\text{th class}) \times \mathrm{Dir}(\alpha_1, \ldots, \alpha_k + \tau, \ldots, \alpha_K)$$

$$= \sum_{k=1}^{K} p_k \, \mathrm{Dir}(\boldsymbol{\alpha} + \tau \mathbf{e}_k),$$

where $\mathbf{e}_k$ represents the $k$-th standard basis vector in $\mathbb{R}^K$.

Thus, the class probability distribution for a testing example $\mathbf{x}^\star$, derived from $\mathcal{F}$-EDL, is expressed as a mixture of Dirichlet distributions. The number of distinct modes in this distribution is determined by the cardinality of $\mathbf{p}$, i.e., $\|\mathbf{p}\|_0$.

**Proof of Theorem 4.5** The predictive class probability for a testing example $\mathbf{x}^\star$ belonging to each class $k$, $\forall k \in [K]$, derived from $\mathcal{F}$-EDL, can be represented as follows:

$$p_{\mathcal{F}\text{-EDL}}(y = k|\mathbf{x}^\star) = \int p(y = k|\boldsymbol{\pi})p(\boldsymbol{\pi}|\boldsymbol{\alpha}, \mathbf{p}, \tau, \mathbf{x}^\star)d\boldsymbol{\pi}.$$

Using the fact that $p(y = k|\boldsymbol{\pi}) = \pi_k, \forall k \in [K]$, this integral simplifies to:

$$p_{\mathcal{F}\text{-EDL}}(y = k|\mathbf{x}^\star) = \mathbb{E}_{\boldsymbol{\pi} \sim \mathrm{FD}(\boldsymbol{\alpha}, \mathbf{p}, \tau)}[\pi_k].$$

Substituting the expression for the expectation of $\pi_k$ and expanding the terms, we obtain:

$$p_{\mathcal{F}\text{-EDL}}(y = k|\mathbf{x}^\star) = \frac{\alpha_0}{\alpha_0 + \tau} \times \frac{\alpha_k}{\alpha_0} + \frac{\tau}{\alpha_0 + \tau} \times p_k.$$

On the other hand, for the EDL framework, the predictive distribution for $\mathbf{x}^\star$ is given by:

$$p_{\mathrm{EDL}}(y = k|\mathbf{x}^\star) = \mathbb{E}_{\boldsymbol{\pi} \sim \mathrm{Dir}(\boldsymbol{\alpha})}[\pi_k] = \frac{\alpha_k}{\alpha_0}.$$

For the softmax-based model, the predictive distribution for $\mathbf{x}^\star$ is given by:

$$p_{\mathrm{SM}}(y = k|\mathbf{x}^\star) = p_k.$$

Define the input-dependent mixture weights as:

$$w_{\mathrm{EDL}}(\mathbf{x}^\star) = \frac{\alpha_0}{\alpha_0 + \tau}, \quad w_{\mathrm{SM}}(\mathbf{x}^\star) = \frac{\tau}{\alpha_0 + \tau}.$$

Using the predictive distributions of EDL and softmax models and substituting these weights, the predictive distribution of $\mathcal{F}$-EDL can be expressed as:

$$p_{\mathcal{F}\text{-EDL}}(y|\mathbf{x}^\star) = w_{\mathrm{EDL}}(\mathbf{x}^\star) \times p_{\mathrm{EDL}}(y|\mathbf{x}^\star) + w_{\mathrm{SM}}(\mathbf{x}^\star) \times p_{\mathrm{SM}}(y|\mathbf{x}^\star).$$

Thus, the predictive distribution for $\mathbf{x}^\star$, obtained by the $\mathcal{F}$-EDL framework, is decomposed into the contributions of the EDL and softmax models.

## C.2 Bayesian Interpretation of $\mathcal{F}$-EDL (Theorem 4.2)

In three stages, we provide an additional explanation about the Bayesian interpretation of $\mathcal{F}$-EDL, outlined in Theorem 4.2. First, we establish a Bayesian framework termed the *input-dependent FD-Categorical model*, extending the *input-dependent Dirichlet-Categorical model* commonly used to interpret EDL methods [24, 12]. Second, we introduce a *prior specification* problem, which has been a significant challenge in traditional EDL methods. Third, we introduce an additional theorem (Theorem C.1) that bridges the gap between the prior specification in $\mathcal{F}$-EDL and traditional EDL methods. By leveraging this theorem together with Theorem 4.2 from the main text, we clarify how $\mathcal{F}$-EDL resolves the *prior specification* problem using an improper prior and input-dependent prior parameters.

**Input-Dependent FD-Categorical Model.** Let $\mathbf{x}$ represent a given input, $\boldsymbol{\pi}(\mathbf{x})$ the corresponding class probabilities, and $\{\tilde{y}_i\}_{i=1}^N$ the set of *pseudo-observations*. Then, the *input-dependent FD-Categorical model* can be described as a Bayesian model as follows:

| | |
|---|---|
| **Prior** | $\boldsymbol{\pi}(\mathbf{x}) \sim \text{FD}(\boldsymbol{\alpha}_{\text{prior}}(\mathbf{x}), \mathbf{p}_{\text{prior}}(\mathbf{x}), \tau_{\text{prior}}(\mathbf{x})),$ |
| **Likelihood** | $\{\tilde{y}_i\}_{i=1}^N \sim \text{Cat}(\boldsymbol{\pi}(\mathbf{x})),$ |
| **Posterior** | $\boldsymbol{\pi}(\mathbf{x})\|\{\tilde{y}_i\}_{i=1}^N \sim \text{FD}(\boldsymbol{\alpha}_{\text{post}(\mathbf{x})}, \mathbf{p}_{\text{post}}(\mathbf{x}), \tau_{\text{post}}(\mathbf{x})),$ |

where $\boldsymbol{\alpha}_{\text{prior}}(\mathbf{x})$, $\mathbf{p}_{\text{prior}}(\mathbf{x})$, and $\tau_{\text{prior}}(\mathbf{x})$ are the prior parameters, and $\boldsymbol{\alpha}_{\text{post}(\mathbf{x})}$, $\mathbf{p}_{\text{post}}(\mathbf{x})$, and $\tau_{\text{post}}(\mathbf{x})$ are the posterior parameters.

Leveraging the results in Lemma 4.1, the relationship between the prior and posterior parameters can be expressed as follows:

$$\alpha_{\text{post}}^{(k)}(\mathbf{x}) = \alpha_{\text{prior}}^{(k)}(\mathbf{x}) + \sum_{i=1}^N \mathbf{1}_{\{\tilde{y}_i=k\}}, \forall k \in [K],$$

$$\mathbf{p}_{\text{post}}(\mathbf{x}) = \mathbf{p}_{\text{prior}}(\mathbf{x}), \quad \tau_{\text{post}}(\mathbf{x}) = \tau_{\text{prior}}(\mathbf{x}).$$

This formulation reveals the intrinsic connection between $\mathcal{F}$-EDL and the input-dependent FD-Categorical model. The fundamental concept of $\mathcal{F}$-EDL, predicting an FD distribution over class probabilities for a given input $\mathbf{x}$, is inherently equivalent to predicting the posterior distribution over class probabilities, given the pre-specified prior distributions. In particular, the mechanism of $\mathcal{F}$-EDL is equivalent to predicting *pseudo-observations* $\{\tilde{y}_i^{(j)}\}_{j=1}^N$ for each data point $\mathbf{x}$ using a neural network. Specifically, $\mathcal{F}$-EDL predicts the counts of the pseudo-observations for each class, i.e., *pseudo-counts*, utilizing a neural network $f_{\boldsymbol{\theta}}$ and $g_{\boldsymbol{\phi}_1}$ as follows:

$$\sum_{i=1}^N \mathbf{1}_{\{\tilde{y}_i=k\}} \approx \exp(g_{\boldsymbol{\phi}_1}(f_{\boldsymbol{\theta}}(\mathbf{x}))_k).$$

Using this, the posterior parameter, $\boldsymbol{\alpha}_{\text{post}}$, is expressed as:

$$\boldsymbol{\alpha}_{\text{post}}(\mathbf{x}) = \boldsymbol{\alpha}_{\text{prior}}(\mathbf{x}) + \exp(g_{\boldsymbol{\phi}_1}(f_{\boldsymbol{\theta}}(\mathbf{x}))).$$

**Prior Specification Problem of EDL Methods.** Standard EDL methods face the *prior specification* problem, as highlighted in recent works, including R-EDL [11] and DAEDL [12]. This issue arises because EDL methods employ a uniform prior, $\boldsymbol{\pi}_{\text{prior}}(\mathbf{x}) \sim \text{Dir}(\mathbf{1})$, when predicting the posterior distribution under the *input-dependent Dirichlet-Categorical model* framework [24, 12]. Notably, the *input-dependent Dirichlet-Categorical model* is analogous to the *input-dependent FD-Categorical model* discussed earlier, differing only in its use of the Dirichlet distribution in place of the FD distribution. To address the *prior specification* problem under this framework, R-EDL [11] treats the prior parameter as an adjustable hyperparameter, adopting a prior of the form $\boldsymbol{\pi}_{\text{prior}}(\mathbf{x}) \sim \text{Dir}(\lambda\mathbf{1})$, where $\lambda$ is a tunable hyperparameter. DAEDL [12] eliminates the prior parameter, effectively utilizing an improper prior $\boldsymbol{\pi}_{\text{prior}}(\mathbf{x}) \sim \text{Dir}(\mathbf{0})$. For a more detailed discussion of the prior specification problem in EDL methods and their proposed solutions, we refer the reader to the respective papers [11, 12].

Table 6: Comparison of prior specification scheme across EDL [9], $\mathcal{I}$-EDL [10], R-EDL [11], DAEDL [12], and $\mathcal{F}$-EDL. Here, $\lambda \in \mathbb{R}$ denotes a tunable hyperparameter, $\mathbf{1} \in \mathbb{R}^K$ is a vector of ones, $\mathbf{0} \in \mathbb{R}^K$ is a vector of zeros, and $\mathbf{p}_{\text{prior}}(\mathbf{x}) \in \mathbb{R}^K$ and $\tau_{\text{prior}}(\mathbf{x}) \in \mathbb{R}$ denote the prior parameters dynamically learned for a given input $\mathbf{x}$.

| Method | EDL, $\mathcal{I}$-EDL | R-EDL | DAEDL | $\mathcal{F}$-EDL (Ours) |
|---|---|---|---|---|
| Prior | $\boldsymbol{\pi}(\mathbf{x}) \sim \text{Dir}(\mathbf{1})$ | $\boldsymbol{\pi}(\mathbf{x}) \sim \text{Dir}(\lambda\mathbf{1})$ | $\boldsymbol{\pi}(\mathbf{x}) \sim \text{Dir}(\mathbf{0})$ | $\boldsymbol{\pi}(\mathbf{x}) \sim \text{FD}(\mathbf{0}, \mathbf{p}_{\text{prior}}(\mathbf{x}), \tau_{\text{prior}}(\mathbf{x}))$ |

While these methods partially address the *prior specification* problem, both approaches have notable limitations. First, R-EDL requires manual tuning of the additional hyperparameter $\lambda$, which can be both challenging and time-consuming, particularly when applied to new tasks or datasets. Since the

choice of $\lambda$ has a non-negligible impact on UQ performance, precise tuning is critical for achieving optimal performance. Second, while DAEDL effectively mitigates the prior specification issue through its parameterization scheme, its success depends heavily on accurate density estimation to represent uncertainty. This dependence can become problematic in complex ID scenarios, where obtaining high-quality density estimates is particularly challenging. Thus, there remains a need for an EDL model that addresses the *prior specification* problem without introducing additional computational overhead or relying on precise density estimation. Such a model would enhance the robustness and generalizability of EDL methods across complex and unforeseen scenarios.

Table 6 provides a comprehensive comparison of the prior specification scheme used by representative EDL methods and $\mathcal{F}$-EDL.

**Resolving the Prior Specification Problem with $\mathcal{F}$-EDL.** $\mathcal{F}$-EDL presents a novel approach to the *prior specification* problem by utilizing an improper prior with learned parameters. Before exploring this further, we introduce an additional theorem that bridges the gap between $\mathcal{F}$-EDL and EDL.

**Theorem C.1.** *(Variant of Theorem 4.2) The predictive distribution for a given input* $\mathbf{x}$*, derived from $\mathcal{F}$-EDL, is equivalent to the predictive distribution derived from EDL, utilizing the prior* $\boldsymbol{\pi}(\mathbf{x}) \sim \mathrm{Dir}(\boldsymbol{\lambda}_{\mathrm{prior}}(\mathbf{x}))$*, where* $\boldsymbol{\lambda}_{\mathrm{prior}}(\mathbf{x})$ *are the input-dependent prior parameters learned from the data.*

**Proof of Theorem C.1** The predictive distribution for a given input $\mathbf{x}$, derived from $\mathcal{F}$-EDL, is expressed as:

$$p_{\mathcal{F}\text{-EDL}}(y = k|\mathbf{x}) = \frac{\alpha_k + \tau p_k}{\sum_{k=1}^{K} \alpha_k + \tau}, \forall k \in [K].$$

Let us define $\boldsymbol{\alpha}' = (\alpha'_1, \ldots, \alpha'_K)$, where

$$\alpha'_k = \alpha_k + \tau p_k, \forall k \in [K].$$

Using this definition, the predictive distribution for $\mathcal{F}$-EDL can be rewritten as:

$$p_{\mathcal{F}\text{-EDL}}(y = k|\mathbf{x}) = \frac{\alpha'_k}{\sum_{k=1}^{K} \alpha'_k}.$$

This expression is equivalent to the predictive distribution derived from the standard EDL model, where the concentration parameters $\boldsymbol{\alpha}'$ are parameterized as follows:

$$\boldsymbol{\alpha}' = \boldsymbol{\lambda}_{\mathrm{prior}}(\mathbf{x}) + \mathbf{e}(\mathbf{x}),$$

with the components defined as:

$$\boldsymbol{\lambda}_{\mathrm{prior}}(\mathbf{x}) = \mathrm{softplus}(g_{\boldsymbol{\phi_3}}(f_{\boldsymbol{\theta}}(\mathbf{x}))) \times \mathrm{softmax}(g_{\boldsymbol{\phi_2}}(f_{\boldsymbol{\theta}}(\mathbf{x}))), \quad \mathbf{e}(\mathbf{x}) = \exp(g_{\boldsymbol{\phi_1}}(f_{\boldsymbol{\theta}}(\mathbf{x}))).$$

Within the *input-dependent Dirichlet-Categorical model* framework, EDL with this parameterization can be interpreted as predicting the posterior Dirichlet distribution for $\boldsymbol{\pi}$, where the likelihood (evidence vector) $\mathbf{e}(\mathbf{x})$ is defined as described earlier, and the prior distribution is specified as:

$$\boldsymbol{\pi}(\mathbf{x}) \sim \mathrm{Dir}(\boldsymbol{\lambda}_{\mathrm{prior}}(\mathbf{x})),$$

where $\boldsymbol{\lambda}_{\mathrm{prior}}(\mathbf{x})$ is the input-dependent prior parameters learned from the data.

In essence, Theorem 4.2 demonstrates that $\mathcal{F}$-EDL can be interpreted as predicting the posterior FD distribution for a given input, under the *input-dependent FD-Categorical model* framework. Furthermore, Theorem C.1 establishes that the predictive distribution of $\mathcal{F}$-EDL is equivalent to that of EDL, with an input-dependent prior, i.e., $\boldsymbol{\pi}(\mathbf{x}) \sim \mathrm{Dir}(\boldsymbol{\lambda}_{\mathrm{prior}}(\mathbf{x}))$.

From these theorems, we can infer that the prior specification scheme of $\mathcal{F}$-EDL demonstrates distinct advantages in two different aspects compared to the conventional approaches. First, $\mathcal{F}$-EDL utilizes an improper prior by setting $\boldsymbol{\alpha}_{\mathrm{prior}}(\mathbf{x}) = \mathbf{0}$. This addresses the limitation of uniform priors, which

Table 7: Subjective logic interpretation of EDL [9], R-EDL [11], and $\mathcal{F}$-EDL. For EDL and R-EDL, a single SL triplet $(\mathbf{b}, u, \mathbf{a})$ represents a single opinion, where $\mathbf{b} \in \mathbb{R}^K$, $u \in \mathbb{R}$, and $\mathbf{a} \in \Delta^{K-1}$ represent belief mass, uncertainty, and base rate, respectively. In contrast, $\mathcal{F}$-EDL yields $K$ distinct SL triplets, each encoding a class-specific opinion.

| Property | EDL | R-EDL | $\mathcal{F}$-EDL (Ours) |
|---|---|---|---|
| **Distribution** | Dirichlet | Dirichlet | Flexible Dirichlet |
| **SL Triplet** $(\mathbf{b}, u, \mathbf{a})$ | $\left( \frac{\boldsymbol{\alpha}-1}{\|\boldsymbol{\alpha}\|_1}, \frac{K}{\|\boldsymbol{\alpha}\|_1}, \frac{1}{K} \right)$ | $\left( \frac{\boldsymbol{\alpha}-\lambda\mathbf{1}}{\|\boldsymbol{\alpha}\|_1}, \frac{\lambda K}{\|\boldsymbol{\alpha}\|_1}, \frac{1}{K} \right)$ | $\left( \frac{\boldsymbol{\alpha}}{\|\boldsymbol{\alpha}\|_1+\tau}, \frac{\tau}{\|\boldsymbol{\alpha}\|_1+\tau}, \mathbf{e}_j \right), \forall j \in [K]$ |
| **Hypothesis assignment** | – | – | $\mathbf{p}$ over $K$ class-specific opinions |
| **Projected Probability P** | $\frac{\boldsymbol{\alpha}}{\|\boldsymbol{\alpha}\|_1}$ | Same | $\frac{\boldsymbol{\alpha}+\tau\mathbf{p}}{\|\boldsymbol{\alpha}\|_1+\tau}$ |
| **Multimodality** | No | No | Yes (Dirichlet mixture) |
| **Epistemic source** | magnitude of $\boldsymbol{\alpha}$ | Same | $\mathbf{p}$ (inter-opinion) + $\boldsymbol{\alpha}, \tau$ (intra-opinion) |

often degrade classification performance due to the difficulty in balancing the prior parameters with the *pseudo-likelihood* [12]. By adopting this approach, $\mathcal{F}$-EDL avoids the challenges associated with using fixed uniform priors. Second, $\mathcal{F}$-EDL employs learnable prior parameters for $\mathbf{p}$ and $\tau$. Specifically, the optimal prior parameters for each data point $\mathbf{x}$ are estimated using neural networks. Compared to R-EDL, which requires manual tuning of prior parameters, $\mathcal{F}$-EDL offers a significant advantage by adaptively determining these parameters, enabling both optimal predictive performance and robust UQ. Additionally, compared to DAEDL, $\mathcal{F}$-EDL reduces the dependence on density estimation to achieve optimal UQ, enabling effective generalization to the challenging scenarios where obtaining high-quality density estimates is infeasible.

### C.3 Subjective Logic Interpretation of $\mathcal{F}$-EDL (Proposition 4.6)

In this section, we elaborate on Proposition 4.6 to clarify the subjective logic (SL) perspective of $\mathcal{F}$-EDL. We begin by revisiting the motivation of $\mathcal{F}$-EDL through the SL lens. Next, we formalize its multimodal predictive structure as a mixture of class-specific Dirichlet-distributed opinions. We then introduce a *generalized subjective logic* interpretation, highlighting how it extends traditional EDL formulations. Finally, we explain how the proposed variance-based uncertainty measures naturally align with this interpretation, enabling faithful UQ.

**Motivation: Limitations of Single-Opinion SL in EDL**   Most EDL methods [9, 10, 11, 12] can be interpreted through the lens of SL [15], where each prediction corresponds to a single subjective opinion—a Dirichlet distribution encoding belief and uncertainty over class probabilities. However, this one-opinion-per-input framework restricts the model's ability to express the ambiguity between multiple plausible hypotheses. Such limitations are especially evident for inputs with overlapping features (e.g., ambiguous digits such as "1" vs "7" in MNIST)—which commonly arise in high-risk, real-world ML applications.

**$\mathcal{F}$-EDL as a Mixture of Opinions.**   To address the limitation of representing only a single opinion, $\mathcal{F}$-EDL models uncertainty as a mixture of class-specific subjective opinions. Each component corresponds to a class-specific hypothesis, represented by a Dirichlet distribution biased toward that class, and weighted by a hypothesis selection probability. For a test input $\mathbf{x}^\star$, the resulting predictive distribution is:

$$p(\boldsymbol{\pi}|\mathbf{x}^\star) = \sum_{k=1}^{K} p_k \, \mathrm{Dir}(\boldsymbol{\alpha} + \tau \mathbf{e}_k),$$

where $\boldsymbol{\alpha}$ is the shared base concentration vector, $\tau$ controls the strength of class-specific bias in each opinion, $\mathbf{p} = (p_1, \ldots, p_K)$ denotes the mixture weights, and $\mathbf{e}_k$ is the $k$-th standard basis vector. This defines an FD distribution—a structured Dirichlet mixture capable of expressing multimodal beliefs over class probabilities.

**Generalized Subjective Logic Interpretation of $\mathcal{F}$-EDL.**   This formulation aligns $\mathcal{F}$-EDL with a generalized SL framework that accommodates multiple competing opinions, rather than collapsing

them into a single fused belief. Each Dirichlet component $\text{Dir}(\boldsymbol{\alpha} + \tau \mathbf{e}_k)$ encodes a class-specific opinion, while the hypothesis selection probabilities $\mathbf{p}$ represent epistemic uncertainty over which opinion to endorse. Unlike classical EDL models, which aggregate all evidence into a single unimodal Dirichlet, $\mathcal{F}$-EDL maintains the structure of uncertainty across hypotheses. This allows it to capture ambiguity between plausible alternatives—enabling a more expressive and faithful representation of model belief.

Table 7 summarizes this interpretation by comparing $\mathcal{F}$-EDL with representative EDL variants from the SL perspective. Unlike conventional EDL models that yield SL triplets per input, $\mathcal{F}$-EDL produces $K$ distinct SL triplets—each representing class-specific subjective opinion. These triples share the belief and uncertainty masses $(\mathbf{b}, u)$, but differ in the base rate $\mathbf{a}$, reflecting the class-dependent bias. The overall opinion is expressed as a mixture of these $K$ opinions, weighted by a categorical distribution $\mathbf{p}$ that is learned per input. This structured formulation enables principled UQ through evidence aggregation grounded in subjective logic.

In particular, it is noteworthy that, unlike EDL variants that yield a single SL triplet per input, $\mathcal{F}$-EDL produces $K$ distinct SL triplet, sharing the belief and uncertainty massess $(\mathbf{b}, u)$ but differing in the base rate $\mathbf{a}$ due to class-dependent bias. The overall subjective opinion is distributed through these $K$ subjective opinions, following the categorical distribution parameterized by $\mathbf{p}$, which is also learned per input. These allow principled uncertainty quantification through the structured evidence aggregation grounded by subjective logic.

**Variance-based Uncertainty Measures in $\mathcal{F}$-EDL**    Classical SL metrics (e.g., inverse total evidence) are not directly applicable to this mixture. Instead, we quantify epistemic uncertainty using the total variance of the FD distribution:

$$\text{EU}(\mathbf{x}^\star) = \sum_{k=1}^{K} \text{Var}(\pi_k(\mathbf{x}^\star)).$$

This formulation captures both inter-hypothesis ambiguity, reflected in the hypothesis selection probabilities (i.e., allocation probabilities) $\mathbf{p}$, and intra-hypothesis variability, driven by the evidence parameters $\boldsymbol{\alpha}$ and dispersion $\tau$. By explicitly modeling these two sources of uncertainty, $\mathcal{F}$-EDL offers more faithful and interpretable estimates of epistemic uncertainty, especially in ambiguous or complex prediction scenarios.

## D    Interpretation of FD Parameters and Justification for Its Use in UQ

This section provides additional explanation for adopting the FD distribution for UQ, demonstrating that its use represents a principled generalization rather than an arbitrary increase in flexibility. First, we clarify the semantics of the FD parameters $(\boldsymbol{\alpha}, \mathbf{p}, \tau)$. Second, we justify the suitability of the FD distribution for UQ, particularly in cases involving multiple plausible class hypotheses. Third, we describe how evidence is structurally extracted in $\mathcal{F}$-EDL.

### D.1    Semantics of FD Distribution Parameters

In the FD distribution, the parameters $\boldsymbol{\alpha}$, $\mathbf{p}$, and $\tau$ jointly characterize different aspects of uncertainty. The vector $\boldsymbol{\alpha}$ represents the total amount of evidence supporting each class, analogous to the concentration parameters in a standard Dirichlet distribution. However, unlike the Dirichlet case, the FD introduces two additional parameters—$\mathbf{p}$ and $\tau$—that disentangle how this evidence is distributed and how sharply it is expressed.

The parameter $\mathbf{p}$ specifies how belief is allocated among class-specific hypotheses. A sharp $\mathbf{p}$, where one component dominates, reflects a decisive belief in a particular class; conversely, a diffuse $\mathbf{p}$ indicates competing hypotheses and greater ambiguity in class assignment. Thus, $\mathbf{p}$ governs the *directional aspect* of uncertainty—how belief is distributed across class-specific hypotheses.

The scalar $\tau$ modulates the *intensity* or *concentration* of each hypothesis. A smaller $\tau$ yields more concentrated, confident distributions for each hypothesis, while a larger $\tau$ produces flatter, more dispersed components, effectively tempering overconfidence in uncertain or ambiguous regions. By dynamically adapting $\tau$, the FD can represent varying degrees of epistemic caution in response to data complexity.

Together, $\mathbf{p}$ and $\tau$ provide structural flexibility that extends beyond the magnitude of evidence encoded by $\boldsymbol{\alpha}$. This decomposition allows the FD to express both *how much* evidence the model has and *how that evidence is organized* across each competing hypothesis—leading to more faithful and principled UQ under complex or unforeseen data conditions.

## D.2  Justification for Using the FD Distribution for UQ

We employ the FD distribution to address a central limitation of traditional EDL—its inability to produce reliable uncertainty estimates in complex or ambiguous scenarios where multiple class hypotheses may be simultaneously plausible.

For instance, when an input resembles both a "7" and a "9", a standard EDL, which relies on a single Dirichlet distribution, produces a unimodal belief that tends to overcommit to the dominant class (see Figure 3, second and third panels). This representation fails to capture the inherent ambiguity of the input and leads to overconfident predictions. In contrast, $\mathcal{F}$-EDL employs a structured mixture of class-specific Dirichlet hypotheses, where each component encodes the belief *this image corresponds to class $k$*. All components share a common base evidence $\boldsymbol{\alpha}$, while the parameters $\mathbf{p}$ and $\tau$ respectively determine the mixture weights and concentration of each component. This formulation enables the model to represent multimodal beliefs in a principled way, naturally capturing conflicting hypotheses and reducing overconfidence in uncertainty regions of the input space.

## D.3  Evidence Extraction in $\mathcal{F}$-EDL

For each input, $\mathcal{F}$-EDL extracts and structures evidence through three conceptually interpretable stages:

- **(1) Base Evidence Extraction:** The model first estimates the base evidence $\boldsymbol{\alpha}$, analogous to standard EDL. This represents the initial strength of belief across classes and may be overconfident in ambiguous cases.

- **(2) Construction of Class-Specific Hypotheses:** Using $\boldsymbol{\alpha}$, $\mathcal{F}$-EDL constructs a Dirichlet hypothesis for each class, treating each as a plausible explanation of the input. This enables the model to represent multiple competing class-level beliefs.

- **(3) Input-Adaptive Mixing via $\mathbf{p}$ and $\tau$:** The parameters $\mathbf{p}$ and $\tau$ govern how these hypotheses are integrated. $\mathbf{p}$ assigns mixture weights to each class-specific Dirichlet— peaked for clean inputs, dispersed for ambiguous samples, and near-uniform for OOD data—reflecting how belief is distributed across hypotheses. $\tau$ modulates the sharpness of each component: it increases for ambiguous inputs to reduce spurious confidence, and remains low for clear inputs to preserve certainty.

This three-stage process aligns naturally with the mixture-based formulation of the FD distribution, emphasizing that $\mathcal{F}$-EDL's architecture arises from a principled probabilistic design rather than an ad-hoc extension of EDL.

# E  Additional Explanations about Experiments

In this section, we present a comprehensive explanation of our experiments. First, we describe the datasets used in our study. Second, we outline the implementation details.

## E.1  Datasets

In line with recent works in EDL [24, 10, 11, 12], we used CIFAR-10 [45] as the primary ID dataset for our evaluations. To assess UQ in more complex scenarios, we also employed CIFAR-100 [45] as an additional ID dataset.

For CIFAR-10, we considered the Street View House Numbers (SVHN) [47] and CIFAR-100 as OOD datasets. When CIFAR-100 was used as the ID dataset, its corresponding OOD datasets included SVHN and Tiny-ImageNet-200 (TIN). To investigate the generalizability of $\mathcal{F}$-EDL in long-tailed ID scenarios, we employed CIFAR-10-LT [46], a version of CIFAR-10 with artificially imbalanced class distributions. Specifically, we evaluated two levels of imbalance: mild ($\rho = 0.1$) and heavy ($\rho = 0.01$), using the same OOD dataset as CIFAR-10. For noisy ID scenarios, we utilized Dirty-MNIST (DMNIST) [8], a noisy variant of MNIST [60], with Fashion-MNIST (FMNIST) [61] as its OOD dataset. Additionally, for assessing distribution shift detection capabilities, we adopted CIFAR-10-C [48], which introduces controlled perturbations to CIFAR-10. Detailed descriptions of these datasets are provided below.

**CIFAR-10** [45] is one of the most widely used image classification datasets in the UQ literature. It consists of color images categorized into 10 classes, representing various animals and objects: airplane, automobile, bird, cat, deer, dog, frog, horse, ship, and truck. The dataset contains 50,000 training images and 10,000 testing images, with each image represented as a tensor of shape $3 \times 32 \times 32$ corresponding to RGB channels and spatial dimensions. The dataset is balanced, meaning each class contains an equal number of samples in both the training and testing splits. For our experiments, the training set is further divided into training and validation subsets with a ratio of $0.95 : 0.05$. CIFAR-10 served as the primary ID dataset and was also used to create CIFAR-10-LT, a long-tailed variant.

**Street View House Numbers (SVHN)** [47] is a dataset comprising cropped images of house numbers obtained from Google Street View. It contains 73,257 training images, 26,032 testing images, and 531,131 additional images, each represented as a tensor of the shape $3 \times 32 \times 32$. In our experiments, the test set of SVHN was employed as the OOD dataset when CIFAR-10, CIFAR-10-LT, or CIFAR-100 served as the ID dataset.

**CIFAR-100** [45] is a more detailed and challenging version of CIFAR-10, comprising 100 classes, each representing a distinct animal and object category. It includes 50,000 training images and 10,000 testing images, with each image represented as a tensor of the shape $3 \times 32 \times 32$. In our experiments, the training set was further divided into training and validation subsets with a ratio of $0.95 : 0.05$. CIFAR-100 was used as the ID dataset in the classical setting. Additionally, its test set was employed as an OOD dataset when either CIFAR-10 or CIFAR-10-LT served as the ID dataset.

**Tiny-ImageNet-200 (TIN)** [62] is a smaller version of the ImageNet [63] dataset, comprising 200 classes, each representing a distinct category. It contains 100,000 training images, 10,000 validation images, and 10,000 testing images, with each image represented as a tensor of the shape $3 \times 64 \times 64$. In our experiments, the test set of TIN was used as the OOD dataset when CIFAR-100 served as the ID dataset. Due to the mismatch in the image sizes, we resized TIN images to $3 \times 32 \times 32$ for compatibility.

**CIFAR-10-LT** [64] is a long-tailed version of CIFAR-10, characterized by artifically imbalanced class distributions. The imbalance severity is controlled using an imbalance factor ($\rho$), defined as the ratio of the number of samples in the head class to the tail class. In our experiments, CIFAR-10-LT datasets with two imbalance factors, $\rho = 0.1$ and $\rho = 0.01$, were used to validate the generalizability of $\mathcal{F}$-EDL in long-tailed ID scenarios.

**CIFAR-10-C** [65] is a corrupted version of the CIFAR-10 dataset, designed to evaluate the robustness of image classification models against various types of corruption and noise. It introduces 19 types of real-world corruptions, including Gaussian noise, shot noise, impulse noise, defocus blur, glass blur, motion blur, zoom blur, snow, frost, fog, brightness, contrast, elastic transform, pixelate, jpeg

Table 8: Comparison of the total parameters and the additional parameters introduced by the MLPs across different architectures. "# of Params." refers to the total number of parameters in the base model architectures (e.g., ConvNet, VGG-16, and ResNet-18). "# of Additional Params." denotes the number of parameters added by the MLPs. "Proportion" indicates the percentage increase in the total parameter count due to the addition of the MLPs.

| ID Dataset | Architecture | # of Params. | # of additional params. | Proportion (%) |
|---|---|---|---|---|
| DMNIST | ConvNet | 248,330 | 6,347 | **2.56%** |
| CIFAR-10, CIFAR-10-LT | VGG-16 | 14,857,546 | 266,507 | **1.79%** |
| CIFAR-100 | ResNet-18 | 11,046,308 | 289,367 | **2.62%** |

Table 9: Average batch inference time (batch size = 64) on the CIFAR-10 dataset using the VGG-16 backbone. The mean and standard deviation are calculated over five independent runs.

| Metric | EDL | DAEDL | $\mathcal{F}$-EDL |
|---|---|---|---|
| Inference Time (s) | $1.262 \pm 0.06$ | $2.683 \pm 0.05$ | $1.279 \pm 0.08$ |

compression, speckle noise, Gaussian blur, splatter, and saturate. Each corruption is applied at five different severity levels $\mathcal{C} \in \{1, 2, 3, 4, 5\}$, resulting in a total of 950,000 corrupted images (10,000 images $\times$ 19 corruptions $\times$ 5 severities). In our experiments, CIFAR-10-C was used for distribution shift detection tasks. Specifically, models trained on CIFAR-10 performed OOD detection using CIFAR-10-C as the OOD dataset to assess their ability to capture distribution shifts through uncertainty measures.

**Dirty-MNIST (DMNIST)** [8] is a noisy variant of the MNIST [60] dataset, containing ambiguous data points. It is generated by integrating Ambiguous-MNIST (AMNIST) [8], which includes artificially synthesized MNIST samples with varying entropy levels, into the original MNIST dataset. DMNIST contains 120,000 training images (60,000 from MNIST and 60,000 from AMNIST) and 70,000 testing images (10,000 from MNIST and 60,000 from AMNIST). In our experiments, DMNIST was used to validate the generalizability of $\mathcal{F}$-EDL in noisy ID scenarios.

**Fashion-MNIST (FMNIST)** [61] is a modern replacement for the MNIST dataset, consisting of grayscale images from 10 classes that represent various fashion items, including clothing, footwear, and accessories. The dataset contains 60,000 training images and 10,000 testing images, each represented as a tensor of shape $1 \times 28 \times 28$, similar to MNIST. In our experiments, FMNIST served as the OOD dataset when DMNIST was used as the ID dataset.

### E.2 Implementation Details

To ensure a fair comparison, we adopted VGG-16 [49] as the model architecture when CIFAR-10 served as the ID dataset, consistent with recent studies [24, 10, 11, 12]. The same architecture was utilized when CIFAR-10-LT was employed as the ID dataset. For CIFAR-100, ResNet-18 [50] was employed as the model architecture, while a simple convolutional neural network (ConvNet) consisting of three convolutional layers followed by three dense layers was implemented for DMNIST.

To compute the parameters $\mathbf{p}$ and $\tau$, we added two shallow MLPs into each architecture. These MLPs consisted of a single layer when DMNIST was the ID dataset and two layers for the other dataset. The additional model complexity introduced by these MLPs was minimal, as demonstrated in Table 8. Moreover, the added overhead becomes negligible for larger architectures, indicating that $\mathcal{F}$-EDL scales efficiently. For instance, with WideResNet-28-10 (36.5M parameters) on TinyImageNet-200, the extra heads introduce only 346K parameters (0.95%). This relative overhead further diminishes as the model capacity increases.

In terms of inference efficiency, $\mathcal{F}$-EDL enables fast prediction and UQ without sampling or post-hoc steps (e.g., DAEDL's density estimation). As shown in Table 9, the average batch inference time, including FD parameter prediction and uncertainty computation, was $1.279s \pm 0.08$—only 1.35% slower than EDL ($1.262s \pm 0.06$) and 52.34% faster than DAEDL ($2.683s \pm 0.05$).

Table 10: Implementation details and hyperparameter settings for our experiments. Here, $T_{\max}$, $B$, and $\eta$ denote the maximum number of epochs, batch size, and learning rate, respectively. Additionally, $L$ and $H$ represent the number of layers and hidden dimensions of the MLPs, respectively.

| ID Dataset | Architecture | Optimizer | Scheduler | $T_{\max}$ | B | $\eta$ | Step size | L | H |
|---|---|---|---|---|---|---|---|---|---|
| DMNIST | ConvNet | Adam | StepLR | 50 | 64 | $5 \times 10^{-4}$ | 20 | 1 | 64 |
| CIFAR-10 | VGG-16 | Adam | StepLR | 100 | 64 | $5 \times 10^{-4}$ | 30 | 2 | 256 |
| CIFAR-10-LT | VGG-16 | Adam | StepLR | 100 | 64 | $5 \times 10^{-4}$ | 30 | 2 | 256 |
| CIFAR-100 | ResNet-18 | Adam | StepLR | 100 | 64 | $10^{-4}$ | 30 | 2 | 256 |

Based on the validation loss, early stopping was applied to mitigate overfitting across all experiments. A fixed batch size of $B = 64$ was used in all settings. Training was conducted for up to 50 epochs on DMNIST and 100 epochs on other datasets. The Adam optimizer [66] and the StepLR scheduler were consistently employed. Hyperparameters for both our proposed method and the baselines were selected through grid search. Importantly, $\mathcal{F}$-EDL eliminates the need for hyperparameter tuning in its objective function, significantly simplifying the training process. The detailed experimental setups and hyperparameter configurations are provided in Table 10. All experiments were implemented in PyTorch. Depending on availability, we used either an RTX 4060 GPU with 8GB memory or a TITAN V GPU with 12GB memory.

Table 11: Comprehensive results for UQ-related downstream tasks in the classical setting using CIFAR-10 as the ID dataset, with additional results from PostNet [24] and DUQ [7].

| Method | Test.Acc. | Conf. | SVHN / CIFAR-100 |
|---|---|---|---|
| Dropout | 82.84 $\pm$0.1 | 97.15 $\pm$0.0 | 51.39 $\pm$0.1 / 45.57 $\pm$1.0 |
| PostNet | 84.85 $\pm$0.0 | 97.76 $\pm$0.0 | 77.71 $\pm$0.3 / 81.96 $\pm$0.8 |
| DUQ | 89.33 $\pm$0.2 | 97.89 $\pm$0.3 | 80.23 $\pm$3.4 / 84.75 $\pm$1.1 |
| EDL | 83.55 $\pm$0.6 | 97.86 $\pm$0.2 | 79.12 $\pm$3.7 / 84.18 $\pm$0.7 |
| $\mathcal{I}$-EDL | 89.20 $\pm$0.3 | 98.72 $\pm$0.1 | 82.96 $\pm$2.2 / 84.84 $\pm$0.6 |
| R-EDL | 90.09 $\pm$0.3 | 98.98 $\pm$0.1 | 85.00 $\pm$1.2 / 87.73 $\pm$0.3 |
| DAEDL | 91.11 $\pm$0.2 | 99.08 $\pm$0.0 | 85.54 $\pm$1.4 / 88.19 $\pm$0.1 |
| $\mathcal{F}$-EDL | **91.19** $\pm$0.2 | **99.10** $\pm$0.0 | **91.20** $\pm$1.3 / **88.37** $\pm$0.3 |

Table 12: AUROC scores for OOD detection using epistemic uncertainty estimates in the classical setting. Results are reported using CIFAR-10 as the ID dataset with SVHN and CIFAR-100 (C-100) as OOD datasets, and CIFAR-100 as the ID dataset with SVHN and TinyImageNet (TIN) as OOD datasets.

| Method | ID: CIFAR-10 OOD: SVHN / C-100 | ID: CIFAR-100 OOD: SVHN / TIN |
|---|---|---|
| EDL | 81.06 $\pm$4.5 / 80.63 $\pm$1.0 | 63.95 $\pm$3.4 / 65.32 $\pm$2.3 |
| $\mathcal{I}$-EDL | 86.79 $\pm$1.3 / 82.15 $\pm$0.5 | 77.85 $\pm$1.5 / 73.34 $\pm$0.3 |
| R-EDL | 87.47 $\pm$1.2 / 85.26 $\pm$0.4 | 77.06 $\pm$2.2 / 71.10 $\pm$1.0 |
| DAEDL | 89.24 $\pm$1.0 / 86.04 $\pm$0.1 | 81.07 $\pm$3.0 / 75.04 $\pm$1.3 |
| $\mathcal{F}$-EDL | **93.74** $\pm$1.5 / **86.37**$\pm$0.3 | **81.59** $\pm$1.8 / **79.24** $\pm$0.2 |

# F   Additional Results for Quantitative Study on UQ-Related Downstream Tasks

**Classical Setting.** Table 11 presents comprehensive results for UQ-related downstream tasks in the classical setting using CIFAR-10 as the ID dataset, comparing $\mathcal{F}$-EDL with additional baselines: DUQ

Table 13: AUROC scores for OOD detection using epistemic uncertainty estimates in the long-tailed setting. Results are reported using CIFAR-10-LT as the ID dataset under mild ($\rho = 0.1$) and heavy ($\rho = 0.01$) imbalance, with SVHN and CIFAR-100 (C-100) as OOD datasets.

| Method | ID: CIFAR-10-LT ($\rho = 0.1$) OOD: SVHN / C-100 | ID: CIFAR-10-LT ($\rho = 0.01$) OOD: SVHN / C-100 |
|---|---|---|
| EDL | 80.36 $\pm$1.5 / 77.03 $\pm$0.6 | 58.25 $\pm$4.4 / 61.05 $\pm$0.7 |
| $\mathcal{I}$-EDL | 84.65 $\pm$3.8 / 80.83$\pm$0.5 | 61.47 $\pm$7.4 / 65.12 $\pm$1.4 |
| R-EDL | 77.39 $\pm$5.3 / 72.12 $\pm$0.9 | 62.24 $\pm$2.9 / 64.03 $\pm$0.6 |
| DAEDL | 80.34 $\pm$2.9 / 73.92 $\pm$1.3 | 64.21 $\pm$4.3 / 63.49 $\pm$0.6 |
| $\mathcal{F}$-EDL | **89.22** $\pm$1.7 / **81.54** $\pm$0.6 | **71.62** $\pm$1.9 / **67.74** $\pm$1.8 |

Table 14: UQ-related downstream task results are reported using the DMNIST dataset as the ID dataset. "Brier." denotes the Brier score for calibration. "FMNIST.AUPR." and "FMNIST.AUROC." indicate AUPR and AUROC scores for OOD detection using epistemic uncertainty, with FMNIST as the OOD dataset.

| Method | Test.Acc. | Conf. | Brier. | FMNIST.AUPR. | FMNIST.AUROC. |
|---|---|---|---|---|---|
| MSP | 83.90 $\pm$0.1 | 96.01 $\pm$0.0 | 2.38 $\pm$0.0 | 98.10 $\pm$0.3 | 89.30 $\pm$1.5 |
| Dropout | 84.13 $\pm$0.1 | 96.09 $\pm$0.0 | 2.44 $\pm$0.0 | 94.68 $\pm$0.6 | 68.97 $\pm$2.8 |
| DDU | 84.05 $\pm$0.1 | 82.73 $\pm$0.1 | 2.35 $\pm$0.0 | 98.49 $\pm$0.4 | 90.16 $\pm$2.1 |
| EDL | 77.37 $\pm$5.8 | 95.19 $\pm$0.3 | 3.25 $\pm$0.5 | 92.23 $\pm$1.7 | 61.51 $\pm$6.6 |
| $\mathcal{I}$-EDL | 83.46 $\pm$0.2 | 95.58 $\pm$0.0 | 2.74 $\pm$0.0 | 94.11 $\pm$0.9 | 68.70 $\pm$4.0 |
| R-EDL | 83.41 $\pm$0.1 | 95.58 $\pm$0.1 | 2.80 $\pm$0.0 | 90.91 $\pm$5.6 | 59.45 $\pm$13.9 |
| DAEDL | 84.12 $\pm$0.1 | 95.93 $\pm$0.0 | 2.78 $\pm$0.1 | 99.44 $\pm$0.2 | 96.34 $\pm$1.4 |
| $\mathcal{F}$-EDL | **84.28** $\pm$0.1 | **96.17** $\pm$0.1 | **2.32** $\pm$0.0 | **99.76** $\pm$0.1 | **98.46** $\pm$0.7 |

Table 15: Ablation study results on $\mathcal{F}$-EDL using the CIFAR-100 dataset as the ID dataset. "Fix-$\mathbf{p}$ (U), $\tau$" fixes $\mathbf{p}$ to a uniform vector ($\mathbf{1}/K$) and $\tau$ to 1, while "Fix-$\mathbf{p}$ (N), $\tau$" fixes $\mathbf{p}$ to the normalized concentration parameters ($\boldsymbol{\alpha}/\|\boldsymbol{\alpha}\|_1$) and $\tau$ to 1. "Fix-$\mathbf{p}$ (U)" and "Fix-$\mathbf{p}$ (N)" fix $\mathbf{p}$ but allow $\tau$ to be learned. "Fix-$\tau$" fixes $\tau = 1$ while learning $\mathbf{p}$. The full "$\mathcal{F}$-EDL" model learns both $\mathbf{p}$ and $\tau$.

| Variant | Test.Acc. | Conf. | SVHN / TIN |
|---|---|---|---|
| Fix-$\mathbf{p}$ (U), $\tau$ | 63.54 $\pm$0.8 | 90.95 $\pm$0.4 | 71.49 $\pm$4.2 / 77.57 $\pm$0.4 |
| Fix-$\mathbf{p}$ (N), $\tau$ | 64.03 $\pm$0.5 | 91.27 $\pm$0.4 | 71.73 $\pm$1.7 / 77.53 $\pm$0.4 |
| Fix-$\mathbf{p}$ (U) | 63.74 $\pm$0.3 | 91.04 $\pm$0.2 | 69.06 $\pm$3.4 / 77.40 $\pm$0.6 |
| Fix-$\mathbf{p}$ (N) | 63.77 $\pm$1.3 | 91.22 $\pm$0.6 | 69.59 $\pm$2.0 / 77.60 $\pm$0.5 |
| Fix-$\tau$ | 65.68 $\pm$0.8 | 92.60 $\pm$0.5 | 73.38 $\pm$4.6 / 78.76 $\pm$0.4 |
| $\mathcal{F}$-EDL | **69.40** $\pm$0.2 | **94.00** $\pm$0.1 | **75.40** $\pm$2.3 / **80.60** $\pm$0.2 |

Table 16: Ablation study on the effect of different regularization terms in $\mathcal{F}$-EDL using CIFAR-10 as the ID dataset. "No Reg." denotes training without regularization, "KL-Div." indicates KL-based regularization applied to the Dirichlet parameters, "CE." refers to cross-entropy regularization applied on $\mathbf{p}$, and "Brier" represents the Brier-based regularization adopted in $\mathcal{F}$-EDL.

| Regularization | Test.Acc. | Conf. | SVHN / C-100 |
|---|---|---|---|
| No Reg. | 91.13 $\pm$0.2 | 99.02 $\pm$0.4 | 89.32 $\pm$0.1 / 87.65 $\pm$1.0 |
| KL-Div. | 34.61 $\pm$27.8 | 50.69 $\pm$34.7 | 45.83 $\pm$21.3 / 58.93 $\pm$13.9 |
| CE | 91.11 $\pm$0.1 | 98.87 $\pm$0.1 | 89.83 $\pm$0.7 / 87.77 $\pm$0.2 |
| **Brier (Ours)** | **91.19** $\pm$0.2 | **99.10** $\pm$0.0 | **91.20** $\pm$1.3 / **88.37** $\pm$0.3 |

[7] and PostNet [24]. DUQ represents deterministic uncertainty methods, while PostNet exemplifies a density-based EDL approach. $\mathcal{F}$-EDL outperforms both by a significant margin, underscoring its robustness. Results on CIFAR-100 are omitted due to the high computational cost and training instability these baselines face when applied to larger label spaces. Table 12 reports AUROC scores for OOD detection in the classical setting, confirming that $\mathcal{F}$-EDL maintains strong performance across different datasets and metrics.

**Long-Tailed Setting.** Table 13 presents additional OOD detection results in the long-tailed setting using the CIFAR-10-LT dataset, evaluated with AUROC. The results show that $\mathcal{F}$-EDL consistently outperforms its competitors regardless of the evaluation metric.

**Noisy Setting.** Table 14 reports the full results for UQ-related downstream tasks in the noisy setting using the DMNIST dataset. Metrics include the Brier score [67], a standard measure of calibration, and AUROC scores for OOD detection based on epistemic uncertainty estimates. For the Brier score, lower values indicate better calibration. The results show that $\mathcal{F}$-EDL consistently outperforms all competing methods across all metrics.

**Ablation Study.** Table 15 presents the additional ablation study results using CIFAR-100 as the ID dataset. The results demonstrate that the trend observed in the ablation study using the DMNIST dataset holds consistently: fixing either $\mathbf{p}$ or $\tau$ leads to degraded performance, while the full model that jointly learns both components yields the strongest results. This confirms that the observed benefits are not dataset-specific but arise from the principled generalization enabled by the FD distribution.

In addition, Table 16 presents the ablation study on the regularization term using CIFAR-10 as the ID dataset. The results indicate that the Brier-based regularization yields the best empirical performance, owing to its improved training stability and better-calibrated allocation of class probabilities.

## G  Additional Explanations and Results for Qualitative Study

In this section, we provide additional details on the toy experiment from Figure 1 and the posterior multimodality analysis from Section 6.3, including extended motivation, minor experimental settings, and supplementary figures.

### G.1  Toy Experiment in the Figure 1

We provide additional explanations about the toy experiments described in Section 1. First, we outline the experimental setup, including the procedures followed and the uncertainty measures utilized. Second, we present the results for the aleatoric uncertainty distributions, along with corresponding figures and explanations, to complement the epistemic uncertainty distribution results shown in Section 1

**Details on Experimental Setup.**  To investigate the challenges EDL models face in generalizing to difficult scenarios and assess whether $\mathcal{F}$-EDL addresses these limitations, we conducted toy experiments to compare their uncertainty distributions. These experiments use DMNIST, a noisy variant of MNIST, as the ID dataset. The objective is to evaluate the ability of EDL and $\mathcal{F}$-EDL to generalize to noisy ID scenarios and reliably distinguish among three types of data: (i) clean ID data (MNIST), (ii) noisy ID data (AMNIST), and (iii) OOD data (FMNIST).

We trained both EDL and $\mathcal{F}$-EDL using the training set of DMNIST. After training, we computed the aleatoric and epistemic uncertainties for data points in the testing sets of DMNIST and FMNIST. For EDL, the aleatoric uncertainty ($\mathrm{AU}(\mathbf{x}^\star)$) and epistemic uncertainty ($\mathrm{EU}(\mathbf{x}^\star)$) for a testing example $\mathbf{x}^\star$ are defined as follows:

$$\mathrm{AU}(\mathbf{x}^\star) = 1 - \max_{k \in [K]} \mathbb{E}_{\boldsymbol{\pi} \sim \mathrm{Dir}(\boldsymbol{\alpha})}[\pi_k], \;\; \mathrm{EU}(\mathbf{x}^\star) = \frac{K}{\alpha_0},$$

where $\alpha_0 = \sum_{k=1}^{K} \alpha_k$ and $\mathbb{E}_{\boldsymbol{\pi} \sim \mathrm{Dir}(\boldsymbol{\alpha})}[\pi_k] = \alpha_k / \alpha_0$.

For $\mathcal{F}$-EDL, we utilized the aleatoric and epistemic uncertainty measures outlined in Section 3.3. To enhance the clarity of the figures, we normalized the uncertainty estimates. For epistemic uncertainty, a logarithmic transformation was applied. Both uncertainty estimates were then scaled to the range

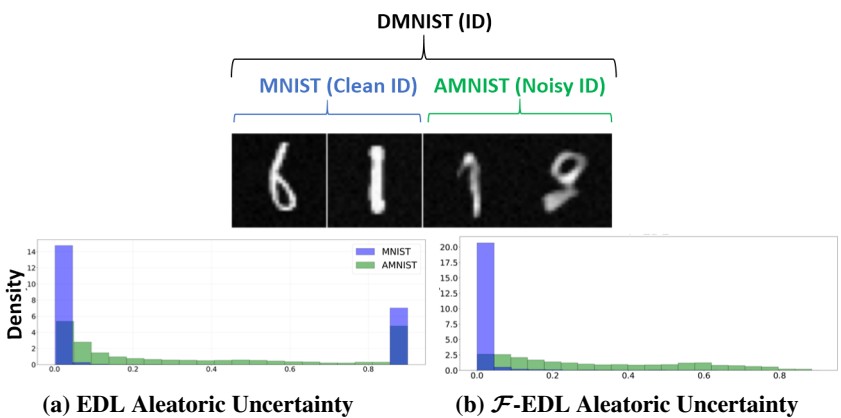

**(a) EDL Aleatoric Uncertainty**   **(b) $\mathcal{F}$-EDL Aleatoric Uncertainty**

Figure 5: Aleatoric uncertainty distributions with DMNIST as the ID dataset. The top row presents sample images from MNIST and AMNIST. Panels (a) and (b) display histograms depicting the aleatoric uncertainty distributions obtained by the EDL and $\mathcal{F}$-EDL (proposed) models, respectively, across these datasets.

$[0, 1]$ as follows:

$$\mathrm{AU}(\mathbf{x}^\star) \leftarrow \frac{\mathrm{AU}(\mathbf{x}^\star) - \mathrm{AU}_{\min}}{\mathrm{AU}_{\max} - \mathrm{AU}_{\min}}, \quad \mathrm{EU}(\mathbf{x}^\star) \leftarrow \frac{\log(\mathrm{EU}(\mathbf{x}^\star)) - \log(\mathrm{EU}_{\min})}{\log(\mathrm{EU}_{\max}) - \log(\mathrm{EU}_{\min})}.$$

Here, $\mathrm{AU}_{\max}$ and $\mathrm{AU}_{\min}$ denote the maximum and minimum values of aleatoric uncertainty, while $\mathrm{EU}_{\max}$ and $\mathrm{EU}_{\min}$ represent the maximum and minimum values of epistemic uncertainty. These values were computed using the combined test sets of DMNIST and FMNIST to ensure consistent scaling across both datasets.

**Results for Aleatoric Uncertainty Distributions.**   The middle panel of Figure 5 illustrates the aleatoric uncertainty distributions generated by the EDL model. Ideally, a UQ model should assign higher aleatoric uncertainty to noisy samples, effectively distinguishing them from clean samples.

The results highlight the limitations of EDL in handling aleatoric UQ in noisy ID scenarios. First, a substantial portion of the MNIST test set exhibits high aleatoric uncertainty, nearing the maximum value shown in the figure. This suggests that EDL struggled to effectively train on the noisy ID dataset consisting of ambiguous data points. Second, there is a significant overlap between the uncertainty distributions of MNIST and AMNIST, suggesting that the model incorrectly assigned low aleatoric uncertainty to AMNIST test points, despite their artificial construction to exhibit higher aleatoric uncertainty than MNIST.

In contrast, the bottom panel of Figure 5, which presents the aleatoric uncertainty distributions produced by $\mathcal{F}$-EDL, demonstrates notable improvements over EDL in generalizing to noisy ID scenarios and delivering robust UQ for ID data. Specifically, AMNIST demonstrates significantly higher aleatoric uncertainty compared to MNIST, with minimal overlap between their uncertainty distributions. The small overlap observed between distributions is expected, as certain AMNIST data points with relatively low aleatoric uncertainty may naturally align with the uncertainty levels of MNIST.

## G.2   Multimodal Uncertainty Representations for Ambiguous Inputs.

**Motivation.**   We investigate whether the theoretical flexibility of $\mathcal{F}$-EDL to represent multimodal class probability distribution translates to semantically meaningful and interpretable uncertainty in practice. While Theorem 4.4 shows that $\mathcal{F}$-EDL can express predictive distributions as a mixture of Dirichlet components, it is crucial to assess whether this capacity emerges empirically—especially for perceptually ambiguous inputs. In particular, we aim to verify whether the model reflects structured hesitation between multiple plausible class hypotheses instead of defaulting to a single, potentially overconfident mode (Proposition 4.6).

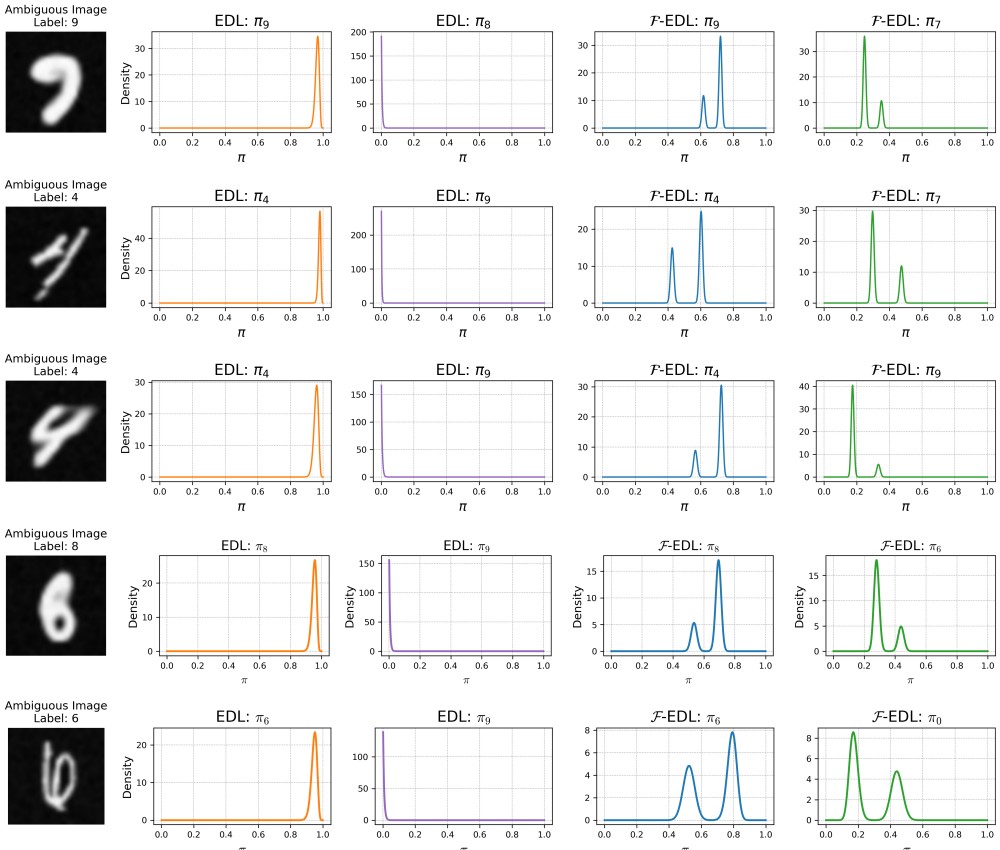

Figure 6: Additional examples supplementing Figure 3, showing posterior class probability distributions for ambiguous DMNIST inputs. Each subfigure includes: the input image, followed by the marginal distributions over the two most probable classes as predicted by EDL (second and third panels) and by $\mathcal{F}$-EDL (fourth and fifth panels).

**Experimental Setup.** We analyze test examples from the DMNIST dataset that exhibit visual ambiguity. For each input $\mathbf{x}^\star$, we identify the top two predicted classes based on the expected class probabilities:

$$y_1^\star = \underset{k \in [K]}{\operatorname{argmax}} \, \mathbb{E}_{\boldsymbol{\pi}}[\pi_k(\mathbf{x}^\star)], \quad y_2^\star = \underset{k \in [K], k \neq y_1^\star}{\operatorname{argmax}} \, \mathbb{E}_{\boldsymbol{\pi}}[\pi_k(\mathbf{x}^\star)].$$

We visualize the marginal distributions $p(\pi_{y_1^\star}|\mathbf{x}^\star)$ and $p(\pi_{y_2^\star}|\mathbf{x}^\star)$. Under the FD distribution, each marginal is a mixture of Beta distributions, parameterized by shared base evidence $\boldsymbol{\alpha}$, allocation probabilities $\mathbf{p}$, and a dispersion parameter $\tau$:

$$p(\pi_k|\mathbf{x}^\star) = p_k \operatorname{Beta}(\alpha_k + \tau, \alpha_0 - \alpha_k) + (1 - p_k) \operatorname{Beta}(\alpha_k, \alpha_0 - \alpha_k + \tau), k \in \{y_1^\star, y_2^\star\}.$$

These marginals exhibit multimodal behavior when the allocation probabilities $\mathbf{p}$ are dispersed, i.e., when $p_k$ is not close to 1 and—leading to a nontrivial mixture of hypotheses. Moreover, the degree of mode separation is captured by:

$$\Delta_{\text{mode}} = \left| \frac{\tau}{\alpha_0 + \tau - 2} \right|,$$

which increases with larger $\tau$, as higher dispersion sharpens and shifts each Beta component toward opposite ends, amplifying their separation.

**Additional Results.** In Figure 6, we present five additional examples of posterior multimodality on visually ambiguous inputs, extending the results from Figure 3. Each examples display the class

probability distributions predicted by EDL and $\mathcal{F}$-EDL for images with overlapping or unclear digit structures. For both models, we extract the marginal distributions of the two most probable classes, denoted as $\pi_{y_1^\star}$ and $\pi_{y_2^\star}$. EDL produces unimodal Beta distributions that average over competing hypotheses, often concentrating mass on a single class. This unimodality results in overconfident predictions—especially problematic in ambiguous cases. In contrast, $\mathcal{F}$-EDL produces multimodal marginals with distinct peaks, each reflecting a plausible interpretation—such as "7" and "9" for the hybrid digit. By retaining *structured* multimodality, $\mathcal{F}$-EDL preserves input ambiguity rather than collapsing it into a single smoothed belief.

This ability to model structured multimodal beliefs offers key advantages. First, it improves interpretability: multimodal outputs reveal not only that the model is uncertain but also *why*, by highlighting plausible alternatives. This property is particularly valuable in safety-critical domains like medical diagnosis, where knowing *which* alternatives the model considers plausible can inform downstream decisions. Second, this flexibility contributes to stronger UQ performance. Structured multimodality helps avoid overconfident beliefs on ambiguous inputs, thereby reducing misclassification and enhancing robustness in OOD detection [68]. Intuitively, EDL is biased toward collapsing uncertainty into a single mode. While this may be sufficient for downstream tasks when the dominant mode aligns with the true class, it risks failure when it favors an incorrect one—leading to confident misclassification or false OOD rejection of ambiguous ID inputs. In contrast, $\mathcal{F}$-EDL preserves a mixture of multiple plausible hypotheses, preventing premature convergence and allowing the model to gradually learn and better represent the underlying uncertainty. As demonstrated in our quantitative evaluations (Section 6.2), $\mathcal{F}$-EDL consistently outperforms existing methods, particularly under ambiguity (e.g., AMNIST) or data imbalance (e.g., CIFAR-10-LT tail classes).

