# OpenReview forum: "Uncertainty Estimation by Flexible Evidential Deep Learning"
_NeurIPS.cc/2025/Conference — NeurIPS 2025 poster_

### Official Review · Reviewer_jVmC · 2025-06-18

**Clarity:** 2
**Significance:** 2
**Originality:** 3
**Rating:** 5
**Confidence:** 5

**Summary:**

The paper introduces flexible evidential deep learning (${\cal F}$-EDL) as an extension to evidential deep learning (EDL) by introducing a generalization to the Dirichlet distribution that the coin the flexible Dirichlet distribution.  The paper provides theory to prove that the flexible Dirichlet distribution is still a conjugate prior for a multinomial likelihood and the distribution can be viewed as mixture of Dirichlet so that it can capture multimodal distribution with better control of adjusting aleatoric and epistemic uncertainty.  The experimental section provides a comprehensive demonstration that ${\cal F}$-EDL is more effective for misclassification detection and out-of-distribution detection that other state-of-the-art uncertainty aware methods including EDL and more recent variants thereof.

**Questions:**

The key questions include:

1) Why would minimization of the cost function not require that  $\alpha_0,\tau \to \infty$?

2) What is the semantics behind the evidence that leads to a likelihood function adjusting the values for $p_k$ and $\tau$?

3) What justifies how evidence is extracted from training data relative to the image under test to justify the flexible Dirichlet distribution?

**Ethical Concerns:**

["NO or VERY MINOR ethics concerns only"]

**Final Justification:**

The authors have answered my most serious concerns about grounding the loss function through regularization.  I am not convinced that the authors have the final story of why the flexible Dirichlet is able to better capture epistemic uncertainty than the less expressive Dirichlet distribution. Nevertheless, in light of the strong empirical results, the EDL research community can try to answer this question moving forward.

Overall, I have moved my recommendation up to an accept decision.

**Limitations:**

The conclusions do point out the limitations of the current approach.  These appear comprehensive.

As basic research for uncertainty quantification, I do not see an potential negative societal impact.  Rather, the techniques discussed in the paper help to identify when not to trust the machine learning model.

**Paper Formatting Concerns:**

None!

**Quality:**

3

**Strengths And Weaknesses:**

The flexible Dirichlet distribution is an interesting generalization of the Dirichlet distribution, and the experimental results are comprehensive and demonstrate consistent improvement via the ${\cal F}$-EDL. I especially liked the idea to separate out aleatoric and epistemic uncertainty for misclassification and OOD detection, respectively.

The main weakness of the paper is that it is not clear how ${\cal F}$-EDL achieves its performance gains via the cost function that is minimizes.  The overall cost function, which is not in the main paper but in the supplementary appendix A.1 (line 848).  Unlike most variants of EDL, it does not include a KL-like regularization term to temper the values of the Dirichlet strength $\alpha_0$ or the other strength parameter $\tau$.   The problem is that the middle two terms in the cost function go to zero as $\tau,\alpha_0 \to \infty$. The  two squared error terms can be minimized where $\alpha_k \propto p_k$.  In other words, minimization of the cost function as stated in the paper would indicate that the two strength parameters should go to infinity.  What is actually tempering these two values?  Why would minimization of the two squared error terms require different values for $p_k$ and $\frac{\alpha_k}{\alpha_0}$?

Figure 3 is really nice to illustrate that the flexible Dirichlet distribution is multimodal.  Perhaps there is an argument that this multimodal nature is necessary in terms of reducing the squared error terms in the cost function. However, I really do not understand it at this moment.

Another issue is that while the flexible EDL is a conjugate prior of the multinomial likelihood, observations from the multinomial distribution only affect the values of $\alpha_k$.  What is the semantics behind the evidence that leads to the value of $p_k$ and $\tau$? Perhaps this could give insights into what notions of uncertainty are captured by the flexible distribution and how it helps the performance of ${\cal F}$-EDL.

The paper hypothesizes that the limitations of EDL is the used of the Dirichlet distribution to express the second order uncertainty. This reviewer believes that the limitation with EDL is more about how it extracts an approximation of evidence for the Dirichlet distribution.  The original version of EDL conflates aleatoric and epistemic uncertainty.  More recent EDL variants try to address this issue by altering the cost function.  Somehow, EDL should capture how the training data serves as evidence for various class labels for the input (usually an image) under test. As more training data is used, the evidence should increase accordingly and the Dirichlet strength should decrease.  Using the original cost function for EDL, it has been demonstrated that this is not the case. I am not convinced that a generalized version of the Dirichlet distribution fixes this problem.

The weaknesses discussed above are driving my current recommendation.  A few additional minor points are provided below.

One particular method to address the problem to disentangle aleatoric and epistemic uncertainty is the generative EDL [R1].  I am very curious how it would fair in the paper's experiments.  Nevertheless, I do not view this generative EDL as a required baseline for publication.

Generalized Dirichlet distributions make sense if there is likelihood function that provides for an argument for its use.  This gets to the semantics questions from above. For instance some papers have looked as heretical set classification to address blurry images where the model might not be able to label the image as a "dog" but can label it as an "animal". In that case, hyper-Dirichlet and group Dirichlet distributions naturally emerge, which are generalizations of the Dirichlet distribution. What justifies how evidence is extracted from training data relative to the image under test to justify the flexible Dirichlet distribution?

[R1] Sensoy, Murat, Lance Kaplan, Federico Cerutti, and Maryam Saleki. "Uncertainty-aware deep classifiers using generative models." In Proceedings of the AAAI conference on artificial intelligence, vol. 34, no. 04, pp. 5620-5627. 2020.

---

> ### Author Rebuttal · Authors · 2025-07-30
>
> Thank you for your careful review of our paper and for the insightful and constructive comments. Please find our detailed answers to your comment below.
>
> **1. (Why would minimization of the cost function not require that $\alpha_{0}$ and $\tau \rightarrow \infty$?)**
>
> We clarify this issue step by step, addressing both the reviewer’s primary question and underlying concerns.
>
> **KL regularization is not suitable for $\mathcal{F}$-EDL.**
> We omit KL regularization as it is ill-defined for the FD distribution, intractable to compute, and empirically ineffective. See our response to **Reviewer a4Mo, Comment 3** for details.
>
>
> **Loss minimization does *not* always imply that $\alpha_{0}$ and $\tau \rightarrow \infty$.**
> For a given batch $B$, the core loss consists of squared error terms:
> $$
> \sum_{\mathbf{x} \in B} \left(\sum_{k=1}^{K} \left( y_k(\mathbf{x}) - \frac{\alpha_k(\mathbf{x}) + \tau(\mathbf{x}) p_k(\mathbf{x})}{\alpha_0(\mathbf{x}) + \tau(\mathbf{x})} \right)^2 + \sum_{k=1}^{K} (y_{k}(\mathbf{x}) - p_{k}(\mathbf{x}))^{2} \right).
> $$
>
> This is minimized when:
> $$
> \frac{\alpha_{k}(\mathbf{x})}{\alpha_{0}(\mathbf{x})} = p_{k}(\mathbf{x}) = y_{k}(\mathbf{x}) \quad \text{for all} \  \mathbf{x} \in B, k \in [K].
> $$
>
> In this ideal case—when predictions are confident and correct for all samples in the batch— increasing $\alpha_{0}$ and $\tau$ toward infinity further minimizes the loss.
> However, if the model is uncertain or incorrect—i.e.,
> $$
> \text{argmax}_k \left( \frac{\alpha_k(\mathbf{x})}{\alpha_0(\mathbf{x})} \right) \neq y
> \quad \text{or} \quad
> \text{argmax}_k \ p_k(\mathbf{x}) \neq y,
> $$
> where $y$ denotes the true class index for $\mathbf{x}$, then increasing $\alpha_0$ and $\tau$ amplifies overconfidence in the wrong direction, thereby *increasing* the loss. This naturally discourages the unbounded growth of these parameters unless the predictions are exactly correct.
>
> In practice, achieving perfect predictions across a batch is rare. Analogously, in logistic regression, weights diverge under perfect separability, but for non-separable data, they remain bounded. Similarly, in our setting, imperfect predictions across batches act as an implicit regularizer—preventing $\alpha_{0}$ and $\tau$ from growing unbounded, even without explicit penalties.
>
> **Spectral normalization implicitly regularizes $\alpha$.**
> We apply spectral normalization to both the feature extractor and the head predicting $\boldsymbol{\alpha}$, which enforces Lipschitz continuity and constrains logit magnitudes, which in turn bounds $\alpha_{0}$. As a result, the model is inherently prevented from producing arbitrarily large $\alpha_{0}$, even without explicit regularization.
>
> **Multimodality emerges to minimize loss on ambiguous inputs.**
> For instance, consider an input resembling both “7” and “9” but labeled as “7”. Due to inductive biases or limited evidence, the model may initially favor incorrect classes (e.g., $\alpha_{9} > \alpha_{7}$). To reduce the squared error, it must adjust $\alpha_{7}$, $p_{7}$, and $\tau$, gradually shifting belief toward the correct class.
> This naturally induces *multimodality* over plausible classes, helping the model avoid overconfident local optima—highlighting that multimodality emerges as a natural consequence of minimizing loss under ambiguity.
>
> **Edge cases can be handled via clamping.**
> In early-stage or synthetic scenarios where perfect classification is attainable, $\alpha_{0}$ and $\tau$ may grow rapidly. While we did not observe this in practice, *clamping* can be applied as a safeguard if needed.
>
> **2. (What is the semantics behind $\mathbf{p}$ and $\tau$ in terms of evidence, given that only $\boldsymbol{\alpha}$ is directly affected by multinomial observations).**
>
> **Clarification on Bayesian Interpretation.**
> We clarify that in $\mathcal{F}$-EDL, the parameters $(\boldsymbol{\alpha}, \mathbf{p}, \tau)$ are learned directly for the input $\mathbf{x}$ via a neural network, not inferred through Bayesian updates. The Bayesian formulation (Theorem 4.2) serves as a *conceptual lens*, not a generative model. The *observations*  used in the conjugate analogy are *pseudo-observations*, not actual samples (see Appendix C.2 for details).
>
> **Semantics of $\mathbf{p}$ and $\tau$.** While $\boldsymbol{\alpha}$ captures the *amount* of evidence for each class, $\mathbf{p}$ and $\tau$ govern its *alllocation* and *concentration*:
> * $\mathbf{p}$ controls how belief is distributed across class-specific hypotheses. A sharp $\mathbf{p}$ indicates a strong preference for a single class; diffuse $\mathbf{p}$ reflects ambiguity or uncertainty across multiple classes.
> * $\tau$ controls the concentration within each hypothesis. A low $\tau$ yields peaked, confident components; a high $\tau$ flattens them, mitigating overconfidence—especially in ambiguous cases.
>
> Together, $\mathbf{p}$ and $\tau$ capture structural uncertainty beyond $\boldsymbol{\alpha}$, enabling better ambiguity modeling and generalization.
>
> **3. (The limitation of EDL may lie more in its evidence approximation mechanism than in the Dirichlet itself. How does the FD distribution resolve this limitation?)**
>
> We address this with three key points:
>
> **$\mathcal{F}$-EDL empirically exhibits faithful epistemic uncertainty.**
> As training data increases, $\mathcal{F}$-EDL consistently shows a decrease in uncertainty—aligning with the desirable behavior noted by the reviewer. This trend is clearly demonstrated in Fig. 4 (Section 6.3).
> Importantly, the *variance of the FD distribution*—epistemic uncertainty measure in $\mathcal{F}$-EDL—serves as a natural generalization of *inverse Dirichlet strength* used in standard EDL. Thus, the observed decrease in FD variance with more training data directly supports that $\mathcal{F}$-EDL faithfully captures epistemic uncertainty—suggesting that its evidence approximation mechanism is effective in practice.
>
> **Structural generalization is a principled path beyond EDL.**
> The key innovation in $\mathcal{F}$-EDL lies in its structurally richer representation, which we view as a principled and underexplored direction for improving EDL.
> Standard EDL struggles with multimodal uncertainty, as a *single* Dirichlet distribution—optimized using one-hot labels—cannot represent *multiple* plausible hypotheses. While refining *evidence approximation* under such constraints may offer marginal gains, it does not address the core representational bottleneck. In contrast, the FD distribution enables a *structured mixture of class-specific beliefs*, allowing the model to represent *multimodal* uncertainty naturally—without overcommitting to a single explanation.
>
> **$\mathcal{F}$-EDL bypasses key theoretical critiques of EDL’s learning mechanism.**
> Similar to the reviewer’s observation, recent works have argued that the core limitation of EDL stems from its inherent learning mechanism. However, $\mathcal{F}$-EDL falls outside the scope and assumptions of these critiques due to its flexible UQ structure. For a detailed discussion, please refer to our response to **“Reviewer a4Mo, Comment 4”**
>
> **Final Takeaway.**
> While disentangling whether EDL’s limitation stems from *evidence approximation* or *representational structure* is challenging, we argue that structural expressiveness is the deeper bottleneck. Our results suggest that richer representational families like FD offer a promising path forward.
>
> **4. (Regarding generative EDL)**
>
> Generative EDL is a promising direction, but it is not directly comparable to our approach, as it relies on auxiliary generative modules that introduce substantial computational and training overhead. Still, it could complement $\mathcal{F}$-EDL—e.g., via applying generative outlier exposure to $\mathcal{F}$-EDL for improved UQ.
>
> **5. (What justifies the use of FD distribution, and how does it relate to how evidence is extracted from the training data for a given input?)**
>
> We adopt the FD distribution to overcome a key limitation of EDL: its inability to provide robust uncertainty estimates in *complex or unforeseen scenarios*—particularly when multiple class hypotheses are plausible.
>
> For instance, when an input resembles both a “9” and a “7”, a standard EDL—relying on a *single* Dirichlet—produces a *unimodal* belief that may overcommit to the dominant class (see Fig. 3, panels 2-3), failing to capture the input’s intrinsic ambiguity.
>
> In contrast, $\mathcal{F}$-EDL models a *structured mixture* of class-specific Dirichlet hypotheses, where each component encodes the belief *“this image corresponds to class $k$”*. All components share a base evidence  $\boldsymbol{\alpha}$, while $\mathbf{p}$ assigns mixture weights and $\tau$ modulates their concentration. This structured design enables natural modeling of *multimodal* beliefs and mitigates overconfident predictions in ambiguous settings.
>
> **Evidence Extraction in $\mathcal{F}$-EDL:**
>
> For each input, $\mathcal{F}$-EDL extracts and structures evidence through three conceptually interpretable stages.
>
> * **1. Extracting Base evidence**: The model estimates the base evidence $\boldsymbol{\alpha}$ as in standard EDL. This may initially reflect an erroneously overconfident belief in ambiguous cases.
>
> * **2. Hypotheses Construction**: A Dirichlet hypothesis is constructed for each class using $\boldsymbol{\alpha}$, treating each as a plausible explanation of the input.
>
> * **3. Input-Adaptive Mixing via $\mathbf{p}$ and $\tau$**:
>
>     - $\mathbf{p}$ assigns mixture weights to each class-specific hypothesis:
>     peaked for clean inputs, spread for ambiguous ones, and near-uniform for OOD.
>
>    - $\tau$ adjusts component sharpness: increasing in ambiguous inputs to reduce erroneous overconfidence, and remaining low for clean inputs to preserve confident beliefs.
>
> The resulting model aligns with the *mixture formulation of the FD distribution*, highlighting it as a principled modeling choice rather than an ad-hoc generalization.

---

> > ### Comment · Reviewer_jVmC · 2025-08-05
> >
> > I would like to thank the authors for replying to my comments.  They have mostly addressed my major concerns
> >
> > Regarding the loss minimization, my point is that for the terms:
> >
> > $$ \sum_{\mathbf{x} \in B} \left(\sum_{k=1}^{K} \left( y_k(\mathbf{x}) - \frac{\alpha_k(\mathbf{x}) + \tau(\mathbf{x}) p_k(\mathbf{x})}{\alpha_0(\mathbf{x}) + \tau(\mathbf{x})} \right)^2 + \sum_{k=1}^{K} (y_{k}(\mathbf{x}) - p_{k}(\mathbf{x}))^{2} \right),$$
> >
> > the value is invariant to the value of the $\tau(\mathbf{x})$ and $\alpha_0(\mathbf{x})$.  Without any other regularization term, the second term (variance) for ${\cal L}^{\mbox{MSE}}$ in line 845 would drive $\tau$ and $\alpha_0$ to infinity. I do see that the spectral normalization is key to drive down these values.  This point should be emphasized after introducing the loss function in line 134. At this point, it would be good to also mention that the KL divergence is usually used to regularize $\alpha_0$.
> >
> > I agree with all the theoretical claims about the flexible Dirichlet distribution. More importantly, the observations of how the parameters for the flexible Dirichlet distribution changes for ID versus ODD data is great empirical evidence for the efficacy of moving to the generalized distribution.  Nevertheless, I struggle greatly with understanding why $\tau$ becomes larger and comparable to $\alpha_0$ for OOD samples in light of how the methods is trained via the loss function. Both $p_{k}(\mathbf{x})$ and $\frac{\alpha_k(\mathbf{x}) + \tau(\mathbf{x}) p_k(\mathbf{x})}{\alpha_0(\mathbf{x}) + \tau(\mathbf{x})}$ are trying to predict the label while the latter is more expressive.  What is forcing the $\tau$ to be comparable to  $\alpha_0$ when the sample $\mathbf{x}$ is far from the training sample, i.e., OOD?
> >
> > While it is not necessary to answer my question in light of the strong empirical evidence, I think a good answer would greatly enhance the paper.

---

> ### Author Response · Authors · 2025-08-06
> **Response to Reviewer jVmC’s Follow-up Comment**
>
> Thank you for carefully reading our rebuttal and for your thoughtful follow-up. We’re pleased to hear that our previous response addressed your main concerns—particularly the theoretical justification for using FD distribution for uncertainty quantification, which is central to our work.
>
> We believe that incorporating the insights discussed in our responses to your comments (Comments 2, 3, and 5) would significantly improve the clarity of our presentation, and we will revise the manuscript accordingly.
>
>
> We address your additional comments below:
>
> **1. (Regarding the loss minimization)**
>
> Thank you for raising this insightful question. We appreciate your observation regarding the tendency of the loss terms to drive $\alpha_{0}$ and $\tau$ toward infinity, and we elaborate below to clarify.
>
>
> By definition,  $\alpha_0(\mathbf{x}) = \sum_k \alpha_k(\mathbf{x})$. Thus, minimizing the loss while driving $\alpha_0(\mathbf{x}) \rightarrow \infty$ requires that only $\alpha_y(\mathbf{x}) \rightarrow \infty$ for the true label $y$, while all others $\alpha_k(\mathbf{x})$ for $k \neq y$ remain finite.
>
> Now, consider an ambiguous or OOD input $\mathbf{x}’$ for which the model assigns a dominant value to $\alpha_{y’}(\mathbf{x}’)$ for some class $y’ \neq y$, i.e., a class other than the true label $y$. In this case, increasing $\alpha_{y’}(\mathbf{x}’) \rightarrow \infty$ leads to $\alpha_{0}(\mathbf{x}’) \rightarrow \infty$, but this does *not* minimize the squared error, since the prediction is incorrect. However, as you correctly stated, doing so *does* reduce the variance term. This creates a trade-off, and the model learn to adjust $\alpha_{0}(\mathbf{x}’)$ and $\tau(\mathbf{x}’)$ accordingly.
>
>
> In particular, arbitrarily increasing $\alpha_{0}$ is only beneficial when the model’s predictions are both highly confident and accurate across the batch. In practice, however, real-world data is rarely perfectly separable. As a result, some predictions are inevitably incorrect or uncertain, and the aggregated loss gradients naturally constrain the growth of $\alpha_{k}$, thereby indirectly bounding $\alpha_{0}$.
>
>
> That said, we agree with your concern that the model lacks *explicit* regularization on $\alpha_{0}$ and $\tau$, which could, in principle, allow uncontrolled growth. Addressing this when describing the objective would improve the clarity of our paper. In the revised version, we will include the following clarification after the equation on line 134:
>
> >Notably, we do not apply KL regularization to $\boldsymbol{\alpha}$—as done in standard EDL—because it is ill-defined for FD, intractable to compute, and empirically ineffective. While $\boldsymbol{\alpha}$ and $\tau$ are not explicitly regularized, spectral normalization enforces a Lipschitz continuity and constrains the logits, thereby implicitly bounding their magnitude and preventing uncontrolled growth.
>
>
> **2. (What is forcing the $\tau$ to be comparable to $\alpha_{0}$ when the sample $\mathbf{x}$ is far from the training sample, i.e., OOD?)**
>
>
> Thank you for the insightful question. Since our model does not impose any explicit regularization or guidance to $\tau$, its value is shaped entirely by the optimization dynamics of the FD training objective. As such, analytically characterizing how $\tau$ is trained to exhibit specific behaviors—particularly in response to unforeseen OOD inputs—is challenging. We therefore offer an intuitive explanation below.
>
>
> First, $\tau(\mathbf{x})$ often appears *relatively large* for OOD inputs—not because it is *absolutely large*, but because $\alpha_{0}(\mathbf{x})$ is typically small. This low value of $\alpha_{0}(\mathbf{x})$ reflects reduced evidence and high epistemic uncertainty, which is standard and desirable behavior in an EDL-based model when faced with an unfamiliar input.
>
>
> Second, $\tau(\mathbf{x})$ can become *relatively large* when the model’s base evidence $\boldsymbol{\alpha}(\mathbf{x})$ is spuriously overconfident—for example, highly concentrated in the incorrect class. In such cases, increasing $\tau(\mathbf{x})$ encourages a more balanced and multimodal mixture of Dirichlet components, thereby mitigating overconfidence and more accurately capturing the ambiguity of the input.
>
>
> In essence, $\tau$ is optimized as an adaptive dispersion parameter that
> * (i) becomes relatively influential when $\alpha_{0}(\mathbf{x})$ is small—as is typical for OOD inputs,
> * (ii) compensates for overconfident or misaligned beliefs by inducing a more balanced, multimodal belief over class hypotheses.
>
>
> This observation is consistent with our empirical findings: for ambiguous or OOD samples, $\tau(\mathbf{x})$ is often comparable to or even larger than $\alpha_{0}(\mathbf{x})$, which contributes to improved performance in downstream tasks related to uncertainty quantification.
>
>
> We would be happy to clarify further if any part remains unclear or if you have additional questions.

---

> > ### Comment · Reviewer_jVmC · 2025-08-06
> >
> > I want to thank the authors for their response.  All of my concerns have been addressed.
> >
> > I am still struggling with the intuition behind the the dispersion parameter, but I do appreciate the insights the authors are providing.

---

> ### Author Response · Authors · 2025-08-07
>
> Thank you very much for your thoughtful follow-up and acknowledging that your concerns have been addressed.
>
> We appreciate your continued engagement and understand that the intuition behind the dispersion parameter may still feel subtle. As we have provided a detailed explanation in our eariler responses, we will refrain from repeating it here, but we sincerely hope that our discussion was at least partially helpful for giving some intuition. We will make sure to further improve the clarity of this explanation in the final version of the paper.
>
>
> We also kindly invite you to share any additional questions or points of clarification you may have during the remainder of the *reviewer-author discussion* period. We would be more than happy to provide further elaboration if it would help inform your final evaluation.

---

### Official Review · Reviewer_jKtY · 2025-07-02

**Clarity:** 4
**Significance:** 3
**Originality:** 2
**Rating:** 5
**Confidence:** 4

**Summary:**

This paper discussed the limitations of using a Dirichlet Distribution on compositional data analysis, i.e., classification with all probabilities summing to one. By introducing a Flexible Dirichlet, the EDL framework is enhanced not only in classification accuracy but also in Uncertainty Quantification, i.e., in-domain and out-of-domain detection.
The authors provided detailed explanations and derivations in the main text and the appendix. The idea of replacing conventional Dirichlet with a Flexible Dirichlet is interesting, and the authors proposed a data-driven approach, i.e., neural head, to learn the parameters of a Flexible Dirichlet.

**Questions:**

If my understanding is correct, based on your inference algorithm, the label prediction process is like a post-adjusting process on allocation probability p using learned concentration parameter \alpha and dispersion \tau.
1. The regularisation term is intuitively trying to align with the GT one-hot encoding by the Berier score from the scoring rule, which is similar to the L2-norm. Is there any ablation compared to CE loss/KL loss?
2. The effectiveness and reliability of learned concentration parameter \alpha, as in one of your baseline I-EDL, which discussed the learning of \alpha. It would be better to pick some test instances to analyse the learned \alpha
3. The EDL has also been widely used in Multiview/Multimodal Learning. Given the existing works, which are mainly based on Subjective Logic to combine different views’ Dirichlet, have you considered how to combine multiple FDS of each modality/view？Is it still eligible to use the Dempher-Shafer combination rule, or is a new combination method needed?

I would like to kindly request similar experiments on other datasets, at least one more dataset, for example, a text-based dataset, a different vision domain or a larger one with more class labels, to better support the effectiveness of your work.

**Ethical Concerns:**

["NO or VERY MINOR ethics concerns only"]

**Final Justification:**

The authors have addressed most of my concerns. Therefore, I decided to raise my rating.

**Limitations:**

The authors have discussed the limitations at the end, together with future work.

**Quality:**

3

**Strengths And Weaknesses:**

Strengths
* The paper introduces Flexible Dirichlet (FD) as a generalisation of the Dirichlet distribution, and provides rigorous theoretical support (e.g., conjugacy, multimodality, subjective logic interpretation). These generalisations are non-trivial and well-motivated.
*   Using an expected MSE over the FD and a Brier-score-like regularizer provides an elegant and interpretable way to align predictions with the one-hot ground truth, improving calibration without requiring OOD data.
*  The proposed F-EDL reduces to standard EDL under specific settings, thus maintaining backward compatibility while enabling greater expressiveness.
* The empirical results on CIFAR-10, CIFAR-100, CIFAR-10-LT, and DMNIST demonstrate improvements across various tasks (classification, OOD detection, misclassification detection, and distribution shift). The proposed method outperforms strong EDL baselines.
* The visualisations of multimodal uncertainty and the faithful trend of decreasing epistemic uncertainty with more data lend strong qualitative evidence of F-EDL's effectiveness.
* The contribution of FD-specific parameters (p and τ) is well-analysed, confirming that the performance gains are not merely due to model capacity but due to the expressive modelling capability of FD.

Weaknesses
*  While the results on vision datasets are comprehensive, the paper lacks evaluation on non-vision domains (e.g., NLP). This raises concerns about cross-domain robustness.
*  Since α is a central component of both EDL and F-EDL, an in-depth analysis of the learned α values across different examples (e.g., low-confidence vs. high-confidence samples) is missing. Such an analysis would shed light on its reliability and distinguishability from prior EDL methods like I-EDL.
* While a subjective logic interpretation is provided, the paper doesn’t discuss the potential of combining FD parameters from multiple views/modalities, which is an active area in multiview UQ. It's unclear whether Dempster-Shafer-like fusion remains valid for FD or new fusion rules are needed.
* The paper uses a Brier score–like term instead of the CE/KL loss commonly used in EDL and other methods. However, no ablation study is included to compare the effects of using Brier vs. CE/KL losses.
* Although the overhead is said to be minor, the architectural complexity introduced by three output heads (for α, p, τ) and the FD computation could be clarified further in terms of inference time and scalability to larger models.

I believe the trusted multiview learning is part of EDL methods, so recommended the authors adding some discussions and references to those works in the Related Works section.
Some recommended references:
 1. Han, Zongbo, Changqing Zhang, Huazhu Fu, and Joey Tianyi Zhou. "Trusted multi-view classification with dynamic evidential fusion." IEEE transactions on pattern analysis and machine intelligence 45, no. 2 (2022): 2551-2566.
 2.  Xu, Cai, Jiajun Si, Ziyu Guan, Wei Zhao, Yue Wu, and Xiyue Gao. "Reliable conflictive multi-view learning." In Proceedings of the AAAI conference on artificial intelligence, vol. 38, no. 14, pp. 16129-16137. 2024.
3. Liang, Xinyan, Pinhan Fu, Yuhua Qian, Qian Guo, and Guoqing Liu. "Trusted Multi-View Classification via Evolutionary Multi-View Fusion." In The Thirteenth International Conference on Learning Representations. 2025.
4. Lu, J., Buntine, W., Qi, Y., Dipnall, J., Gabbe, B. and Du, L., 2024. Navigating Conflicting Views: Harnessing Trust for Learning. arXiv preprint arXiv:2406.00958.

---

> ### Author Rebuttal · Authors · 2025-07-30
>
> Thank you for your careful review of our paper and for the insightful and constructive comments. Please find our detailed answers to your comment below.
>
> **1. (Including the results on non-vision or large-scale datasets would enhance effectiveness.)**
>
> We agree that evaluating on non-vision or large-scale datasets further strengthens the evidence for the effectiveness of our approach. Accordingly, we conducted additional experiments on the following benchmarks:
>
> * **UCI Adult (Tabular)**: Binary classification dataset with 48,842 examples, each consisting of 14 demographic features. We used a 3-layer MLP as the backbone.
> * **AG News (Text)**: 4-class news categorization dataset, containing 120k train / 7.6k test samples. We subsampled 10k train / 2k test examples for efficiency and used the DistilBERT encoder with linear classification heads.
> * **TinyImageNet-200 (Large-Scale Vision)**: 200-class image classification dataset containing 100k training and 10k validation images of size 64 $\times$ 64 $\times$ 3. We used a modified ResNet-18 as the backbone.
>
>
> In all settings, we compared $\mathcal{F}$-EDL with softmax and EDL, and included either R-EDL or DAEDL as a recent baseline.
> For tabular/text tasks, we evaluated in-distribution (ID) UQ using test accuracy, misclassification detection AUPR and AUROC, Brier score, and expected calibration error (ECE). For vision tasks, we focused on uncertainty-based OOD  detection, measured by AUPR scores across diverse OOD datasets.
>
>
>
>
>
> #### **[Table 1. UQ-related downstream task performance on UCI Adult]**
>
> | Method          | Test.Acc. ↑       | Brier ↓          | AUROC ↑         | AUPR ↑          |
> |-----------------|------------------|------------------|------------------|------------------|
> | Softmax         | 82.41 ± 0.4     | 13.30 ± 0.2     | 84.88 ± 0.6     | 68.38 ± 0.3     |
> | EDL             | 82.40 ± 0.5     | 15.17 ± 0.3     | 84.76 ± 0.5    | 68.20 ± 1.6     |
> | R-EDL           | 82.40 ± 0.5     | 15.35 ± 0.3     | 84.77 ± 0.5     | 68.23 ± 1.5     |
> |**$\mathcal{F}$-EDL** | **82.51 ± 0.4** | **12.41 ± 0.1** | **85.83 ± 0.6** | **70.86 ± 1.5** |
>
>
>
>
>
> #### **[Table 2. UQ-related downstream task performance on AG News]**
>
> | Method           | Test.Acc. ↑ | AUPR ↑ | ECE ↓ | Brier ↓ |
> |------------------|------------|--------|--------|-----------|
> |Softmax                              |   84.80       | 80.40      |    15.15  |  30.27  |
> | EDL                                  | 88.25      | 76.15  | 16.81  | 5.41      |
> | R-EDL                              | 86.75      | 90.76  | 12.45  | 6.26      |
> | **$\mathcal{F}$-EDL** | **89.85**  | **96.99**  | **10.20**  | **4.73**  |
>
>
> #### **[Table 3. AUPR scores for OOD Detection on TinyImageNet]**
>
> | Method           | Places | SVHN  | FMNIST | Avg.    |
> |------------------|--------|-------|--------|--------|
> | Softmax          | 74.08  | 71.06 | 67.23  | 70.79  |
> | EDL              | 69.41  | 68.41 | 67.85  | 68.56  |
> | DAEDL            | 85.69  | 79.87 | 86.01  | 83.86  |
> | **$\mathcal{F}$-EDL** | **86.65** | **92.06** | **90.01** | **89.57** |
>
> Table 1-3 shows that $\mathcal{F}$-EDL consistently outperforms the baselines across all datasets and tasks, demonstrating its robustness across modalities and scalability to large-scale vision settings.
>
> We will formalize these results, include additional baselines, and include them in the *Appendix* of the revised version.
>
> **2.  (In-depth analysis of the learned $\alpha$ across different examples would enhance the distinguishability.)**
>
> We agree that analyzing the learned $\boldsymbol{\alpha}$ values—along with $\mathbf{p}$ and $\tau$—can improve the distinguishability of our model. Thus, we report the FD parameters for clean (e.g., digit “9”) and ambiguous samples (e.g., digit resembling both “4” and “9”), and compare them with those of EDL.
>
> As shown in Table 4, $\mathcal{F}$-EDL captures structured multimodality for ambiguous inputs by assigning large values of $\alpha$ and $p$ to the plausible classes (“4” and “9”), along with a large $\tau$ to reflect uncertainty. In contrast, EDL tends to exhibit bias toward a single dominant class, failing to reflect the semantic ambiguity of such inputs.
>
>
> #### **[Table 4. Parameter values for clean vs. ambiguous sample]**
> | Model              | Input               | Dirichlet Parameters $\boldsymbol{\alpha}$                            | Flexible Parameters $(\mathbf{p}, \tau)$                              |
> |--------------------|---------------------|-----------------------------------------------------------------------|------------------------------------------------------------------------|
> | EDL     | Ambiguous (“4” vs. “9”) | $\alpha_9 = 14.71,\ \alpha_4 = 1.00$    | —                                                                      |
> | EDL                | Clean (“9”)           | $\alpha_9 = 139.09,\ \alpha_4 = 1.00$                                 | —                                                                      |
> | $\mathcal{F}$-EDL  | Ambiguous (“4” vs. “9”) | $\alpha_9 = 161.05,\ \alpha_4 = 157.65$                               | $p_9 = 0.54,\ p_4 = 0.32,\ \tau = 323.27$                              |
> | $\mathcal{F}$-EDL  | Clean (“9”)           | $\alpha_9 = 10048.80,\ \alpha_4 = 374.79$                             | $p_9 = 0.98,\ p_4 = 0.01,\ \tau = 55.16$                               |
>
> We will formalize these by adding more test instances and include them in the revised version.
>
> **3. (Regarding the comments on trusted multi-view learning)**
>
> We agree that *trusted multi-view learning* is a timely and promising research direction in EDL. We will add a brief discussion in *Related Work*, citing the recommended references as follows:
> > Recently, EDL methods have been widely applied for trusted multi-view learning by fusing Dirichlet opinions across views using subjective logic and Dempster-Shafer fusion rules [1-4].
>
> **On the applicability of Dempster-Shafer fusion in $\mathcal{F}$-EDL.**
> While many multi-view EDL methods rely on the Dempster-Shafer (D-S) combination under the subjective logic (SL) framework, such strategies do not directly extend to our setting. $\mathcal{F}$-EDL uses the FD distribution, which models a structured mixture of class-specific SLs via additional parameters $\mathbf{p}$ and $\tau$, alongside $\boldsymbol{\alpha}$. This richer structure renders classical D-S rules incompatible, highlighting the need for new fusion mechanisms tailored to this setting.
>
> **Potential for leveraging FD for trusted multi-view learning.**
> Leveraging FD-based subjective opinions for multi-view UQ opens up promising avenues that go beyond traditional fusion strategies. Below, we outline three potential directions:
> * **(i) FDs $\rightarrow$ FD fusion**: Develop novel fusion rules to combine multiple view-specific FD distributions into a unified FD. This may involve extending D-S combination principles, incorporating learnable gating mechanisms, or applying attention-based modules to adaptively weigh each view based on its reliability.
> * **(ii) FDs $\rightarrow$ Dirichlet fusion**: Fuse view-specific FD distributions, learned via $\mathcal{F}$-EDL, into a unified Dirichlet using $\mathbf{p}^{(v)}$ and $\tau^{(v)}$ as reliability indicators for each view $v \in [V]$—enabling informed fusion.
> * **(iii) Dirichlets $\rightarrow$ FD fusion**: Starting from view-specific Dirichlets, construct a fused FD distribution, in which $\mathbf{p}$ and $\tau$ are learned to encode inter-view ambiguity and conflict—enabling more expressive representation of fused belief.
>
> Given $\mathcal{F}$-EDL’s strong UQ performance under challenging ID scenarios and across diverse modalities, we believe it provides a solid foundation for advancing *trusted multi-view learning*. We will briefly mention this in the *Conclusion* and include a detailed discussion in the *Appendix*.
>
> **4. (Why does the paper use a Brier-score-like term in the loss/regularization instead of CE/KL? Any ablations?)**
>
> **Regarding the loss:** We adopt the expected MSE (i.e., a Brier-style loss) over the FD distribution due to its tractability and empirical effectiveness. This choice is consistent with the original EDL (Sensoy et al., 2018) and is widely used in recent variants ($\mathcal{I}$-EDL, R-EDL, and DAEDL).
> In contrast, expected CE is intractable under FD, and KL-based losses lack a closed-form solution and a well-defined target distribution.
>
> **Regarding the regularization:** We adopted a Brier-style regularizer to promote input-dependent calibration and empirical performance. To support this choice, we conducted an additional ablation study. For further details of the rationale behind this regularization and the corresponding ablation results, please refer to our response to **Reviewer a4Mo, Comment 3**.
>
>
>
>
> **5. (Clarify the inference time and scalability to larger models.)**
>
> We clarify the architectural and computational overhead of $\mathcal{F}$-EDL below:
>
> **Scalability.** We predict $\boldsymbol{\alpha}$ using the backbone’s existing MLP head and add two lightweight MLPs for $\mathbf{p}$ and $\tau$. The added overhead is minimal for especially for larger architectures—e.g., for WideResNet-28-10 (36.5M parameters) on TinyImageNet-200, the extra heads add only 346K parameters (\~0.95\%). This relative overhead further diminishes as model capacity increases.
>
> **Inference Time.** $\mathcal{F}$-EDL supports efficient prediction and UQ without sampling or post-hoc steps (e.g., DAEDL’s density estimation). On CIFAR-10 with VGG-16, the average batch inference time (including FD parameter prediction and uncertainty computation) was 1.279s $\pm$ 0.08—only 1.35\% slower than EDL (1.262s $\pm$ 0.06) and 52.34\% faster than DAEDL (2.683s $\pm$0.05).
>
> We will add this analysis to the main text, including quantitative comparisons of the inference time and a discussion of scalability for larger models.

---

> > ### Comment · Reviewer_jKtY · 2025-08-07
> >
> > Thanks very much for the authors' clarification. My concerns have mainly been addressed. I am happy to increase my rating.

---

> > > ### Author Response · Authors · 2025-08-07
> > >
> > > **Dear Reviewer jKtY,**
> > >
> > > Thank you very much for your kind follow-up and for considering an increased rating. We're especially glad to hear that your concerns have been largely addressed. We truly appreciate your thoughtful engagement, insightful feedback, and the time you have dedicated throughout the review process. Your comments have been invaluable in helping us improve the clarity and impact of our work.
> > >
> > >
> > > **Best regards,**
> > >
> > > **Authors**

---

### Official Review · Reviewer_a4Mo · 2025-07-03

**Clarity:** 4
**Significance:** 3
**Originality:** 3
**Rating:** 4
**Confidence:** 5

**Summary:**

This study argues that the restrictive assumption of Dirichlet-distributed class probabilities undermines the robustness of EDL. To address this limitation, the authors propose Flexible Evidential Deep Learning (F-EDL), which generalizes EDL by predicting a flexible Dirichlet distribution. Theoretically, we establish several advantageous properties of F-EDL, and extensive empirical evaluations demonstrate its SOTA uncertainty quantification performance across diverse experimental settings.

**Questions:**

See weaknesses.

**Ethical Concerns:**

["NO or VERY MINOR ethics concerns only"]

**Limitations:**

Lately, Deep Evidential Learning (EDL) has been criticized for lacking a sound theoretical foundation. In particular, the work by Jürgens et al. [2024] (ICML '24, pre-print: arXiv:2402.09056) has analyzed many EDL methods and found that they all fail to represent epistemic uncertainty faithfully. In my opinion, this manuscript should address these issues and argue how the proposed changes overcome/bypass them.

**Quality:**

3

**Strengths And Weaknesses:**

1. (Strength) Relaxing the restrictive assumption of Dirichlet-distributed class probabilities presents an interesting and meaningful direction for enhancing the UQ capability of EDL. This work shares a similar motivation with Relaxed-EDL (R-EDL), as both aim to improve EDL by relaxing existing restrictive assumptions. However, the proposed method adopts a fundamentally different implementation, offering novel insights that could significantly benefit the EDL community.
2. (Strength) The paper is well-structured and clearly written, with rigorous theoretical analysis. The experimental design is comprehensive, and the results demonstrate strong performance across diverse settings. Overall, this is a high-quality contribution to the field of EDL.
3. (Question) To my knowledge, I-EDL was the first to interpret EDL as a generative process and represent it via a graphical model. The graphical model and generative process presented in Figure 2 appear to differ in form from those in I-EDL. Could the authors clarify whether there are any substantive differences between the two formulations?
4. (Weakness) Theorem 4.5 (Predictive Distribution Decomposition) is somewhat challenging to follow. I recommend that the authors provide a more intuitive and accessible explanation in the main text to improve readability.
5. (Weakness) The rationale behind the choice of regularization in Section 3.2 remains unclear. Why does F-EDL employ the proposed regularization form instead of traditional KL-divergence-based regularization or no regularization (as seen in some other EDL-related works)? A more thorough theoretical and empirical analysis of how regularization selection impacts F-EDL's performance would strengthen the paper.
6. (Weakness) Recent critiques, particularly the work by Jürgens et al. [2024] (ICML ’24, arXiv:2402.09056), argue that many EDL methods lack a solid theoretical foundation and fail to faithfully represent epistemic uncertainty. While the authors briefly discuss this issue in "Faithfulness of Epistemic Uncertainty Estimates," their analysis relies primarily on empirical observations (e.g., uncertainty trends with varying training dataset sizes) rather than addressing the theoretical concerns raised by Jürgens et al. A more rigorous theoretical response would significantly enhance the manuscript's robustness.
7. (Suggestion) I suggest that the authors include a brief discussion in the Related Work section comparing the motivation of this work with that of R-EDL. Since both approaches aim to enhance EDL by relaxing restrictive assumptions, such a discussion would help readers better contextualize the contributions of this work without diminishing its novelty.

---

> ### Author Rebuttal · Authors · 2025-07-30
>
> Thank you for your careful review of our paper and for the insightful and constructive comments. Please find our detailed answers to your comment below.
>
> **1. (Can you clarify the differences between the graphical models of $\mathcal{I}$-EDL and $\mathcal{F}$-EDL?)**
>
> At a high level, both models follow a similar generative process: (i) compute the parameters of the distribution over class probabilities, (ii) sample class probabilities, and (iii) generate the observed label.
>
> However, there are several key differences:
> * **Parameterization:**  $\mathcal{F}$-EDL models the class probability distribution using three parameters—$\boldsymbol{\alpha}$, $\mathbf{p}$, and $\tau$—whereas $\mathcal{I}$-EDL only predicts $\boldsymbol{\alpha}$.
> * **Label generation:**  $\mathcal{I}$-EDL assumes that the label $\mathbf{y}$ is sampled from a multivariate Gaussian distribution whose mean corresponds to the class probabilities and maximizes the corresponding log-likelihood. This formulation may be intended to motivate their use of Fisher information. However, it is functionally equivalent to assuming that $\mathbf{y}$ is drawn from a Categorical distribution and minimizing the expected MSE, which aligns with our approach.
> * **Conditional dependence:** In $\mathcal{I}$-EDL, the label $\mathbf{y}$ is conditionally dependent on both the predicted class probabilities $\boldsymbol{\pi}$ (or $\mathbf{p}$ in their notation) *and* the Dirichlet parameter $\boldsymbol{\alpha}$, reflecting an additional dependency related to Fisher information. In contrast, $\mathcal{F}$-EDL adheres to the standard EDL formulation where $\mathbf{y}$ depends only on $\boldsymbol{\pi}$.
>
> We also note notational differences: $\mathcal{I}$-EDL adopts plate notation, possibly for clarity, and $\mathcal{F}$-EDL denotes deterministic variables (e.g., $\mathbf{z}, \boldsymbol{\alpha}, \mathbf{p}, \tau$) with rectangular nodes, following standard conventions.
>
>
> **2. (Theorem 4.5 is challenging to follow; please provide an intuitive explanation in the main text.)**
>
> Theorem 4.5 aims to provide intuition for how $\mathcal{F}$-EDL behaves as an adaptive mixture of EDL and softmax predictions, interpolating between them based on input characteristics.
>
> For example, on clean in-distribution (ID) data, $\alpha_{0}$ is typically much larger than $\tau$, so the predictive distribution is dominated by the EDL component. Since EDL provides reliable uncertainty estimates in such regions, this results in well-calibrated predictions.
>
> For ambiguous or out-of-distribution (OOD) data, neither EDL nor softmax alone is sufficient—$\mathcal{F}$-EDL learns to optimally combine two, adjusting the balance for each input. Theorem 4.5 formally characterizes this behavior, showing that $\mathcal{F}$-EDL functions as an input-dependent mixture of the EDL and softmax models.
> We will include a clearer, more intuitive explanation in the revised version.
>
> **3. (Why does $\mathcal{F}$-EDL use Brier-based regularization instead of KL-divergence or no regularization?)**
>
> We adopt a Brier-based regularization term for both theoretical and empirical reasons: it encourages input-dependent calibration and demonstrates strong empirical performance.
>
> Theoretically,  Brier-based regularization promotes calibrated allocation probabilities $\mathbf{p}$. To ensure that $\mathbf{p}$ remains both interpretable and aligned with the true label, it should be softly encouraged toward the one-hot-target—without collapsing into overconfident assignments. The Brier score naturally supports this goal: it penalizes the squared deviation from the ground-truth label, thereby encouraging calibrated, smooth distributions. Compared to cross-entropy (CE)—which often drives sharp and overconfident outputs—Brier provides softer gradients suited for learning well-calibrated $\mathbf{p}$.
>
> Empirically, we chose Brier-based regularization for the following reasons:
>
> * **No regularization** led to unstable training and poor calibrated $\mathbf{p}$ due to lack of guidance.
> * **KL-based regularization** as used in standard EDL is ill-suited, as we do not explicitly model the Dirichlet distribution parameterized solely by $\boldsymbol{\alpha}$. Moreover, computing the KL divergence between FD distributions is analytically intractable, making direct regularization infeasible.
> * **CE-based regularization** yielded less stable optimization and inferior empirical performance. In particular, it led to poorer misclassification detection, likely due to its tendency to produce less calibrated $\mathbf{p}$.
>
>
> Among these options, Brier-based regularization consistently yields the most robust and reliable UQ performance across tasks. To support this claim, we report the ablation study results on UQ-related downstream tasks using CIFAR-10 as the ID dataset. Table 1 summarizes the results averaged over five runs, comparing the effects of different regularization strategies: none, KL, CE, and Brier.
>
>
> #### **[Table 1. Ablation study results on regularization term on CIFAR-10]**
> | Regularization   | Test.Acc. ↑       | Conf. ↑           | SVHN / C-100 ↑                     |
> |------------------|-------------------|-------------------|----------------------------------|
> | No Reg.          | 91.13 ± 0.2       | 99.02 ± 0.4       | 89.32 ± 0.1   / 87.65 ± 1.0      |
> | KL-Div           | 34.61 ± 27.8      | 50.69 ± 34.7      | 45.83 ± 21.3 / 58.93 ± 13.9      |
> | CE               | 91.11 ± 0.1       | 98.87 ± 0.10      | 89.83 ± 0.7  / 87.77 ± 0.2       |
> | **Brier (Ours)** | **91.19 ± 0.2**   | **99.10 ± 0.0**   | **91.20 ± 1.3 / 88.37 ± 0.3**    |
>
>
> These results demonstrate that Brier-based regularization yields superior empirical performance, attributable to improved training stability and more calibrated allocation probabilities.
>
>
> We will include the rationale for our choice of regularization in the main text and provide additional ablation study results in the *Appendix* of our revised version.
>
>
> **4.  (How does $\mathcal{F}$-EDL respond to recent critiques of EDL, particularly those raised by Jurgens et al. [2024]? Does the proposed method overcome these issues or bypass them?)**
>
> In essence, we argue that $\mathcal{F}$-EDL bypasses most of the core theoretical critiques while also empirically overcoming their practical implications. Below, we elaborate further on the critique by Jürgens et al. [2024], as well as more recent and intuitively framed criticism by Shen & Ryu [NeurIPS 2024, arXiv:2402.06160].
>
> **(On Jürgens et al.)**  This paper claims that an ideal second-order learner (such as EDL) should approximate a *reference second-order distribution*—obtained by bootstrapping first-order predictors trained on resampled datasets. They demonstrate that standard EDL fails to match this distribution in practice.
> However, this argument does not directly apply to $\mathcal{F}$-EDL for two reasons:
> * **(i) Ill-defined first-order distribution**: $\mathcal{F}$-EDL introduces additional parameters ($\mathbf{p}$ and $\tau$), which fundamentally change the hypothesis space. As a result, the corresponding first-order learner is no longer well-defined in the classical sense, rendering the notion of a *reference second-order distribution* assumed by Jürgens et al.—ambiguous or inapplicable.
> * **(ii) Lack of regularization on the second-order distribution**: The core theoretical result in Jürgens et al. relies on explicit regularization of the second-order distribution, typically via a KL divergence term. In contrast, $\mathcal{F}$-EDL applies regularization only to the parameters $\mathbf{p}$, not the second-order distribution itself—thereby violating a key assumption required for their analysis to apply.
>
> Moreover, the foundational assumption—that second-order learners *should* be aligned with a bootstrapped reference distribution—is a strong modeling choice, not a universally accepted principle. We view Jürgens et al. as valuable in identifying potential limitations of standard EDL formulations, but not as establishing a definitive criterion that all EDL variants must satisfy.
>
> **(On Shen & Ryu.)** This paper claims that standard EDL models are implicitly constrained to match a *fixed target meta-distribution*. As a result, they claim EDL acts more as an energy-based OOD detector than as a faithful quantifier of epistemic uncertainty—particularly in the sense that uncertainty should decrease as more data becomes available.
> While thought-provoking, their analysis assumes KL or reverse-KL divergence-based losses, which do not directly extend to alternative formulations like Brier-style (expected MSE) loss used in $\mathcal{F}$-EDL. Furthermore, extending their notion of a *fixed target meta-distribution* to our FD family is non-trivial, and a closed-form is not readily available.
>
> In summary, while these theoretical critiques offer valuable insights, $\mathcal{F}$-EDL falls outside their core assumption due to its more expressive parameterization and distinct objective. Moreover, given the difficulty of theoretically certifying faithful epistemic uncertainty, we take an empirical approach—showing that $\mathcal{F}$-EDL’s uncertainty estimates decrease as more data becomes available (Fig. 4).
>
> We will clarify how $\mathcal{F}$-EDL relates and bypasses the critiques highlighted in prior work in the revised version.
>
> **5. (Comparison with R-EDL in Related Work will help contextualize contributions.)**
>
> In the revised version, we will include the following sentence in the *Related Work* section when discussing R-EDL:
> >R-EDL and $\mathcal{F}$-EDL share the common goal of improving EDL by relaxing restrictive modeling assumptions.
> >R-EDL addresses this by introducing a tunable prior parameter and removing variance terms from the loss function.
> >In contrast, $\mathcal{F}$-EDL generalizes the Dirichlet assumption itself by adopting a more flexible distribution, thereby enhancing the expressiveness and generalizability of UQ.

---

> ### Author Response · Authors · 2025-08-06
> **Gentle Reminder Regarding Rebuttal Period**
>
> **Dear Reviewer a4Mo**,
>
> We would like to kindly remind you that the *reviewer author discussion* period is entering its final few days. If you have any further comments, questions, or concerns regarding our responses, we would be truly grateful for your feedback.
>
>
> Your insights are extremely valuable to us, and we sincerely appreciate your time and effort in reviewing our work.
>
>
>
> **Best regards**,
>
> **Authors**

---

### Official Review · Reviewer_bX31 · 2025-07-04

**Clarity:** 4
**Significance:** 3
**Originality:** 4
**Rating:** 5
**Confidence:** 4

**Summary:**

This paper proposes Flexible Evidential Deep Learning (F-EDL), which extends EDL by predicting a flexible Dirichlet (FD) distribution parameterized by concentration parameters, allocation probabilities, and dispersion. The objective function combines expected MSE over the FD distribution with a Brier score-based regularization term, with a closed form. The advantage of the proposed method is demonstrated by both theoretical analysis and empirical experiments. It is proved the F-EDL is a generalization of traditonal EDL while enabling greater expressiveness.

**Questions:**

1. Can the authors explain the complementary strengths of softmax-based (MSP) and evidential (EDL) approaches? It is not clear when the softmax-based approach dominates and when EDL dominates, and how F-EDL leverages the complementary strengths to achieve optimal UQ.
2. Fig. 4a is weird. Can the authors confirm the accuracy drops when training size increases from 30k to 50k for EDL?

**Ethical Concerns:**

["NO or VERY MINOR ethics concerns only"]

**Limitations:**

Yes.

**Paper Formatting Concerns:**

No.

**Quality:**

4

**Strengths And Weaknesses:**

# Strengths:
1. It is novel to introduce the Flexible Dirichlet Distribution to evidential deep learning.
2. It provides closed form formulation of total, aleatoric, and epistemic uncertainty.
3. The form of objective function is elegant.
4. The theoretical analysis is solid and experiments are comprehensive.

# Weaknesses:
This is pretty solid paper. Honestly I did not find much signficiant weaknesses.

---

> ### Author Rebuttal · Authors · 2025-07-30
>
> Thank you for your careful review of our paper and for the insightful and constructive comments. Please find our detailed answers to your comment below.
>
> **1. (When does the softmax-based approach or EDL dominate, and how does $\mathcal{F}$-EDL combine its strengths to achieve optimal UQ?)**
>
> In short, on clean in-distribution (ID) data, EDL dominates, while for ambiguous or out-of-distribution (OOD) data, neither EDL nor softmax alone is sufficient—$\mathcal{F}$-EDL learns the optimal combination between the two predictions for each input.
>
> Specifically, on clean ID data, the total evidence $\alpha_{0}$ tends to be large compared to the dispersion parameter $\tau$, causing the predictive distribution to be dominated by the EDL component. Since EDL provides reliable uncertainty estimates in such regions, this results in well-calibrated predictions.
>
> However, in more challenging scenarios such as ambiguous or OOD data—where EDL may be less reliable due to its restrictive Dirichlet assumption—$\alpha_{0}$ is often not significantly larger than $\tau$. In these cases, $\mathcal{F}$-EDL learns to interpolate between the EDL and softmax predictions on a per-input basis. Specifically, it learns an adaptive combination of EDL ($p_{\text{EDL}}$) and softmax ($p_{\text{SM}}$) predictions, where the combination ratio is determined by the relative strengths of $\alpha_{0}$ and $\tau$.
>
> Theorem 4.5 formally characterizes this behavior, demonstrating that $\mathcal{F}$-EDL functions as an input-dependent mixture of the EDL and softmax models. This formulation allows the model to adaptively adjust its predictions based on the properties of each input, providing robust UQ across diverse data conditions. We will include an intuitive explanation of Theorem 4.5 in the main text to enhance clarity.
>
> **2. (Why does EDL accuracy drop as training size increases in Fig. 4a?)**
>
> This is an empirical pattern that emerged repeatedly across several runs. We attribute this counterintuitive behavior to the following contributing factors:
>
>  * **Unstable learning dynamics:** EDL is known to exhibit unstable training behavior under its original formulation. As the training data size increases, the model may fail to calibrate the predicted evidence accurately, leading to degraded performance. This issue has motivated the development of several variants designed to improve the training dynamics of EDL, including $\mathcal{I}$-EDL [1], R-EDL [2], and DAEDL [3]. For example, DAEDL (middle panel of Fig.4) applies spectral normalization to the feature extractor and adopts an alternative parameterization of the concentration parameters—both of which likely contribute to its comparatively more stable behavior.
>
> * **Misrepresentation of uncertainty:** As illustrated by the red dashed line in Fig. 4a, EDL’s average epistemic uncertainty *increases* as the training set size grows from 30k to 50k—an undesired behavior. Ideally, epistemic uncertainty should *decrease* as the model observes more data and gains confidence. This miscalibration suggests that the model becomes *more uncertain* despite increased supervision, which may account for the observed drop in test accuracy.
>
>
> We will include a brief discussion of this observation in the main text alongside the explanation of Fig. 4 to improve clarity.
>
>
> **References**
>
> [1] Deng et al., "Uncertainty estimation by fisher information-based evidential deep learning.", ICML 2023
>
>
> [2] Chen et al., "R-edl: Relaxing nonessential settings of evidential deep learning.", ICLR 2024
>
>
> [3] Yoon & Kim., "Uncertainty Estimation by Density Aware Evidential Deep Learning.", ICML 2024

---

> > ### Comment · Area_Chair_5kZt · 2025-08-04
> >
> > Dear Reviewers,
> >
> > Please engage in the discussion with the authors. The discussion period will end in a few days.
> >
> > Thanks,
> >
> > AC

---

> > ### Author Response · Authors · 2025-08-06
> > **Gentle Reminder Regarding Rebuttal Period**
> >
> > **Dear Reviewer bX31**,
> >
> > We would like to kindly remind you that the *reviewer author discussion* period is entering its final few days. If you have any further comments, questions, or concerns regarding our responses, we would be truly grateful for your feedback.
> >
> >
> > Your insights are extremely valuable to us, and we sincerely appreciate your time and effort in reviewing our work.
> >
> >
> >
> > **Best regards**,
> >
> > **Authors**

---

### Note · Authors · 2025-08-12

**Highlights of the Rebuttal and Discussion**

We summarize the key points clarified in our rebuttal; reviewer ID and comment number in parentheses indicate the corresponding point in our rebuttal (e.g., “a4Mo C1” refers to our response to Comment 1 under reviewer a4Mo’s review).

* **Relationship with recent theoretical critiques of EDL:** We showed that $\mathcal{F}$-EDL bypasses key assumptions in recent critiques (due to its flexible UQ structure and distinct loss) and empirically mitigates their implications. (a4Mo C4; jVmC C3).
* **Justification of the FD distribution and interpretation of its parameters:** We clarified the roles of $\mathbf{p}$ and $\tau$ (jVmC C2), analyzed learned parameter values for clean vs. ambiguous samples (jKtY C2), and explained how evidence is extracted under FD—showing it as a principled UQ model rather than an ad-hoc generalization (jVmC C5).
* **Loss/regularization choice:** We justified the use of Brier-based loss/regularization with theoretical rationale and ablation results, showing advantages over CE/KL (a4Mo C3; jKtY C4).
* **Cross-domain generalization, extensibility, scalability, and computational cost:** We demonstrated that $\mathcal{F}$-EDL generalizes effectively across domains (tabular, text, and large-scale vision) (jKtY C1), discussed its potential for trusted multi-view learning (jKtY C3), and clarified that computational overhead and inference time are minimal (jKtY C5).
* **Loss function concerns:** We explained why real-world data non-separability and spectral normalization implicitly bound parameter magnitudes, preventing divergence (jVmC C1)


**Planned Revisions**

We will carefully incorporate the clarifications above into the revised version. Moreover, we will add an intuitive explanation of Theorem 4.5 (bX31 C1; a4Mo C2), address the trend in Fig. 4a (bX31 C2), and expand the *Related Work* section to compare with R-EDL (a4Mo C5) and discuss recent theoretical critiques of EDL. These targeted updates will enhance clarity and completeness, and are readily incorporable into the revised version without changes to the core methodology.

We sincerely thank the reviewers and AC for their thoughtful consideration, which has strengthened both the presentation and the scope of our work.
We believe $\mathcal{F}$-EDL offers a principled and versatile advance in uncertainty quantification, with strong potential to benefit both theoretical research and practical applications across the UQ community.

---

### Decision · Program_Chairs · 2025-09-17

**Decision:**

Accept (poster)

**Comment:**

The authors elaborate on the evidential deep learning (EDL) approach to uncertainty quantification, in which a second-order distribution is modeled by a Dirichlet distribution. In order to increase expressivity, they propose to replace the Dirichlet by the Flexible Dirichlet (FD), which is a mixture of Dirichlet distributions. They propose a (regularised) loss to me minimised, analyse formal properties of the new model, and conduct experimental studies.

The reviewers are quite positive about the paper and highlight a several strengths, such as novelty, solid theoretical analysis, and especially the impressive empirical results. They also raised a number of critical comments and concerns, though most of them could be clarified during the discussion phase. So there is no doubt that this is a good paper.

During the discussion phase, I asked the authors a question regarding a promised improvement of their ICML submission. The answer was not really to the point. I was mainly referring to the problem that the second-order loss of EDL, without regularisation, is minimised by a degenerate solution (a Dirac delta function putting all mass on a single first-order probability), thereby suggesting zero epistemic uncertainty (Bengs et al. 2022, 2023); consequently, epistemic uncertainty is only incorporated "externally" through regularisation, and the amount of epistemic uncertainty expressed by the model is (artificially) controlled by how much one regularises. Instead, the authors answers referred to other issues, e.g., the problem of non-vanishing epistemic uncertainty, approximation of a reference distribution, etc.

Regarding the requirement to regularise the loss to keep the epistemic uncertainty from going to zero, it seems that this issue was also subject to discussion between the authors and a reviewer. Apparently, the flexible EDL proposed in the paper still has this problem, and the way to tackle it is via the spectral normalisation. The reviewers seem to be fine with this, though I have to say that control of epistemic uncertainty through regularisation of the loss still looks dubious to me.

Admittedly, I am also still unconvinced by the very motivation behind the generalised Dirichlet distribution, in particular the motivation for multimodality. For example, looking at the illustration in Fig. 3, why should a probability of 0.6 for digit 9 be plausible, and a probability of 0.8, too, but a probability of 0.7 not? Frankly, this doesn't make much sense to me, and it violated the commonly accepted property of coherence of beliefs.


Bengs et al. Pitfalls of epistemic uncertainty quantification through loss minimisation. NeurIPS 2022.

Bengs et al. On second-order scoring rules for epistemic uncertainty quantification. ICML 2023.